# To What Extent Do Flood-inducing Storm Events Change Future Flood Hazards?

Mariam Khanam[1], Giulia Sofia[1], and Emmanouil N. Anagnostou[1]

[1]Civil & Environmental Engineering, University of Connecticut, Storrs, 06269, USA

*Correspondence to*: Mariam Khanam (mariam.khanam@uconn.edu), Giulia Sofia (giulia.sofia@uconn.edu)

**Abstract**. Flooding is predicted to become more frequent in the coming decades because of global climate change. Recent literature has highlighted the importance of river morphodynamics in controlling flood hazards at the local scale. Abrupt and short-term geomorphic changes can occur after major flood-inducing storms. However, there is still a widespread lack of ability to foresee where and when substantial geomorphic changes will occur, as well as their ramifications for future flood

hazards. This study sought to gain an understanding of the implications of major storm events for future flood hazards. For this purpose, we developed self-organizing maps (SOMs) to predict post-storm changes in stage-discharge relationships, based on storm characteristics and watershed properties at 3,101 stream gages across the Contiguous United States (CONUS). We tested and verified a machine learning (ML) model and its feasibility to: (1) highlight the variability of geomorphic response to flood-inducing storms across various climatic and geomorphologic regions across CONUS, and (2) understand the impact

of these storms on the stage-discharge relationships at gaged sites as a proxy for changes in flood hazard. The established model allows us to select rivers with stage-discharge relationships that are more prone to change after flood-inducing storms, for which flood recurrence intervals should be revised regularly so that hazard assessment can be up to date with the changing conditions. Results from the model show that, even though post-storm changes in channel conveyance are widespread, the impacts on flood hazard vary across CONUS. The influence of channel conveyance variability on flood risk depends on various

hydrologic, geomorphologic, and atmospheric parameters characterizing a particular landscape or storm. The proposed framework can serve as a basis for incorporating channel conveyance adjustments into flood hazard assessment.

## 1 Introduction

Several factors contribute to the non-stationarity in flow regimes, including variations in human activities, changes in land cover and land use, climate changes, and low-frequency internal climate variability (i.e., multidecadal oscillations)

(Cunderlik and Burn, 2003; Mostofi Zadeh et al., 2020). Consequently, flood trends over the past decades have changed worldwide (Chang et al., 2007; FEMA, 2013; Karagiannis et al., 2017; McEvoy et al., 2012; Ziervogel et al., 2014), resulting in adverse impacts on society and the environment (Blöschl et al., 2019; Dottori et al., 2022, 2018; Hattermann et al., 2014; Milly et al., 2002; Mostofi Zadeh et al., 2020; Slater et al., 2015). Traditional "cause-effect" studies have focused on the time dependency or non-stationarity of individual hydrologic flood drivers (Alfieri et al., 2015; Khanam et al., 2021; Lisenby and

Fryirs, 2016; Mallakpour and Villarini, 2015; Mostofi Zadeh et al., 2020; Munoz et al., 2018). However, these studies might be under or overestimating the actual damage, especially in regions where the landscape is changing rapidly, because of the magnitude and prevalence of the hydroclimatic variability that is now underway. Nonetheless, the flood risk estimation traditionally has been based on flood frequency, derived from variability in streamflow, assuming constant channel capacity (Merz et al., 2012; Slater et al., 2015). The relationship between magnitude and frequency is also generally built upon the peak flow distribution, whereas peaks are discretized as either annual maxima or peaks over the threshold, but mostly assuming that river capacity remains constant over the investigation records. For decades, fluvial geomorphology research has focused on changes in river characteristics (Baker, 1994; Benito and Hudson, 2010; Stott, 2013). Various recent works (Ahrendt et al., 2022; Naylor et al., 2016; Slater et al., 2015, 2019; Sofia and Nikolopoulos, 2020a; Sofia et al., 2020; Stephens and Bledsoe, 2020, 2023) have suggested that the time has come to move beyond flood hazard assessment based on this "fixed river" idea. River channels and their adjacent floodplains continuously evolve because of the interactions of hydrology, landscape, and climate drivers and the interdependencies of processes at different spatial and temporal scales (Lane et al., 2007; Pinter et al., 2006b; Slater et al., 2015; Stover and Montgomery, 2001; Blench, 1969). Humans and water resources are intertwined, and they are now more than ever active players in these intricate geomorphic dynamics of rivers and floods (Ceola et al., 2019; Grill et al., 2019; Wohl, 2019). Rivers naturally modify their geometry (i.e., their breadth, depth, and slope) to reflect changes in discharge and sediment in the upstream catchment in addition to the obvious alterations brought on by human involvement (Lisenby et al., 2018). Any changes in these characteristics possibly will also alter the magnitude, frequency, and risk of future flooding.

The ability of rivers to store and move floodwaters downstream affects the probability that floods would destroy riverbanks or flood barriers, even while the total volume of water that flows through river systems during floods remains constant. Therefore, these abrupt changes in channel capacity alter flood properties, even when the magnitude of the flood remains unchanged (Blench, 1969; Criss and Shock, 2001; Lane et al., 2007; Neuhold et al., 2009; Pinter et al., 2008; Slater et al., 2015; Stover and Montgomery, 2001). Some obvious evidence of the effects of channel changes on flood properties (e.g. extent, depth, etc) has been presented by recurring flooding in different dynamic rivers (Brierley and Fryirs, 2016; Pinter et al., 2001; Zischg et al., 2018; Tate, 2019; Munoz et al., 2018). During these flood events, impacts are most evident at sites where the rivers' channel capacity has been drastically reduced (Munoz et al., 2018; Tate, 2019; Sofia et al., 2020). Neglecting the possibility of rapid changes in streamflow regime and channel conveyance capacity can conceal short-term shifts in flood threats. Li et al., 2020, for example, demonstrated that long-term trends comprise numerous short-term transients of much larger magnitude. These transient stages are often caused by abrupt scouring or deposition during flood-inducing storm events and are comparable in magnitude to long-term trends in peak streamflow. Additionally, short- and long-term climate variability can at the same time impact the streamflow patterns and channel conveyance changes, with the channel form adjusting to precipitation and sediment supply (Death et al., 2015; Rathburn et al., 2017; Ruiz-Villanueva et al., 2018; Scorpio et al., 2018; Surian et al., 2016; Wicherski et al., 2017). Figure 1, for example, shows changes in Boulder Creek in Colorado before and after a flash flood in 2013. Comparing the channel planform and width, it is evident the channel got wider after the flood.

Images from 2015 and 2019 show that the secondary channel on the right eventually disappeared, and the main channel acquired a more prominent bend than in the 2013 image. Such relatively quick alterations have the potential to further modify the geomorphic characteristics of rivers and to produce feedback that will affect the properties of future floods (depth, frequency, duration, and spatial extent).

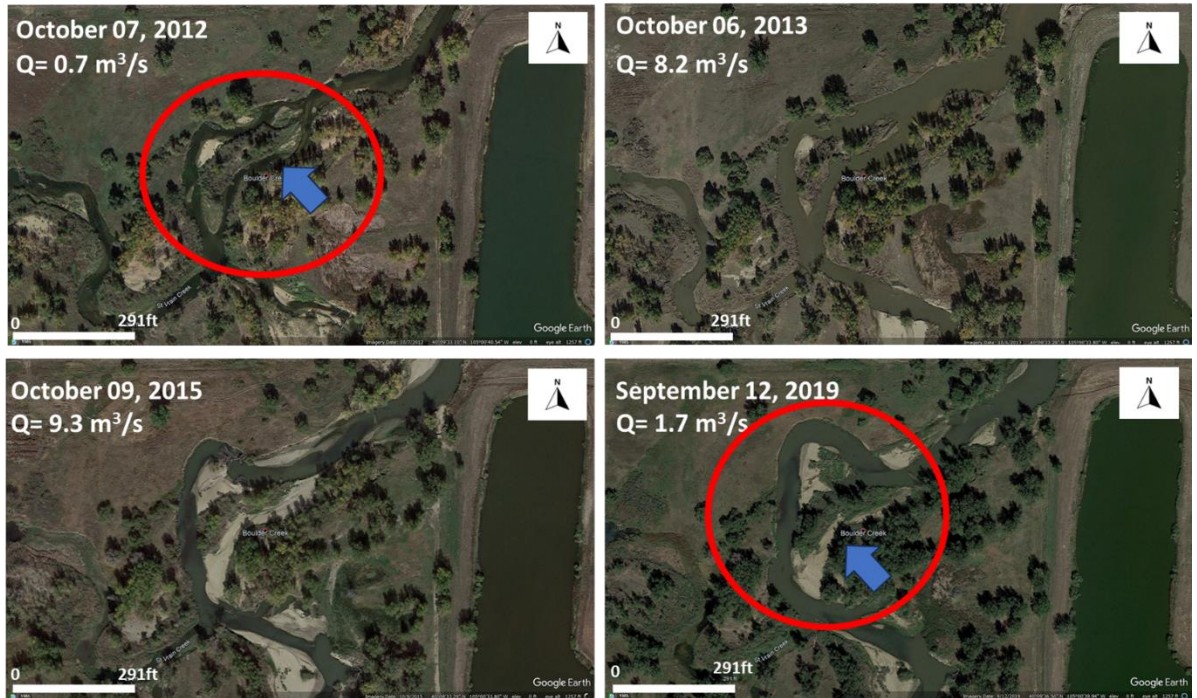

**Figure 1: Change in channel width in Boulder Creek, Colorado, before (2012) and after (2013-2015-2019) a flash flood in 2013 (Google Earth imagery). The Discharge reported here is the daily discharge measured at USGS 06730200 Boulder Creek at north 75th St. near Boulder, co. The red circles denote the section of the channel that has changed over the years and the blue arrow shows the missing channel from the year 2012 to 2019.**

Systematic shifts in a river's stage-discharge relationships identify the need for sharp upward revisions in hazard levels and stage-based flood-frequency analysis. Adjustments to the river stage-discharge relationship account for, at least partly, climate variability and long-term change. Nonetheless, while some river changes might be persistent in time, others could be more sudden and persist for a shorter time frame, like in the case of flood-inducing storms. These short-term channel changes are difficult to predict, but they could substantially increase the post-flood hazard, especially in the case of subsequent storms. Understanding the scale and severity of channel changes after flood-inducing storm events is key to improving flood management and building the resilience of critical infrastructure. What is missing from our current knowledge is a comprehensive study that shows the impacts of storm-induced channel changes on future flood hazards. Buraas et al., (2014) cited a general shortage of capability to predict where significant geomorphic changes will occur following flood-inducing events. Other authors have pointed to multidirectional approaches as promising contributions to the analysis of channel response to severe floods and the identification of controlling factors (Rinaldi et al., 2016; Scorpio et al., 2018; Surian et al.,

2016; Wicherski et al., 2017; among others). At regional scales, when it is often either impracticable or impossible to identify the precise events responsible for periods of channel shift, linking geomorphic cause and effect becomes increasingly difficult. However, this does not negate the requirement to comprehend and recognize short-term geomorphologic and hydrologic behavior that can exacerbate or mitigate flood threats. For this purpose, the availability of a large dataset representing a wide range of flood-inducing storm characteristics and channel morphology under different boundary conditions, such as underlying climatic, hydrologic, and geomorphologic settings, is crucial. This set of information forms a complex interacting system. The processes underlining these boundary conditions vary in spatial and temporal scale, and this calls for the use of improved analysis methods, able to draw predictions interlocking data of varying nature. In this context, machine learning (ML) is gaining popularity in the field of hydrology, geomorphology, and climate studies (Bergen et al., 2019; Schlef et al., 2019; Valentine and Kalnins, 2016), thanks to its ability to tackle coupled processes across space and time. Despite some limitations (Karpatne et al., 2019), and provided that the benchmark data used for the training are of high quality (Bergen et al. 2019), ML offers a valuable tool to gain new data-driven insights with high accuracy, transferability, and scalability (Houser et al., 2022; Sarker, 2021; Schlef et al., 2019; Sofia, 2020a).

In the context of river morphology, specifically, in the last few years recent studies highlighted their capability to predict channel types (Guillon et al., 2020), providing a geomorphological characterization of channels (Rabanaque et al., 2022), quantifying below-water (Woodget et al., 2019), or spatiotemporal changes (Boothroyd et al., 2021) in rivers, and guiding discharge estimation building from river morphology (Brinkerhoff et al., 2020), These works highlight how ML, when properly guided by field-based interpretation, can offer a valuable potential to push geomorphology into an increasingly predictive science (Fryirs and Brierley, 2022; Brierley et al., 2021). Tackling on the opportunities offered by ML potential, in this study, we sought to understand and predict the effects of flood-inducing storms on channel conveyance and, consequently, flood hazards. To achieve this, we have utilized stage-discharge "Residual" as a proxy of the channel capacity change, and we introduced an ML framework (section 2.3) that characterizes the interdependence of flood drivers, including atmospheric drivers (precipitation), hydrologic drivers (flow, stage), and geomorphologic drivers (channel width, depth, drainage area, geophysical characteristics). Overall, the analysis aims to: (1) highlight the variability of geomorphic response to flood-inducing storms across various climatic and geomorphologic regions in the Contiguous US (CONUS), and (2) understand the impact of these storms on the stage-discharge relationships at gaged sites as a proxy for changes in flood hazard. The study provided an independent test of discharge-based results and produced a tool for generating timely short-term updates of flood hazard estimates for dynamic rivers.

## 2 Materials and Methods

### 2.1 Quantifying the Impact on Flood Hazard

For this study, we used data from 3,101 U.S. Geological Survey (USGS) gaging stations distributed across the contiguous United States (Figure 2). The dataset allows us to cover a wide range of physiographic and climatic (See Fig. 2)

regions. We selected stations for which both historical field-measured data on channel properties and flood stages assigned by the National Weather Service (NWS) were available. The data for channel properties were retrieved following a procedure developed by (Slater, 2016; Slater et al., 2015) and using the codes provided by the authors at https://github.com/LouiseJSlater/Hydromorphology.

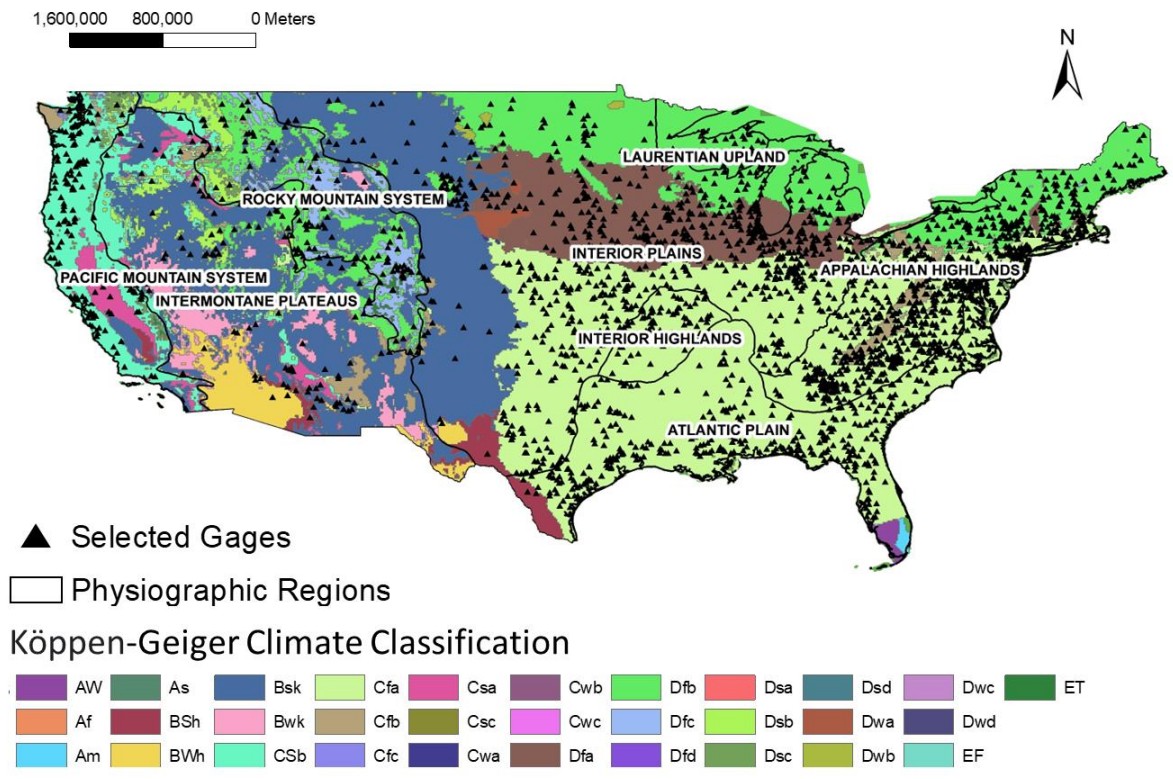

**Figure 2: USGS gage stations considered for in this study overlain on physiographic and climatic regions. For the acronym description of Physiographic regions and climate types please refer to Table A1 and A2 in the Appendix A.**

To model the average state of conveyance capacity for each stream gage site, we used theoretical single stage-discharge relationships (rating curves) at the height associated with the Flood Stage, as described by Slater et al. (2015). The Flood Stage, from the US National Weather Service, indicates a gauge height above which water level begins to impact lives and human activities, and it generally corresponds to the first flood warning threshold. The procedure, therefore, can be adapted for other gage datasets, in different parts of the world, by assuming similar warning thresholds. Deviations from the theoretical stage-discharge relationship indicate that at a moment in time, a discrete stage-discharge relationship existed, which highlights that there might have been temporal changes in channel conveyance. As described by Li et al. (2020), Slater et al. (2015, 2019), and Slater and Villarini, (2016), using a constant flood level enables the quantification of "conveyance residuals (Res)" that represent temporal changes in the discharge needed to reach the specific flood level (for example due to shifts in channel

capacity). In a temporal analysis of residuals, a positive to negative shift indicates a sudden decrease in channel capacity and a potential increase in flood hazard (Slater et al., 2015), as a lower discharge is needed to meet the warning threshold. We followed this procedure to capture the sudden changes in channel conveyance following major storm events. We focus mainly on sudden shifts, rather than on permanent shifts. The main reasons for this were, 1. short-term conveyance capacity changes are not considered in typical flood hazard assessments and could substantially overstate or understate flood threats at any particular time for subsequent floods; 2. there is a plethora of complex and sometimes not linear- processes and coupled feedback that we would need to 'model' in the training set, to provide a comprehensive benchmark to identify permanent shifts. This could be potentially interesting research that may be tackled by further studies building on our model.

To define the stage-discharge relationship, we considered only measured values of stage and discharge, as suggested in (Slater, 2016; Slater et al., 2015). Aside from considering consistent gages present in the Shen et al. 2017 database, and covered by stream measurements, we applied the same criteria as Slater et al. 2015, who only considered field measurements in which the discharge is within one percent of the product of channel velocity and cross-sectional channel area, as reported by the USGS, and those made close to the gage station. Following the work of (Slater, 2016; Slater et al.,2015, 2015a) we detected and excluded sites featuring artificial controls at the gauging station that could impede the natural adjustment of the channel's shape. Additionally, we eliminated all field measurements conducted at a different location or potentially different location, along with those taken in icy conditions, as these factors could impact the accuracy of channel geometry measurements. Our selection process retained only sites with comprehensive time series data, and as per Slater's et al. 2015 work, only kept gages with 99.7% completeness in streamflow records and 40 channel cross-section measurements. The stage-discharge relationship was evaluated through a Locally Weighted Scatterplot Smoothing (LOESS) fitting (Cleveland, 1979), as suggested by Li et al. (2020), Slater et al. (2015, 2019), and Slater and Villarini (2016). The fitting requires the definition of a smooth parameter, which we set automatically based on the bias-corrected Akaike Information Criterion (AIC) (Hurvich et al., 1998). We performed the analysis using the R package fANCOVA (https://CRAN.R-project.org/package=fANCOVA).

Before performing the above-mentioned steps, we excluded from the analysis measurements taken before the most recent datum change, if any reported measurement datum change was provided. We have excluded the gages that do not have continuous data for the timeframe from 2002-2013. By taking into account field data when the discharge was within a range of half the flood stage depth on either side of the flood stage, we also accepted the standards employed by Slater et al. (2015). We evaluated the readings visually to look for clusters of outliers in the scatterplots of the channel measurements that could be signs of changes in the measurement location (or datum). We systematically eliminated these metrics. According to the information of the gage, the measurements did not shift in location. For the work itself, consistently with Slater et al. (2015) and the open codes provided in her work, we removed all field measurements made in a location where there is known infrastructure like a bridge for example, and all field measurements made in icy conditions, as these might affect measurements of channel geometry. Figure 3 provides an example of changes in residuals after a flood-inducing storm event for the Quinnipiac River in Connecticut. From April 15 to April 18, 2007, a spring nor'easter hit the East Coast of the United States. The streamflow-gaging station recorded stages during this occurrence that were more than 0.2 meters higher than the FEMA-

projected 100-year levels  (Ahearn, 2009). For this gage, the flood stage is at 3.05m, the peak discharge of the 2007 event was 3.51m, and the Quinnipiac River itself (at the gage right upstream of the one in the figure) measured the maximum discharges for the period of record of the station during the 2007 flood. In figure. 3a, the stage and discharge data retrieved from field measurements taken after the flood appear to shift toward higher values of the stage for comparable discharges from before

2007. The curve fitted at the flood stage (black line in Figure. 3a) ultimately aligns between the two sets of data. Looking at the residuals concerning the fitted curve (Figure. 3b), the shift from positive residuals, before 2007, to negative is noticeable (outlier residual points were filtered out before performing the ML training). This suggests a loss of conveyance capacity due to deposition, assuming no changes in velocity. The time series of widths (Figure. 3c) and capacity (Figure. 3d) confirm this loss of conveyance; for this site, slightly changed channel widths (Figure. 3c) and an abrupt change in capacity (Figure. 3d)

can be seen. Such a change may result in a potential increase in flood hazard for a given flood volume.

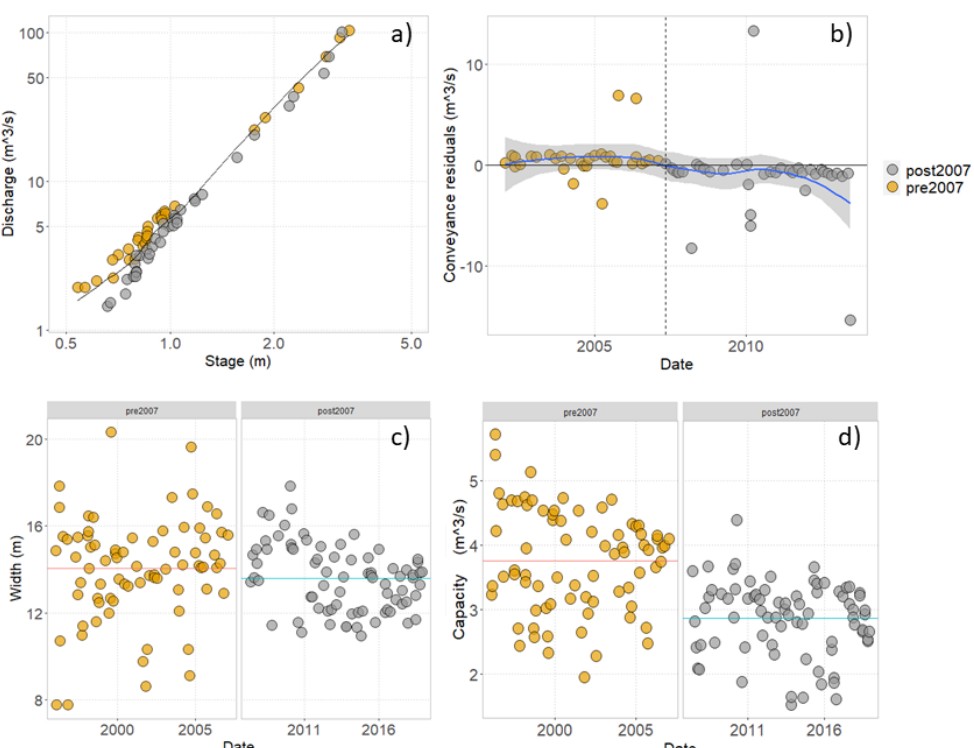

**Figure 3: Illustration of the conveyance analysis for the USGS stream gage QUINNIPIAC RIVER AT WALLINGFORD (USGS station 01196500), before and after the storm of April 2007. Stage-discharge relationship fitted to flood hazard level is shown in (a),**
**and residuals fitted to the rating curve in (b). In (b), some outlier residuals are evident, likely due to shifts in measurement locations. These points were filtered out before performing the ML training. Time series of channel widths as measured in the field (c) and channel capacity (d) are also shown, to highlight that possibly, the major change in residuals is due to a difference in channel depth, given a constant velocity.**

## 2.2 Considered Predictors

To obtain information on the watershed's hydrologic and geomorphologic properties, we collected data for each gage from the GAGES II dataset (Falcone, 2011). This dataset provides geomorphologic variables for each gage associated with watersheds' typical characteristics (e.g., Drainage area, Elevation, etc). These properties can be considered likely to change at a speed much slower than river discharge and localized channel measurements. Hence, we may consider these variables as 'static' in time. However, even if they are static in time, these characteristics are highly variable in space as they are spread
across the CONUS, providing us with a large sample of values for the ML training.

We also investigated several flood-inducing events that occurred from 2002 to 2013 in the same watershed and were included by Shen et al., 2017 in the flood event database. We ended up with 291201 events total for the 3101 gages. The minimum and maximum numbers of events per gage varied from 1 to 520. For each available field measurement of channel properties, we consider all the storms that happened in the previous 15, 30, 90, 180, and 365 days (accounting for the lag times
between each storm and the response of the river system) and calculate the median values of the storm characteristics (as defined in Shen et al., 2017; Table 1) in that timeframe excluding situations where we only had 1 storm. We only kept the gages in the analysis where we had more than 10 events. Therefore, for every single field measurement (i.e., dot in Fig 3) we had 5 different median storm characteristics – 1 median storm characteristic for the five different lag times considered, these medians represent a "typical flood-inducing storm" for that lag time, reducing the effects of small variability. The information
included by Shen et al., 2017 reported the percentile of the peak flows in the entire time series of the watershed, and all the reported events show a value greater than 80 for all storms. The reader should consider that while the median characteristic per se is not a 'severe' value, given the sample of data in Shen et al., 2017, it is a value representative of the typical event, for storms which in general encompass events having peak flows greater than 80th percentile of the entire flow series.

From these integrated data sources, we identified three groups of drivers: atmospheric, hydrologic, and
geomorphologic (Table 1). The integrated dataset provided direct and statistically derived information regarding flows and associated precipitation characteristics of each storm event.

**Table 1: variables Readers should refer to Shen et al. (2017) and Falcone (2011) for a complete description of the variables. Variables in bold letters are those used for ML analysis after the variable importance analysis.**

| Variable | Description | Unit | Data Source |
|---|---|---|---|
| | **Hydrologic variables** | | |
| TOPWET | Topographic wetness index | ln(m) | Falcone (2011) |
| HLR100M_SITE | Hydrologic Landscape Region (HLR) at the stream gage location. | unitless | Falcone (2011) |
| **Peak** | Peak flow associated with the storm event | | Shen et al. (2017) |
| **Res** | Residual | unitless | Estimated |
| **IBF** | Base flow index | $m^3/m^3$ | Shen et al. (2017) |

| | | | |
|---|---|---|---|
| Perc | Percentage of peak flow: The corresponding percentile of the peak flow in the entire flow series of the gauge | % | Shen et al. (2017) |
| Q2 | Second-order moment of the flow | unitless | Shen et al. (2017) |
| Els | Mean water travel distance to the drainage outlet | m | Shen et al. (2017) |
| EQ | Centroid of flow hydrograph | h | Shen et al. (2017) |
| Vt | Normalized flow volume ~ average flow volume per unit drainage area | mm | Shen et al. (2017) |
| HYDRO_DISTURB_INDX | Anthropogenic modification | unitless | Falcone (2011) |
| RunoffCoef | Runoff coefficient | unitless | |
| CLASS | Reference/non-reference class: REF = reference (least-disturbed hydrologic condition); NON-REF = not reference. | N/A | Falcone (2011) |
| BFI_AVE | Base Flow Index (BFI): Base flow to total streamflow ratio, given as a percentage ranging from 0 to 100. The persistent, slowly fluctuating component of streamflow that is commonly attributed to ground-water discharge to a stream is known as base flow. | % | Falcone (2011) |
| RFACT | Rainfall and Runoff factor | 100s ft-tonf in/h/ac/yr | Falcone (2011) |

| **Geomorphologic variables** | | | |
|---|---|---|---|
| GEOL_REEDBUSH_DOM | Dominant (highest percent of the area) geology, derived from a simplified version of Reed & Bush (2001) - Generalized Geologic Map of the Conterminous United States. | N/A | Falcone (2011) |
| STREAMS_KM_SQ_KM | Stream density, km of streams per watershed sq km, from NHD 100k streams | km/sq km | Falcone (2011) |
| STRAHLER_MAX | NHDPlus's maximum Strahler stream order in the watershed. | unitless | Falcone (2011) |
| MAINSTEM_SINUOUSITY | Sinuosity of mainstem streamline | unitless | Falcone (2011) |
| ELEV_MEAN_M_BASIN | Mean watershed elevation (meters) from 100m National Elevation Dataset | m | Falcone (2011) |
| ELEV_MAX_M_BASIN | Maximum watershed elevation (meters) from 100m National Elevation Dataset | m | Falcone (2011) |

| | | | |
|---|---|---|---|
| **ELEV_MIN_M_BASIN** | Minimum watershed elevation (meters) from 100m National Elevation Dataset | m | Falcone (2011) |
| **ELEV_MEDIAN_M_BASIN** | Median watershed elevation (meters) from 100m National Elevation Dataset | m | Falcone (2011) |
| **ELEV_STD_M_BASIN** | Standard deviation of elevation (meters) across the watershed from 100m National Elevation Dataset | m | Falcone (2011) |
| **ELEV_SITE_M** | Elevation at gage location (meters) from 100m National Elevation Dataset | m | Falcone (2011) |
| **RRMEAN** | Dimensionless elevation - relief ratio, calculated as (ELEV_MEAN - ELEV_MIN)/(ELEV_MAX - ELEV_MIN). | unitless | Falcone (2011) |
| RRMEDIAN | Dimensionless elevation - relief ratio, calculated as (ELEV_MEDIAN - ELEV_MIN)/(ELEV_MAX - ELEV_MIN). | unitless | Falcone (2011) |
| SLOPE_PCT | Mean watershed slope | % | Falcone (2011) |
| **ASPECT_DEGREES** | Mean watershed aspect | degrees (0-360) | Falcone (2011) |
| **ASPECT_NORTHNESS** | Aspect "northness". Ranges from -1 to 1. A value of 1 means the watershed is facing/draining due north, and a value of -1 means the watershed is facing/draining due south | unitless | Falcone (2011) |
| **ASPECT_EASTNESS** | Aspect "eastness". Ranges from -1 to 1. A value of 1 means the watershed is facing/draining due east, and a value of -1 means the watershed is facing/draining due west | unitless | Falcone (2011) |
| **Physio** | Physiographic divisions of CONUS | N/A | (Fenneman and Johnson, 1964) |
| **DRAIN_SQKM** | Drainage area | km$^2$ | Falcone (2011) |
| **Atmospheric variables** | | | |
| **CovTrLs** | Covariance of precipitation and water travel distance | mh | Shen et al. (2017) |
| **Etr** | Centroid of precipitation | h2 | Shen et al. (2017) |

| | | | |
|---|---|---|---|
| **VarTr** | Spreadness of precipitation | h$^2$ | Shen et al. (2017) |
| **VarLs** | Variance of water travel distance | m$^2$ | Shen et al. (2017) |
| **Vb** | Base flow volume | mm | Shen et al. (2017) |
| **Vp** | Precipitation volume | mm | Shen et al. (2017) |
| **Pmean** | Mean Precipitation | mm/h | Shen et al. (2017) |
| | Climate types (was not included in the ML model) | unitless | (Beck et al., 2018) |

## 2.3. Modeling the Impact of Flood-inducing Storms

The ML-based methodology developed in this study for predicting the median "Residual" is based on clusters of the gages. Using a self-organizing map (SOM) with event-specific characteristics, explained in Table 1, we developed a framework for understanding and predicting channel changes due to flood-inducing storm events. The SOM developed by (Kohonen, 1982), is one of the most popular clustering/ classification methods used in many research areas such as medical science, hydrology, and signal processing (e.g., (Zanchetta and Coulibaly, 2022; Rahmati et al., 2019). The SOM method has become

a very useful prediction tool in hydrological and environmental studies because it can predict a target variable without learning any physical relationship among a collection of variables. The main advantage of the SOMs is that they allow to reduce the data dimensionality, by organizing the data into a two-dimensional array (Kohonen, 1982) using topology-preserving transformations (Rahmati et al., 2019). SOMs, being a form of artificial neural network, can be thought of as a regression technique with a higher level of nonlinearity between the dependent and independent variables (Geem et al., 2007). The

proposed SOM framework (Figure 4) consisted of four phases: unsupervised clustering, supervised mapping, trained regression, 10-fold validation, and prediction. The whole procedure is described in the sub-sections below.

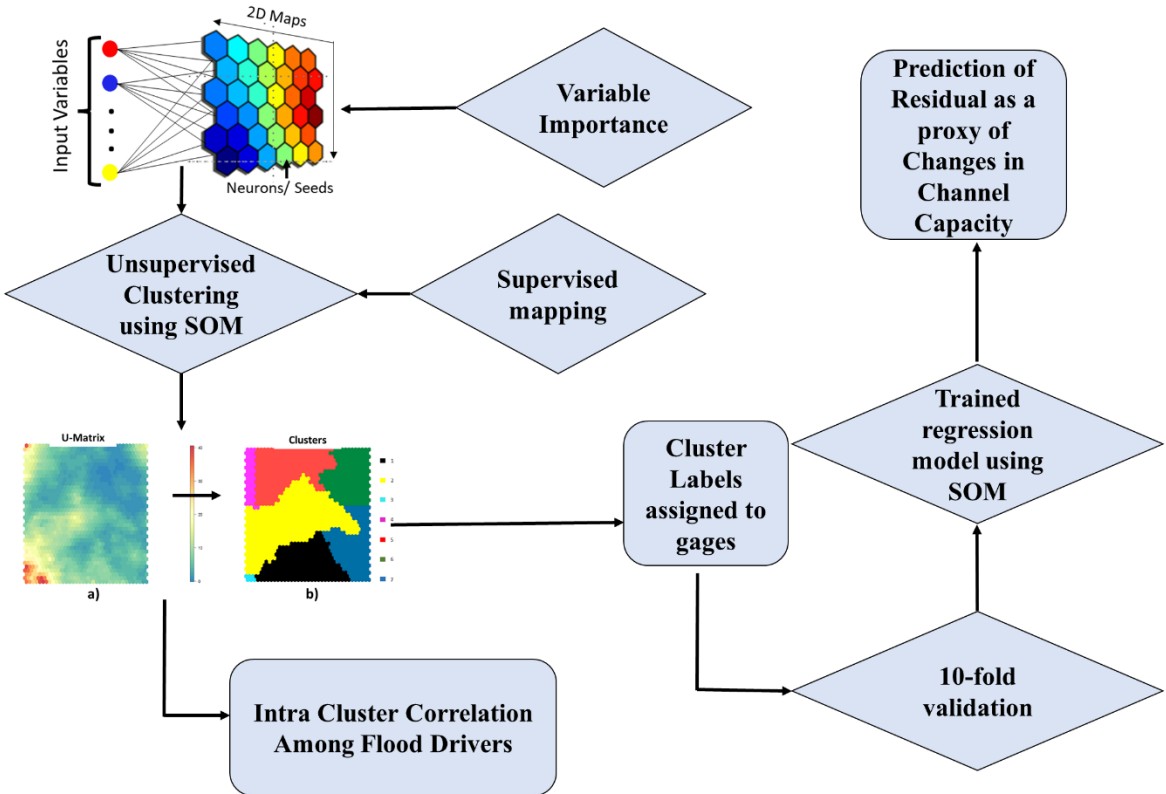

**Figure 4: Schematic of the SOM framework proposed in this study.**

The SOM algorithm is technically conceived for numerical datasets. This means that SOMs cannot be used to analyze variables with non-numerical data types, such as categorical values. To present the categorical variables to the machine learning model selected for this study, we therefore converted all the categorical values into binary digits. Each binary digit was then transformed into one feature column.

Most storm variables (except Perc- Percentage of peak flow and Percentile-Percentile corresponds to peak flow) were normalized considering the range of values available for each station. This normalization was performed to account for the influence of the watershed sizes on the various storm properties. Continuous geomorphologic and hydrologic variables, not coded in the range 0-1 (or 0-100) (aside from RRMEAN-Mean relief ratio and RRMEDIAN- Median relief ratio, SLOPE_PCT- Mean watershed slope, and Aspect) were normalized considering the overall range across CONUS. The stage-discharge residuals were kept as is because they are already "relative" in value to the stage-discharge fitted at flood stage for each gage. To reduce the dataset dimensionality, and avoid collinearity, we performed a variable importance analysis using the misclassification rate (section 2.3.1).

### 2.3.1. Unsupervised Clustering

The first module used, a SOM algorithm to cluster together gages based on similar characteristics. The main objective of this step is to group gages having similar underlying patterns of variables. The SOMs are organized in two-dimensional space where the neighboring neurons learn similar patterns, and neurons mapped far away have dissimilar patterns (Stefanovič and Kurasova, 2011) This unsupervised mapping was performed automatically using the Kohonen package in R (Wehrens and Kruisselbrink, 2018; Wehrens and Buydens, 2007; Kohonen, n.d., 1982; Wehrens, 2019). The optimal number of nodes was set at five times the square root of the number of observational data, as per Kohonen's general rule of thumb for determining the sizes of two-dimensional grids (Fytilis and Rizzo, 2013).

Typically, SOM data clustering involves two steps: first, the data set is clustered using SOMs, which offer the organization of the data into the various nodes, and then the nodes are clustered (Vesanto and Alhoniemi, 2000) Clustering speeds significantly increase when nodes are used in place of actual data. The result of the first step is that gages are grouped in neighboring nodes if the underlining patterns of variables are similar. After the SOM is trained, its U-matrix gives insight into how all the data are organized, as it displays the nodes and the distance that the weight nodes create between each weight and all its neighbors. This matrix can be used for the second step of identifying and labeling the actual clusters, through image-analysis tools (Pacheco et al., 2017; Wang et al., 2010; Wu and Li, 2022; Vincent et al., 1991). In this work, the first unsupervised clustering was accomplished by using all the data together, including the residuals in the process. Each gage was assigned a cluster number based on all the variables of that location. Gages grouped in the same cluster are expected to have similar patterns of the input variables, including the residuals. For each cluster, then, we re-train the model, retaining only the gages for that cluster, to provide the most typical residual given by the combination of hydrologic, geomorphologic, and atmospheric variables.

The most common approach is to segment the U-matrix using the watershed technique of gray-scale image processing (Costa and Netto, 1999; Vincent et al., 1991). Using a watershed analogy, the U-matrix (Figure 5) can be used to locate the clusters. Large "heights" and ridges imply significant distances in the feature space, while little "valleys" represent data subsets that are similar (Ultsch and Lötsch, 2017). The segmentation is performed by flooding the valleys (similar nodes with very close distances from one to the other) until a ridge (high dissimilarity) is reached. Where the water converges, watersheds will form, having close boundaries. One cluster is represented by all the items in a segmented area or watershed. According to this approach, a minimum height threshold can be selected to define the clusters (valleys). We followed automatic thresholding and set the threshold to a statistical value equal to half the standard deviation of the values. To perform this step, we applied watershed transformation and watershed-based object detection using the function "watersheds" in the R Bioconductor package (Torres-Matallana, 2016).

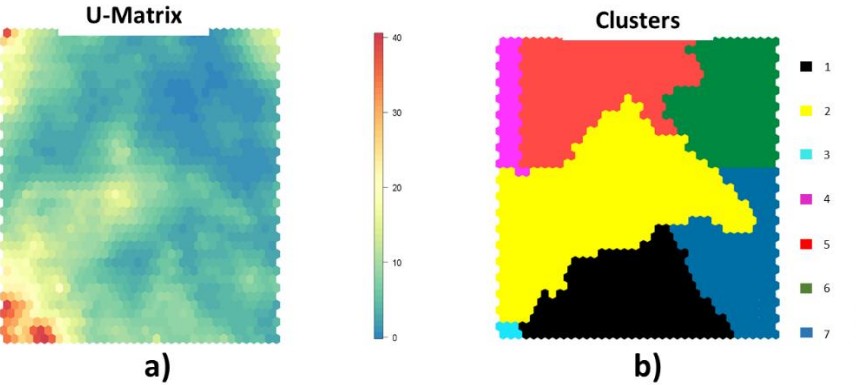

**Figure 5: Example of (a) U-Matrix and (b) derived clusters. Red colors in the U-Matrix stand for significant distances in the feature space, whereas blue colors are "valleys" that group subsets of related data. The watersheds shown in (b) are collections of related data.**

We assessed the relevance of each feature according to its misclassification rate relative to a baseline cluster assignment produced by a random permutation of feature values to find the most crucial features and prevent data duplication (Molnar, 2022; Breiman, 2001; Fisher et al., 2018). We preferred this approach considering that permutation feature importance does not call for retraining of the model before the analysis. This approach states that a variable (feature) is "important" if changing its values results in a cluster reassignment because, in this scenario, the model primarily relies on that

feature to forecast the predictors. In contrast, a feature is considered "unimportant" if changing its values does not affect the anticipated cluster. The variable identified as important with the shuffling does not necessarily mean they have high variability among watersheds. It rather means that this variable is highly correlated with the target variable (the cluster association), because shuffling its values essentially destroys any relationship between that feature and the target variable, as indicated by the decrease in the training performance. After randomly permuting the values of a feature, the model is NOT refitted to the

training data. This technique has been recognized in the literature (e.g., (Breiman, 2016; Wei et al., 2015; Fisher et al., 2018) and it is widely implemented in many statistic packages as well (e.g., Biecek et al., 2018, 2019; Molnar & Schratz, 2008) Please refer also to Wei et al (2015) for a review. We ran the clustering algorithm 10 times with different seeds. At each run, we trained the clustering using 90% of the data and predicted the remaining 10%; and, for each run, each feature of the dataset was permuted 10 times. The permutation misclassification rate of a feature was calculated as the number of observations for

which the cluster assignment differed from the original cluster assignment, divided by the number of observations given a permutation of the feature. The overall average misclassification rate iterations were interpreted as variable importance. We decided to keep only the variables producing a misclassification rate higher than the mean values. Figure 6 shows the most important variables for the interval N = 365 days. This variable selection indirectly checks for collinearity by keeping only the variables that have the largest effect on the changes.


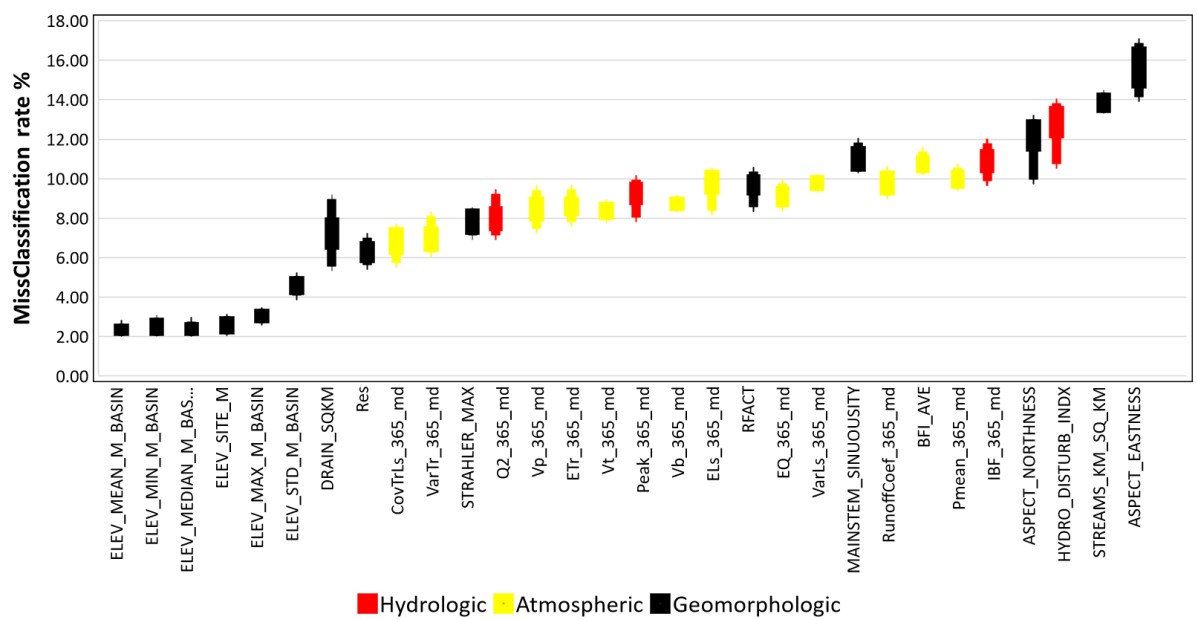

**Figure 6: Selected variables based on misclassification rate (%).**

### 2.3.2. Supervised Mapping and Trained Regression

Self-organizing maps (SOM) are extensively applied for clustering and visualization purposes. Nonetheless, they can
be used for regression learning. (e.g., (Riese and Keller, 2019, 2018)). In the first step, the data (geomorphological, atmospheric, hydrologic variables, and measured residuals) are clustered together, based on patterns of variables. The resulting SOMs are composed of nodes, each of which is connected to a "weight" vector that represents the node's location in the input space. The map can be used to categorize further observations after training by locating the node whose weight vector is closest to the input space vector (best matching unit, or BMU).

300       The regression algorithm of the SOM proceeds similarly to the clustering SOM algorithm. However, the regression differs for these main points: 1) Within the finalized input SOM that was created in the first stage, the BMU search is carried out.; 2) For the regression instance, the weights of the supervised SOM are based on one single parameter (a continuous number, which in our case is the residuals). Combining the unsupervised and supervised SOM allows for the selection of the BMU for each data point while also connecting the chosen best-matching unit to a particular residual estimation. In other
words, each gage is mapped to a certain cluster, based on the median characteristics of the storms. For the regression part, the data extracted from the SOM are restricted to the best matching cluster, and given the input storm and watershed properties, we can predict the most likely residual.

For the supervised mapping and trained regression step, the gages were tagged to their corresponding SOM clusters. Once a cluster is defined, we aimed to determine which features were the most significantly correlated. For this, we considered
the distance correlation index (dCorr) (Székely et al., 2007) to quantitatively identify the correlation of the important variables

with the residuals within each cluster. The range of dCorr values, from 0 to 1, represents the dependence of two independent variables. The stronger the dependence, the closer the value is to 1, and the statistical independence of the two variables is implied by a value of zero (Sofia & Nikolopoulos, 2020). We used inverse distance correlation (1-dCorr) to measure the dissimilarity of the variables within the cluster and create organized dendrograms. The attribute distances between every pair of drivers that have been successively clustered are depicted in a dendrogram.

Having tagged the gages, we performed supervised training with them to predict the residuals based on the atmospheric, hydrologic, and geomorphologic variables. The main outcome of this part is to have an ML system able to predict the most probable residual after a storm having certain properties, for a location with specific watershed characteristics. To this point, we retrained the SOMs independently for each cluster, using only the data retrieved from the stations within that cluster. For this part, we applied an extension of Kohonen's self-organizing map algorithm, the growing self-organizing map (GSOM) (Alahakoon et al., 2000; GrowingSOM package | R Documentation, 2020, https://rdrr.io/cran/GrowingSOM/). We chose GSOM to refine the analysis and improve the prediction within each cluster. The GSOM hierarchical clustering technique enables the data analyst to locate important and unique clusters at a higher level and to focus on a more precise grouping of the interesting clusters only. (Alahakoon et al., 2000). The GSOM is computationally expensive, so we decided to apply it to the already clustered data. A spread factor parameterizes the GSOM. This measure can generate maps of different sizes without previous knowledge about the dataset, samples, or attributes. We set the spread factor to 0.8, as suggested by Alahakoon et al. (2000).

Finally, we trained the model by selecting 90% of the data randomly and validated its performance using the remaining 10% for each cluster. The traditional method of identifying the quality of the SOM, proposed by Kohonen, is to compute the quantization error by summing the distances between the nodes and the data points, with smaller values indicating a better fit. This method has been used successfully by many researchers, requiring minimal computation time, to compare changes across time-series images (e.g., (Bação et al., 2005; Dresp et al., 2018; Wandeto and Dresp-Langley, 2019). For quality assessment, we also followed the approach used by (Swenson and Grotjahn, 2019). We performed cross-validation for a particular SOM, fitting the SOM to the data first to ensure a unique cluster assignment. Then we conducted 100 trials, excluding the data used in initialization, as suggested by Swenson and Grotjahn (2019). We utilized a typical subdivision of 90-10, which meant that 90% of the data was used as training data to fit a new SOM, and the SOM was then utilized to forecast the cluster assignments of the remaining 10% of validation data. The percentage of gages whose cross-validation cluster assignment changed from the original assignment in at least 10% of the 100 trials was calculated. We further tested the quality of the ML by evaluating the RMSE and the correlation distance between the actual residuals and the predicted ones for the validation dataset.

### 2.3.3. Predicting Major Storm Effects on Flood Hazard

Using the trained model (section 2.3.2), we predicted the residuals for each gaging station, based on all the variables (table 1) selected from Shen et al. (2017), Falcone (2011), and Fenneman and Johnson, (1964). We compared the predicted

residual for a given storm at a given gage with the average residual measured in the most recent years focusing on prediction showing a sudden deviation from positive (before the storm) to negative (post-storm). This sudden deviation, as illustrated in Figure 3, can indicate a quick shift in channel conveyance in response to sediment deposition, which can trigger increased flood hazard even when the flood event's return period remains unchanged (Blench, 1969; Lane et al., 2007; Pinter et al., 2006b, a; Stover and Montgomery, 2001).

To highlight the criticality of this sudden shift, we considered as highly at risk those watersheds for which the predicted residual, shifting from positive to negative, was outside the lower bound of the 95% confidence interval of the current stage-discharge relationship. As LOESS smoothers fit a unique linear regression for every data point by including nearby data points to estimate the slope and intercept, the correlation in nearby data points helps ensure obtaining a smooth curve fit. Therefore, the $\mu+1.96\sigma$ of the nearby data points considered for each fitted value can be considered as a measure of the 95% confidence interval. This information is calculated directly from the R package fANCOVA (https://CRAN.R-project.org/package=fANCOVA) used for the fitting. Overall, a watershed having positive residuals for the most recent measurements, for which we predict a sudden shift to negative outside the confidence bound of the stage-discharge curve, represents a critical condition that should be monitored, as the current flood stage might underestimate the flood risk.

## 3. Results Analysis

### 3.1. Variable Importance

Figure 6 demonstrates the outcome of the variable importance. Based on the results shown in Figure 6, we found that the same variables were always important for all interval analyses. Table 1 shows all the selected variables in bold for N = 365. In this case, out of a total of 40 variables we have selected 30 based on the misclassification rate (%). Of the selected variables 15 were geomorphologic variables, followed by 10 atmospheric variables and 5 hydrologic variables. The most important variables were the Aspect (ASPECT_NORTHNESS, ASPECT_EASTNESS), and stream density (STREAMS_KM_SQ_KM). The most important hydrologic variable was HYDRO_DISTURB_INDX, which explains the condition of the watershed, whether it is anthropogenically modified or natural.

### 3.2. Evaluation of SOM accuracy

The quantization error (Table 2) provided a measure of the accuracy of SOMs. The quantization error reported a higher accuracy as the number of training samples increased (increasing the number of days, resulting in more channel measurement and flood properties for each training sample). Homogeneous areas in the U-Matrix became more evident (Figure 7) as the quantization error diminished (Table 2). As Table 2 indicates, the 365 days interval had the best quality, as represented by the lowest quantization error. For this reason, the following sections will present an investigation of the maps produced with this interval. Table 2 also shows the SOM quality in terms of distance to the closest units of the SOMs trained for each

cluster. The results suggest that the retraining of the individual cluster using GSOM improved the prediction quality of the SOM significantly.

Table 2 also represents the correlation distance and RMSE between the measured and predicted residuals for each cluster of the validation datasets. The average correlation was close to 1 for all N values, suggesting the performance of the SOM model was satisfactory. The average RMSE was in the range of 0.09 – 0.14 m, which indicates a low random error relative to the dynamic range (-3 to 3) of the predicted variable. Both the unsupervised correlation distances and the average correlation showed the best results for N- 365 days. The RMSE diminished with the increase in the interval.

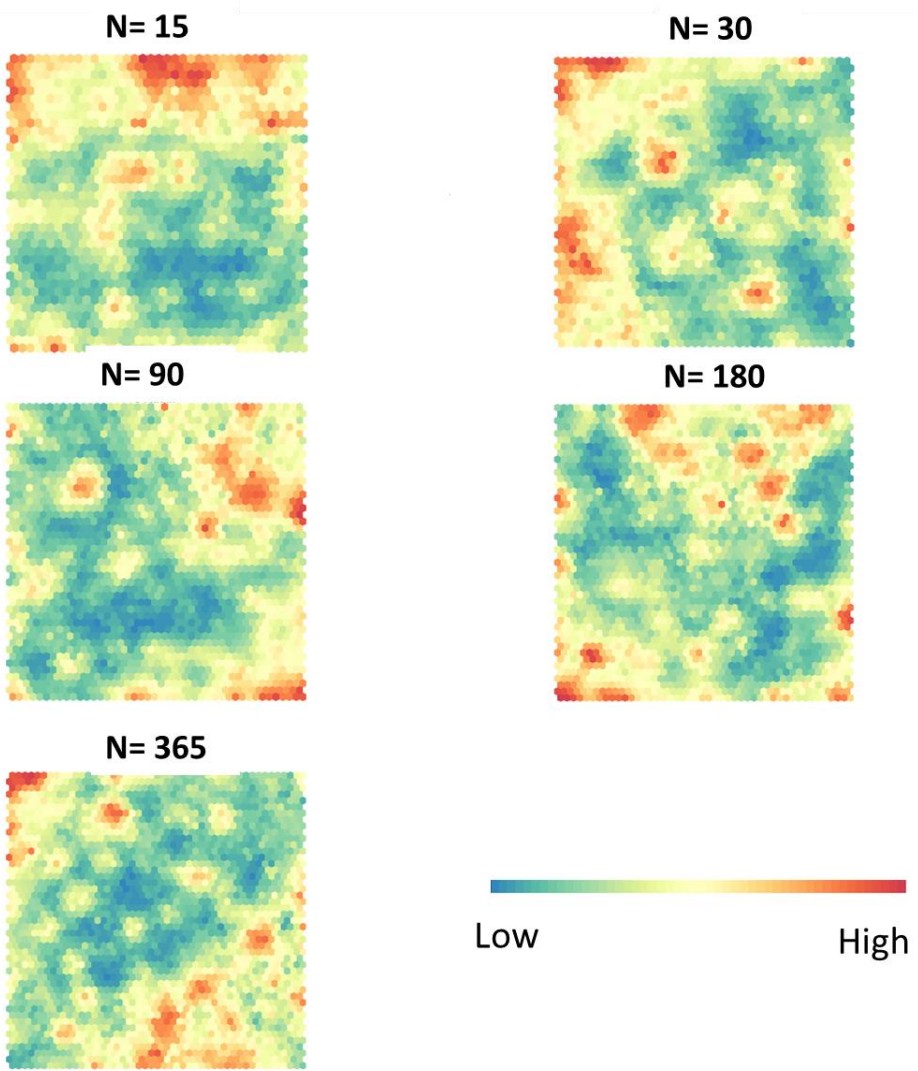

**Figure 7: U-Matrix for different intervals (N days). The red colors represent large distances in the feature space, while the blue colors represent "valleys "grouping subsets of similar data.**

**Table 2: Accuracy assessment parameters of the ML analysis. This table reports the average correlation and RMSE between the predicted and observed residuals for the different intervals.**

| Interval (days) | Avg. Corr. (10-fold) | Avg. RMSE (m) (10-fold) |
|---|---|---|
| 15 | 0.81 | 0.13 |
| 30 | 0.84 | 0.14 |
| 90 | 0.80 | 0.13 |
| 180 | 0.80 | 0.09 |
| **365** | **0.86** | **0.09** |

Figure 8 presents the results of the unsupervised clustering for N = 365 for the variables used. In the figure, the contrast between high (red) and low (blue) value areas emphasizes the spatial patterns of the various parameters we investigated. Based on this clustering, a combined U-Matrix is produced (discussed in Figure 7) and a cluster label is assigned to each gage. Gages with similar characteristics presented by the variables are tagged with the same cluster number. There are 12 clusters of gages for 365 days interval and we have plotted the clusters individually on a map showing how they spread across different physiographic regions and climate zones in Figure A1 in Appendix A. Clustering does not have a geographical meaning, rather gages behave more consistently between adjacent clusters than non-adjacent clusters, but this does not necessarily follow the spatial proximity of the gages. This is reflected in the spatial pattern of the different clusters of gages in Figure A1.

Visually, the SOMs in Figure 8 highlight the co-oscillation of hydrologic and geomorphologic variables as a standard component of watershed behavior. Drainage area (DA) and discharge/peak flow (Peak), for example, are positively correlated, with a cluster of high values in the bottom part of the SOMs. We can see that, other hydrologic variables like ELS (Mean water travel distance to the drainage outlet), EQ (Centroid of flow hydrograph), Q2 (Second-order moment of the flow), Vp (Precipitation volume), Vt (average flow volume per unit drainage area), and VaTr (Spreadness of precipitation), have similar patterns. The centroid of precipitation (EQ) and hydrograph (ETr) appear to be highly correlated. Some specific co-oscillations of variables are evident in multiple regions. Percentage (Perc) and percentile (Percentile) of peak flow show the highest values spread across the SOM nodes. If we focus on the SOM of "Res", we can see that the nodes on the righthand side of the SOM seem to be associated with high values of the residuals (Figure 8). Nevertheless, a small cluster of high residuals is seen in the upper lefthand corner. At the global level, this highlights a lack of regional synchrony in stage-discharge shifts at the yearly scale. (Pfeiffer et al., 2019) reported similar findings on the decadal scale.

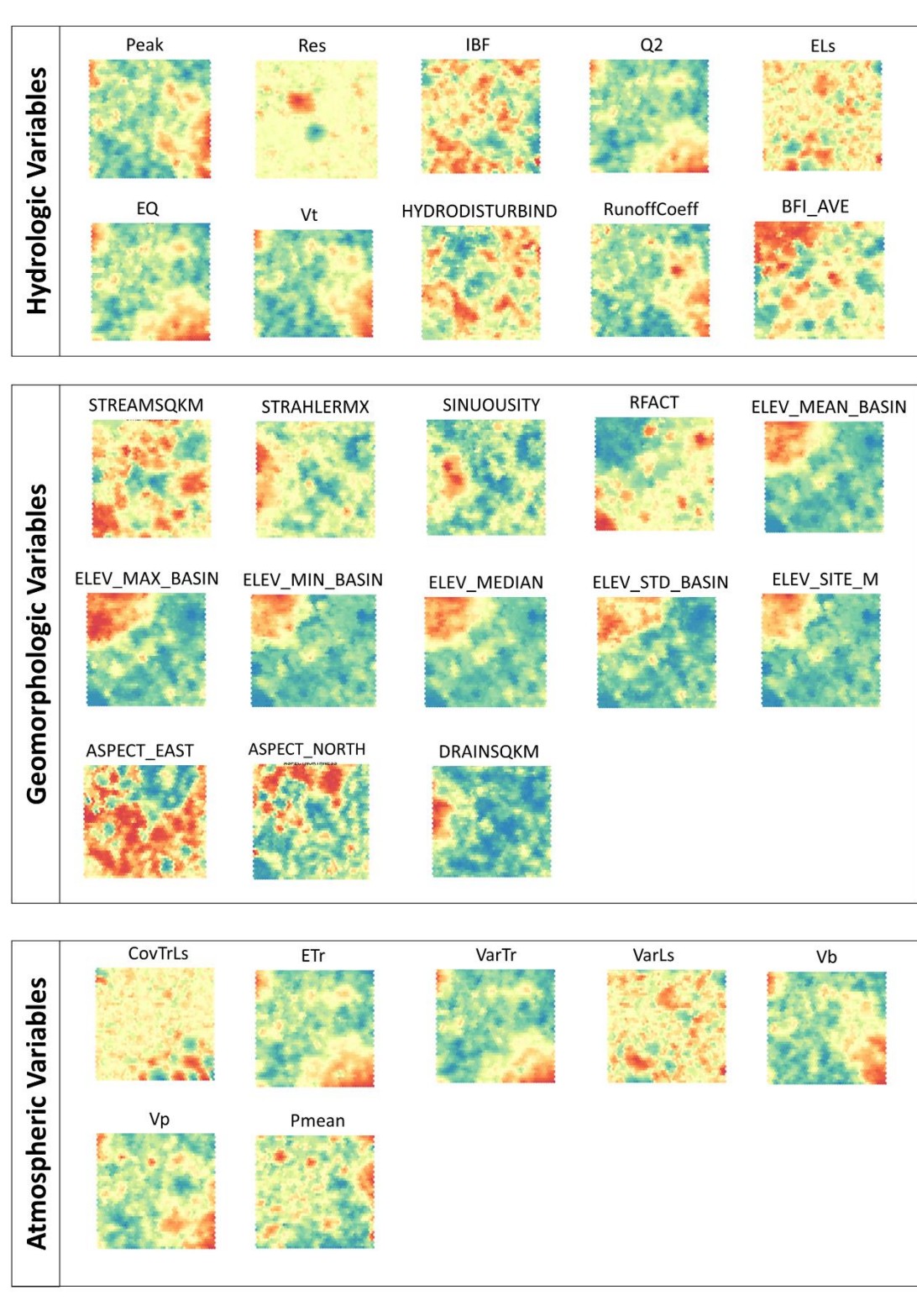

410

**Figure 8: Individual SOMs of all the flood drivers for N= 365. Similar to the U-Matrix the red colors represent large distances in the feature space, while blue colors represent "valleys "grouping subsets of similar data.**

### 3.3. Variables Associated with shifts in the residuals.

Focusing on the changes in the stage-discharge relationship residuals (Res), we next investigated the correlation between predicted and measured residuals on the one hand and other variables on the other (Figure 9, Table 3). For the proposed ML framework, the training was unsupervised. In general, the predicted and measured residuals were highly correlated, validating the SOM performance. Table 3 summarizes the correlations among the considered predictors in Figure 9 for N = 365 days. It presents the analysis of the group of variables based on the dendrogram branches for different likelihood of change levels (e.g., 0–10%, 10–30%, and 30–50%). This section discusses the correlations for the 30–50% category as an example; the other two categories showed similar outcomes. We do not have more than 50% here in the table because the highest percentage of gages that showed sudden change was 30-50%. In Table 3, level 1 shows the group of variables highly correlated to each other and with residuals. Level 2 shows variables that are highly correlated to each other but related to a lesser degree to the variables in Level 1.

For level 1, the physiography of the basins is represented by ELEV_* (Elevation) highly correlated with EQ (Centroid of flow hydrograph), Q2 (Second-order moment of the flow), and ETR (Centroid of precipitation), which are correlated with all the other variables in Group 2. For level 2, Residuals (Res) are shown to be correlated with different variables. A noticeable pattern is group 1 contains mostly hydrologic variables, while group 2 contains atmospheric variables. In group 1, the residuals (Res) belong to the tree containing the variables RFACT (Rainfall and Runoff factor), HYDRO_DISTURB_INDX (Anthropogenic modification), STREAMS_KM_SQ_KM (Stream density), BFI_AVE (Base Flow Index), ASPECT_NORTHNESS, ASPECT_EASTNESS, STRAHLER_MAX (Maximum Strahler stream order in the watershed), MAINSTEM_SINUOUSITY (Sinuosity), DRAIN_SQKM (Drainage area), Peak (Peak flow), and CovtrLs (Covariance of precipitation and water travel distance ) (level 2 in table 3). RFACT- Rainfall runoff factor, directly affects rainfall runoff influencing the channel changes. HYDRO_DISTURB_INDX (see section 3.1) represents the channel condition, whether the channel is altered by manmade construction or not. A group of highly connected elements comprises a series of drainage properties (STREAMS_KM_SQ_KM, STRAHLER_MAX, MAINSTEM_SINUOUSITY, DRAIN_SQKM) that modulate the way precipitation is routed through the basin and directly affect flood properties.

In level 2, group 2, the tree contains Pmean (Mean Precipitation), ELS (Mean water travel distance to the drainage outlet), EQ (Centroid of flow hydrograph), Q2 (Second-order moment of the flow), Vp (Precipitation volume), Vt (average flow volume per unit drainage area), and VaTr (Spreadness of precipitation), VarLs (Variance of water travel distance), Vb (Base flow volume), and RunoffCoef (Runoff coefficient). These are mostly related to rainfall properties. While they are important fingerprints for the attribution of regional flood changes, these variables are related to changes in flood hazard to a lesser degree than physiography and flow properties.

Overall, the results of our analysis highlight how the impacts of a flood-inducing storm event on channel properties and flood hazards are highly correlated with flow characteristics and a region's geophysical signature.

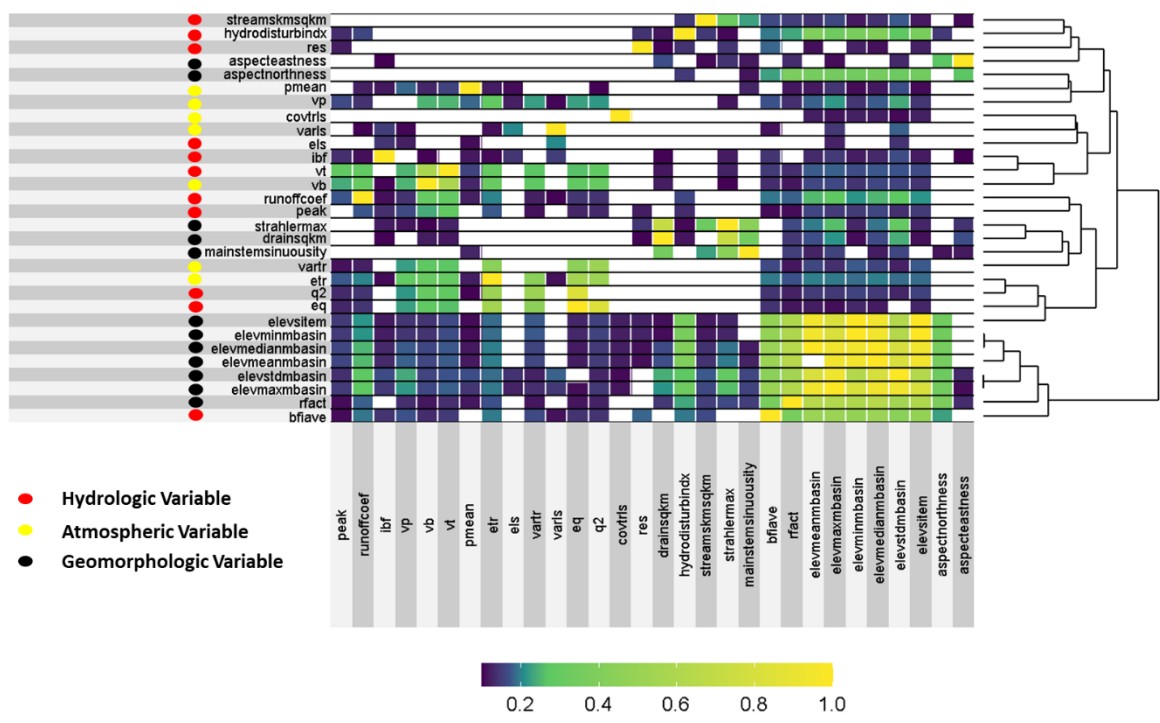

445

**Figure 9: Example of intercorrelation among the flood drivers for N = 365 days for the likelihood of change between 30 and 50%. The white color signifies that there is no correlation between those variables. The color bar from blue to yellow shows high to low correlations.**

**Table 3: Highly correlated variable groups for different percentages (%) of the "likelihood of change" from the interpretation of the dendrogram in Figure 11. Levels in the table represent the main branches of the dendrograms and groups represent the sub-branches under the main levels.**

450

| | 0-10% | 10-30% | 30-50% |
|---|---|---|---|
| Variable groups | **Level1:** | **Level1:** | **Level1:** |
| | Group1: | Group1: | Group1: |
| | ELEV_MEAN_M_BASIN, | ELEV_MEAN_M_BASIN, | ELEV_MEAN_M_BASIN, |
| | ELEV_MAX_M_BASIN, | ELEV_MAX_M_BASIN, | ELEV_MAX_M_BASIN, |
| | ELEV_MIN_M_BASIN, | ELEV_MIN_M_BASIN, | ELEV_MIN_M_BASIN, |
| | ELEV_MEDIAN_M_BASIN, | ELEV_MEDIAN_M_BASIN, | ELEV_MEDIAN_M_BASIN, |
| | ELEV_STD_M_BASIN, | ELEV_STD_M_BASIN, | ELEV_STD_M_BASIN, |

| | | |
|---|---|---|
| ELEV_SITE_M, RFACT Group 2: All the other variables | ELEV_SITE_M, EQ, Q2 Group 2: All the other variables | ELEV_SITE_M, EQ, Q2, ETR Group 2: All the other variables |
| **Level 2:** Group1: HYDRO_DISTURB_INDX, STREAMS_KM_SQ_KM, Res, ASPECT_NORTHNESS, ASPECT_EASTNESS, Vp, Pmean,CovtrLs, Vb, Vt, Els, IBF, VarLs Group 2: EQ, ETR, Q2, VarTr, RunoffCoef, Peak, STRAHLER_MAX, MAINSTEM_SINUOUSITY, DRAIN_SQKM | **Level 2:** Group1: RFACT, HYDRO_DISTURB_INDX, STREAMS_KM_SQ_KM, BFI_AVE, Res, ASPECT_NORTHNESS ASPECT_EASTNESS Pmean, Els, IBF, VarLs Group 2: Vp, CovtrLs, IBF, Vb, Vt, ETR, VarTr, RunoffCoef, Peak, STRAHLER_MAX, MAINSTEM_SINUOUSITY, DRAIN_SQKM | **Level 2:** Group1: RFACT, HYDRO_DISTURB_INDX, STREAMS_KM_SQ_KM, BFI_AVE, Res, ASPECT_NORTHNESS ASPECT_EASTNESS STRAHLER_MAX, MAINSTEM_SINUOUSITY, DRAIN_SQKM, IBF, Peak, CovtrLs Group 2: Pmean, Els, VarLs, Vp, Vb, Vt, VarTr, RunoffCoef |

## 4. Discussions

### 4.1. Channel Changes and Watershed Characteristics

Our model highlighted in Figure 6, that the most important hydrologic variable was the condition of the watershed,
whether it is anthropogenically modified or natural. This confirms that human modifications are an important element to be
considered when analyzing flood hazard changes (Bormann et al., 2011; Pinter et al., 2006a, b). Ahrendt et al. (2022)
demonstrated that channel regulation is important to conveyance changes which resonates with the variable importance
analysis results from Figure 6. Similarly, the construction of dikes, bridges, dams, meander cutoffs, channel constriction by
wing dikes, groins, and other engineering projects can alter channel conveyance within rivers and the characteristics of their
floodplains (Bormann et al., 2011; Pinter et al., 2006b, a). The importance of this variable in the model highlighted the potential
interaction of flood-inducing events that generate high sediment deposition with the effects of channel modification. As well

numerous works in literature (Feng et al., 2021; Mazzoleni et al., 2022) also highlighted how urbanization processes and landscape changes induced by human activities have large impacts on flood hazards worldwide.

465     The model gave high importance to drainage density, which is an essential characteristic of the Earth's surface that regulates erosion and the movement of water and sediments (Clubb et al., 2016). Drainage density is also correlated with subsurface permeability (Luo et al., 2016). The control these factors exert on sediment production and delivery and soil permeability may explain the importance of these variables to post-storm changes in river conveyance. Drainage density is also correlated to other hydrologic and climatic variables such as precipitation and climate types (Moglen et al., 1998).

470     Based on the visual interpretation of the unsupervised SOMs (Figure 8), taking the atmospheric, hydrologic, and current geomorphologic conditions as single independent drivers is not sufficient to predict the magnitude of the shift in stage-discharge at the flood stage. This suggests the co-occurring fluctuations in the various parameters, rather than variation in a single peak parameter, are the primary drivers of change in flood hazard at the continental scale. The patterns visible in the SOM depend on existing relationships among processes. For example, along with the drainage area, the duration and spatial pattern of rainfall are responsible for the variability in lag time and basin response (Granato, 2012; Woods and Sivapalan, 475 1999). The correlation among Drainage area (DA), peak discharge (Peak), and Mean water travel distance to the drainage outlet (Els) is evident for various clusters, as is the correlation between Normalized flow volume (Vt) and Baseflow (Vb).

    This is not surprising, considering that the basin size is generally the most important basin characteristic in determining the amount and timing of surface runoff at the outlet (Gupta and Dawdy, 1995). The relationship between flood flow quantiles and drainage area is expressed by power-law equations (Villarini and Smith, 2010). It also confirms how 480 catchments with larger drainage areas display higher values of specific discharge and how morphodynamic properties (including frequent flows such as the bankfull discharge) tend to cluster with drainage network characteristics and scaling properties (Saghafian, 2005; Reis, 2006; Sofia and Nikolopoulos, 2020b). Further cross-cluster variability occurs with some atmospheric and hydrologic variables, namely the Centroid of precipitation (ETr), Centroid of flow hydrograph (EQ), and Spreadness of precipitation (VarTr). All the previously mentioned variables present their co-occurring peaks in Cluster 6 (the 485 Upper Mississippi and Missouri region), which is in line with the fact that for this area (and cluster), snowmelt, rain on snow, or rainfall can cause major flooding.

    The physiography of the basin deeply controls the complex land-atmospheric interactions and storm types resulting in rainfall runoff. Thus, this is no surprise that physiography alone is highly correlated (Figure 9, Table 3) to all other (hydrologic, geomorphologic, and atmospheric) variables used in this study. This highlights the importance of basin attributes 490 in prompting stage-discharge variability at gage locations. Investigations of the influence of the flow stage on channel conveyance often focus on the impacts of peak or minimum bankfull discharge. From Figure 9 and Table 3, we can see that recession rates matter in sediment delivery, as highlighted in the literature (e.g., Hassan et al., 2006), and these two properties are highly correlated with the impact of large storms on flood hazards. The findings of this study provide needed insight, and managers could use the results to determine the flow hydrograph shapes that potentially alter short-term flood hazards. Such 495 knowledge is necessary for the design of river infrastructure.

Many papers in the literature (e.g., Borga et al., 2008; Woods and Sivapalan, 1999; Woods, 1999; Smith et al., 2004, 2005, 2002; Zhang et al., 2001) highlighted the relationship between the centroid of precipitation and runoff production. Most works showed that, for example, the position of the storm centroid relative to the watershed outlet is an important driver of runoff: storms having a precipitation centroid positioned in the central portion of the watershed tend to produce a higher runoff than storms having a centroid near the outlet or the head of the watershed. This is in line with the fact that rainfall runoff spatial variability influences flash flood severity relative to basin physiography and climatology. Flash flood severity, or flashiness, as defined by Saharia et al., (2017), assesses a basin's capacity to produce severe floods by considering both the volume and timing of a flood. It is, therefore, not unexpected that the centroid of precipitation appears to be highly correlated with the shifts in residuals.

Also, as shown in Figure 9 the significance of "Aspect" attributes can be understood in terms of the various runoff and soil loss yields that can result from changes in slope properties. For example, soils on south-facing slopes always seem to be much more eroded or degraded than those on more humid north-facing slopes due to differences in aspect, steepness, lithology, and flora type. ASPECT_NORTHNESS and ASPECT_EASTNESS influence the daily cycle of solar radiation affecting the temperature, humidity, and soil moisture (Desta et al., 2004) that control the vegetation and, hence, the sediment movement of the floodplain. The variability of these factors can, therefore, affect sediment production and movement, with consequences for flood hazard changes.

In Figure 9 and Table 3, our model suggests drainage properties related to the routing of the precipitation and flood water are highly correlated with residual changes and indirectly linked to post-storm modifications of flood hazards. Greater network sinuosity lowers peak flows and flooding (Seo and Schmidt, 2012; Seo et al., 2015; Saco and Kumar, 2002). Higher peak flow, faster time to peak, and shorter duration are produced by lower variability of flow path lengths (Saco & Kumar, 2002). Also, flood frequency/event increases with the decrease of the fractal dimension of the river network (Zhang et al., 2015). Lastly, the Base Flow Index and Peak discharge are intricately connected to runoff and, consequently, alterations in channel conveyance. This connection is evident as they characterize the volume of water within the channel. When the volume surpasses the channel's conveyance capacity, flooding is anticipated, and substantial sediment movement implies potential channel adjustments. The significance of these properties is a reaffirmation of the established notion that regular flows, such as baseflow below bankfull levels, are sufficient to determine channel shape, as they prevent the substantial accumulation of fine sediments and organic matter (Phillips, 2002). On the other hand, rare extreme floods are essential for transporting coarser bed material and eroding channel banks (Phillips, 2002).

## 4.2. Changes in Flood Risk after Major Floods

Figure 10a shows the groups of gages representing different percentages of "likelihood of change." If the reported value is <10%, for example, the predicted residuals for those gages show a sudden change from negative to positive in less than 10% of storms. The higher the percentages are, the more likely we expect a drastic abrupt reduction of channel capacity after a large storm. Comparing with the literature (Slater et al., 2015), we can see that, in our study, the locations with the

highest likelihood of change coincided with those with significant channel capacity and net changes in flood hazard frequency.

While the post-storm change was not as widespread as the effects highlighted by Slater et al. (2015), this was expected, as we were analyzing post-storm effects and not considering the persistence in time of these changes at this stage. Also, a higher rate of change (high percentage) might be representative of very dynamic rivers, whose changes are likely to smooth out in time. On the other hand, rivers changing less frequently might be witnessing changes with a magnitude sufficient to last longer. This fact should be addressed carefully. Another thing to consider is that, because USGS gages are purposely placed at stable

locations, our analysis, as well as other works (e.g., Li et al., 2020; Slater et al., 2015), probably underestimates the consequences of conveyance changes.

Nonetheless, our results highlighted how substantial changes had occurred even for these locations. When we focused on the amount of change relative to the current confidence bound of the stage-discharge (Figure. 10b), we could see that the magnitude of change was higher for gages that changed less frequently. The northwestern part of CONUS, where Slater et al.

(2015) highlighted clustering of increase in hazard due to consistent channel capacity changes with clusters of gages for which we predicted negative residuals outside the confidence bound of the stage-discharge relationship. For the Northeast, on the other hand, our model predicted high-magnitude changes for areas identified by Slater et al. (2015) as areas significantly impacted by flow frequency effects. It is known that existing stage-discharge relationships present uncertainty in estimating the discharge because of the variation in the individual measurements from which the estimation is derived. Our model

highlighted that the post-storm increased change lay outside the range of acceptable uncertainty at many gages. As Figure. 10b shows, this change was as widespread as the effects highlighted by Slater et al. (2015) for total positive changes in flow hazard frequency (FHF). For gages, the total FHF increased logarithmically in Slater et al., 2015, our model predicted changes further in the negative domain, outside the lower confidence bound.

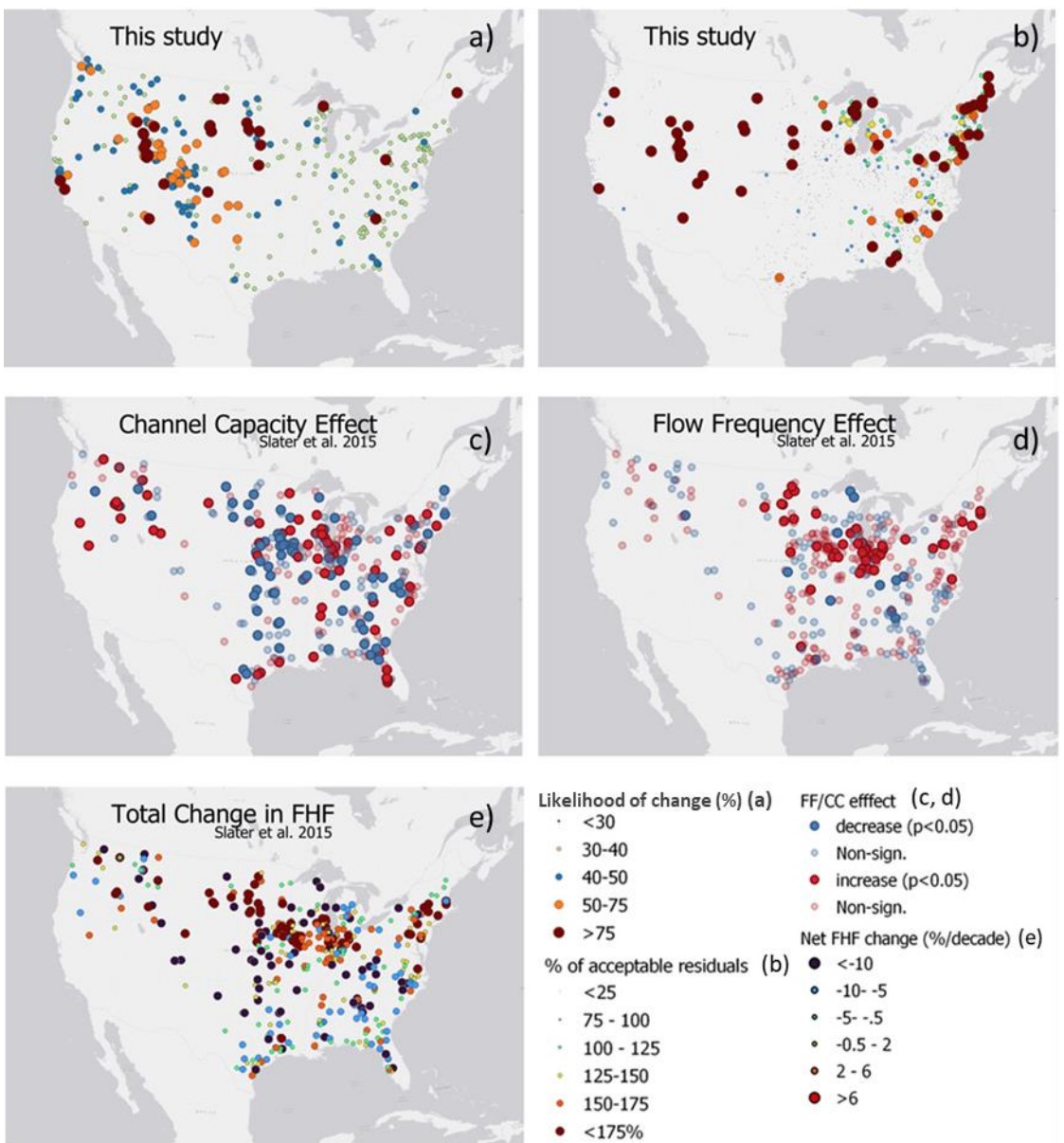

**Figure 10: Predicted changes as compared to the results of Slater et al. (2015) showing Channel Capacity (CC) and Flow Frequency (FF) effects on flood hazard frequency (FHF). In (a) "Likelihood of change"- the percentage represents the number of times the model predicts a residual change from positive to negative after a major flood (for N = 365); in (b) the panel shows the ratio between average prediction and lower 95% confidence bound of the current stage-discharge relationship for the stations showing a drastic change positive to negative. In (a, b) gages with small variations from this study have been reduced for clarity. Panel (c,d, and e) are results from Slater et al. 2015.**

From the predicted results of the channel changes at the gage level, we next analyzed which locations were more prone to changes based on the number of gages with predicted changes within each physiographic region and climate type (Figure 11). Overall, one must keep in mind the limits and the variability of the gage coverage across CONUS, as described

in the chapter related to the model limitation. Nonetheless, observing how variability changes across regions allows us to grasp how varying the post-storm effects are. Overall, rivers across the US are highly dynamic per se, and their variability depends on a combination of factors, mostly driven by how sediment moves across the landscape (Montgomery and Buffington, 1998; Flores et al., 2006). This, in turn, depends on a variety of landscape properties, as well as climate conditions, and human modifications as well (Wu et al., 2023).

Among the physiographic regions (Figure 11a), the Laurentian uplands and intermontane plateaus had the most changes (75% of all gages in this region). Rocky Mountain and Pacific Mountain systems followed the trend with the second most changes (50–75%). The changes in the <10% of the gages resided in the Interior Highlands, Atlantic Plains, and Appalachian highlands. The Appalachian Highlands regions are mountainous. In contrast, the interior plains are mostly flat agricultural lands whose river system consists of the upper Mississippi River, the Ohio River, parts of the Great Lakes, and

small wetlands. This region has very dynamic hydrology, with very cold winters and hot summers. Snowmelt in the spring and heavy precipitation in the summer and winter result in big floods. Naturally, this can potentially lead to changes in the river reaches. While the Atlantic Plain is also relatively flat, it covers the Mississippi Delta, the Gulf of Mexico, and the Atlantic seaboard in the East (see Figure 2). Moving toward the coastline, frequent tropical storms and cyclones are recorded, which could increase sediment activity overall (Tweel and Turner, 2014). As well, lots of human activities can alter river

morphology, especially in the deltas, due to sediment movements (Nienhuis et al., 2020). The literature (Bracken and Croke, 2007; Kalantari et al., 2019; Croke et al., 2013; Sofia and Nikolopoulos, 2020a; Wohl et al., 2019) has highlighted sediment connectivity as a potentially critical factor in flood hazards, being linked to both changes in channel characteristics and increasing decadal trends in flood hazard, independent of scale. In addition, for these regions, and in the eastern United States more generally, peak flows are highly variable (Villarini & Smith, 2010), and tropical cyclones affect the distribution of

sediments as well (Tweel and Turner, 2014). All these characteristics contribute to the presence of very dynamic rivers, which, as confirmed by our model, quickly react to flood-inducing events, adjusting their geometry and altering flood hazards in the case of subsequent floods.

We made the same comparison for the climate types (Figure 11b). We detected high predicted variability mainly in hot and humid climate regions, while cold and dry regions showed minimal changes. Humid Continental climate (Dsb, Dfa,

Dfb) led with the highest variability (>75% of the gages resided in these climate regions). The gages with 50–75% channel changes were in the Tundra Climate (ET) and Warm Summer Mediterranean Climate (Csb). Gages with the least changes (<10%) were located in Humid Continental Hot Summers with Dry Winters (Dwa), Continental Subarctic-Cold Dry Summer (Dsc), Cold Desert Climate (Wk), and Hot Semi-Arid Climate (BSh). These climate zones are mostly dry either year-round or seasonally. The impact of major storms on rivers depends on both underlying long-term climate signatures (Chen et al., 2019;

Stark et al., 2010) and short-term (year-to-year) climate variability (Slater et al., 2019). For many river systems, coarse sediment mobilization and transportation rates are controlled by regional climate (Anderson and Konrad, 2019). Climate variability is projected to trigger a chain reaction of geomorphic responses, including changes in downstream channel properties (East and Sankey, 2020; Wendland, 1996; Harrison et al., 2019; Knight and Harrison, 2012). Other studies focusing

on long-term changes rather than flood-inducing events have shown how decadal-scale changes in river morphology may be accounted for as a downstream propagating channel reaction to regional climate variability, which is frequently accompanied by cyclical changes in channel geometry and conveyance (Scorpio et al., 2015; Slater et al., 2019). The joint contribution of physiographic regions (as a proxy for sediment characteristics) and climate properties has also highlighted the nonlinearity of system response and the potentially harmful and sequential effects that result from the coupled direct impacts of climate conditions and sediment connectivity (Lane et al., 2007).

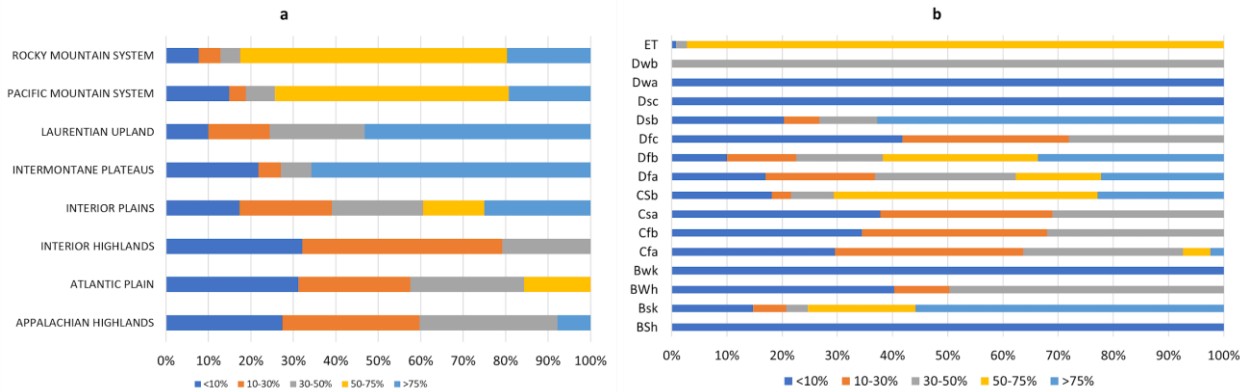

**Figure 11: Percentages of gages presenting changes in channel capacity in different (a) physiographic regions and (b) climate types.**

## 5. Advantages and Limitations of the Framework

This work is based on gage measurements, and across CONUS there is a known bias of stream size representation and spatial density in the gaging network, whereas some river sizes and landscape areas are vastly under- and over-represented (Kiang et al., 2013). Regarding the coverage of stream gages, the intrinsic limits of the dataset, in general, have been addressed in the literature and are very well summarized in the publication by Kiang et al., (2013). Broadly speaking, the Eastern United States has better coverage compared to its Western counterpart. Particularly, the arid Southwestern United States shows notably lacking spatial coverage. Discrepancies in hydrology contribute to variations in the statistical uncertainty calculated across different parts of the country (Kiang et al., 2013). The Central and Southwestern United States, characterized by arid and semiarid conditions, generally display higher interannual variability in flow, resulting in increased uncertainty in flow statistics. Despite these distinctions, it's essential to recognize that any research relying on gaging sites faces similar limits and is overall affected by potential over or underrepresentation of flows. We believe that as USGS stream gage information could potentially be transferred from nearby stream gages if there is sufficient similarity between the gaged watersheds and the ungaged watersheds of interest, our model could also be applied to ungaged sites. However, one must always keep in mind that the successful 'translation' to ungaged environments depends on the correlation of the stream gages in the surrounding areas. For example, there are areas of CONUS (mostly mountainous) that show highly correlated stream gages (Kiang et al., 2013), whereas the Central United States and coastal areas of the Southeastern United States show much uncorrelated gages.

Therefore, the goodness of the information transfer might not work as well. Also, transferability would be most likely to be successful when basin attributes show high similarity and storm properties are within the range of variability of the training

set used for this work. We do not recommend the use of this model for engineered rivers, where channel changes are expected to be limited by infrastructures such as concrete levees, as the model was trained excluding specifically sites featuring artificial controls at the gauging station that could impede the natural adjustment of the channel's shape.

The ML model was trained considering both storm properties and watershed properties. The system is not capable of highlighting which element triggers the change, nonetheless, we provided an assessment of feature importance to stress that

the shifts in how the model works, are mostly explained by a combination of storm and watershed properties. We would not suggest using the model, as it is trained currently, to predict changes without having information on the storm properties. Regarding storm properties, this study uses a published dataset (Shen et al. 2017) of storm events ranging from 2002 to 2013. The framework displays the intercorrelation of the different event properties that can affect channel changes, and this framework could be used for identifying variable gages outside the time range covered by the storm event database.

Nonetheless, researchers can use the trained model with additional years of data, if they have available the same storm properties proposed by Shen et al. for more recent events.

A further thing to consider refers to the watershed properties considered in the model. The Gage Dataset includes several hundred watershed characteristics compiled from national data sources, Actual stream density, as other properties, for example, could be different from those derived from national data sources, due to time and landscape changes happening in

the watersheds, The advantage of the considered dataset, however, is that it is available consistently for all gages. Researchers could also consider using different methods to define the watershed properties and consider improved geomorphological parameters from high-resolution terrain data, derived from LIDAR sources for example (Passalacqua et al., 2015). In this case, it would be recommended to re-train the model and verify once again the importance of this parameter in the re-trained model, as the literature strongly highlights the higher variability of geomorphological and hydrological parameters derived from

varying resolution terrain (Sofia, 2020b).

One must note that the permutation feature importance changes with the shuffling of the feature; this process introduces randomness to the process (Molnar, 2022), which might not be representative of a physical process. When repeating the permutation, the results may vary considerably (Molnar, 2022). To increase robustness and stabilize the measure, we repeated the permutation and averaged the importance measures over the various reiterations. A further aspect to consider is

that if the features are correlated, the permutation feature importance may be biased, with unrealistic data examples. The randomness added by the permutation might result in an unlikely combination of the parameters. This issue is more evident if real-world variables are directly or inversely correlated; by shuffling one of the features, we may be creating new unlikely or physically impossible instances. Therefore, as Molnar (2022) suggested, we may be potentially looking into a decrease in the model performance only due to values that we would never observe in the real world.

We should point out that channel conveyance change is known to vary spatially across a region and strongly correlates with climate variations and landscape properties. The feature permutation randomness for our study case was, however,

counteracted by the two main features of SOMs: (1) the topological preservation of the neighborhood, which results in spatial clusters of comparable patterns in the output space; and (2) the adaptation property in which the winner neuron and its neighbors are changed to make the weight vectors more similar to the input. The SOM method can recognize new patterns

during the training process. Besides that, using multiple attributes, such as combined atmospheric, hydrologic, and geomorphologic variables, can improve the pattern generated by the SOM. In our approach, the variable importance did not change, considering the various N intervals used to group storm properties. The high correlation between estimated residuals and measured ones during the 10-fold validation confirmed the accuracy of the model.

Careful interpretations that explain how and why channel conveyance changes happen as they do are essential to
guiding reliable predictions of river conveyance behavior and evolution. Another aspect to consider, as for any ML approach, is that SOMs are stochastic, as there are no physical constraints in their prediction. The use of randomness as a feature in the SOM analysis exerts confidence in the results mainly when the results are agreeable with the theoretical aspect of the variables. We suggest referring to (Brierley et al., 2021) for a recent review of ML limitations in geomorphology in general.

## 6. Conclusions

The variability of geomorphologic processes and future flood patterns can only be understood by evaluating all the critical flood drivers responsible. In this era of flood-inducing events and rapidly changing landscapes, accurate flood hazard assessment is paramount. Atmospheric, hydrologic, and geomorphologic parameters constitute both the main driving force behind and the detector of changes resulting from a flood-inducing event. This study focused on the impact of flood-inducing events on flood hazards by exploring the channel changes following them. We utilized the interdependencies of the
atmospheric, hydrologic, and geomorphologic flood drivers to gain an understanding of the impact of flood-inducing events on channel capacity and identified important drivers for predicting residuals from the average stage-discharge curve.

Our results confirm existing knowledge of watershed hydrology and further strengthen the compound importance of climate and geomorphology as drivers of changes in flood hazards. The sequential processes during and after a big flood event can only be understood by considering the contribution of all the flood drivers together. The results show how the variables of
different flood drivers are interrelated and can create effects that are more adverse together. Channel conveyance change is often regarded as stationary in flood hazard modeling and is acknowledged as one of the most important sources of uncertainty. The bankfull discharge and flood occurrences are directly related to channel conveyance capacity. Our research reveals that the assumption of channel stationarity may result in either over or under-prediction of the river discharge for a certain flood stage, as the existing stage-discharge relationship might be temporarily (or permanently if the shift pertains) underperforming.
This would in turn eventually over/under-estimate flood hazard (recurrence interval, duration, depth, and inundation extent of flooding), especially in the case of subsequent floods. These models incorrectly feed flood control planning procedures, which raises the level of uncertainty in evacuation and rescue operations. Additionally, flood insurance plans created using these models' results are likewise incorrect. Furthermore, if engineering designs are based on data collected before periods when

major flood events have lowered channel conveyance, there is a risk that surveyed channel dimensions and flood conveyance will be overestimated in the long run.

The proposed ML model allows us to identify dynamic rivers more prone to changes in the stage-discharge relationship after major flood events. The proposed model does not account for the persistence of changes; that being said, the results highlight the risk of an abrupt reduction in channel capacity after a large storm. For rivers more prone to changes, periodic revision of flood frequency statistics is advisable for hazard assessments to keep pace with altered conditions. Understanding the temporal duration of these changes would offer valuable insights into the practicality of implementing these updates or exploring alternative approaches to assessing flood risk, especially if the process exhibits significant variability over time. The model predicts a shift in the discharge at the flood stage (residuals), as a proxy for flood hazard changes, implying that a certain discharge, expected to produce floodings, will be reached for lower stages than expected (residual shifting from positive to negative, at a specific gage). The approach starts from the concept that, typically, discharge time series are derived from water level measurements through an existing stage-discharge relationship. This is the general case for most gaging sites in the US, as well as other realities in other countries. As rating changes often happen during episodic storms, the proposed model can be adapted for other gage datasets, in different parts of the world, by assuming the operational existence of a similar approach.

The gages used in the study although distributed across CONUS have intrinsic limitations in terms of stream size representations and spatial coverage of the river network. Therefore, careful considerations should be applied while considering the model for predicting the impact of flood-inducing storms on abrupt loss of channel capacity outside the basins used in our study.  This study considered a limited set of drivers, excluding, for example, human activities in the watersheds and vegetation properties. Channel changes can be due to other geographically significant events (e.g. landslides, debris flow, etc), however, such occurrences could also be triggered by the storm events that caused the flood hazards. At this stage, we have a complete database of storm properties, but we did not include an analysis of additional event parameters such as mass movements and the volume (if known) of sediment/Debris delivered during such events. Future research could improve the method by adding predictors and investigating the sensitivity of median storm characteristics to different intervals (lag times). In response to increased flow, we do not anticipate channel conveyance to rise consistently everywhere. The intricate interaction of dynamic anthropogenic, climatic factors and their consequential processes within each basin, are expected to be evident in the fluvial changes. Hence, sediment connectivity, Land-Use, and Land-Cover Change anthropogenic factors could also be included to retrain the model to produce changes in the stage-discharge relationship at the flood stage and potentially create scope for the prediction of channel changes due to flood-inducing events.

**Competing interests:** The contact author has declared that none of the authors has any competing interests.

**Acknowledgments:** This study was supported by the Eversource Energy Center at the University of Connecticut
**Data Sources:**

- Flood stage values are provided by the US National Weather Service (National Oceanic and Atmospheric Administration, 2021).

- Historical mean daily streamflow records are stored by the US Geological Survey (USGS) and made publicly available online (U.S. Geological Survey, 2021a).

- The flood event database used in the study was generated by Shen et al. (2017).

- Historical field measurements of channel properties are made publicly available online by the USGS (U.S. Geological Survey, 2021b).

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

**Appendix A**

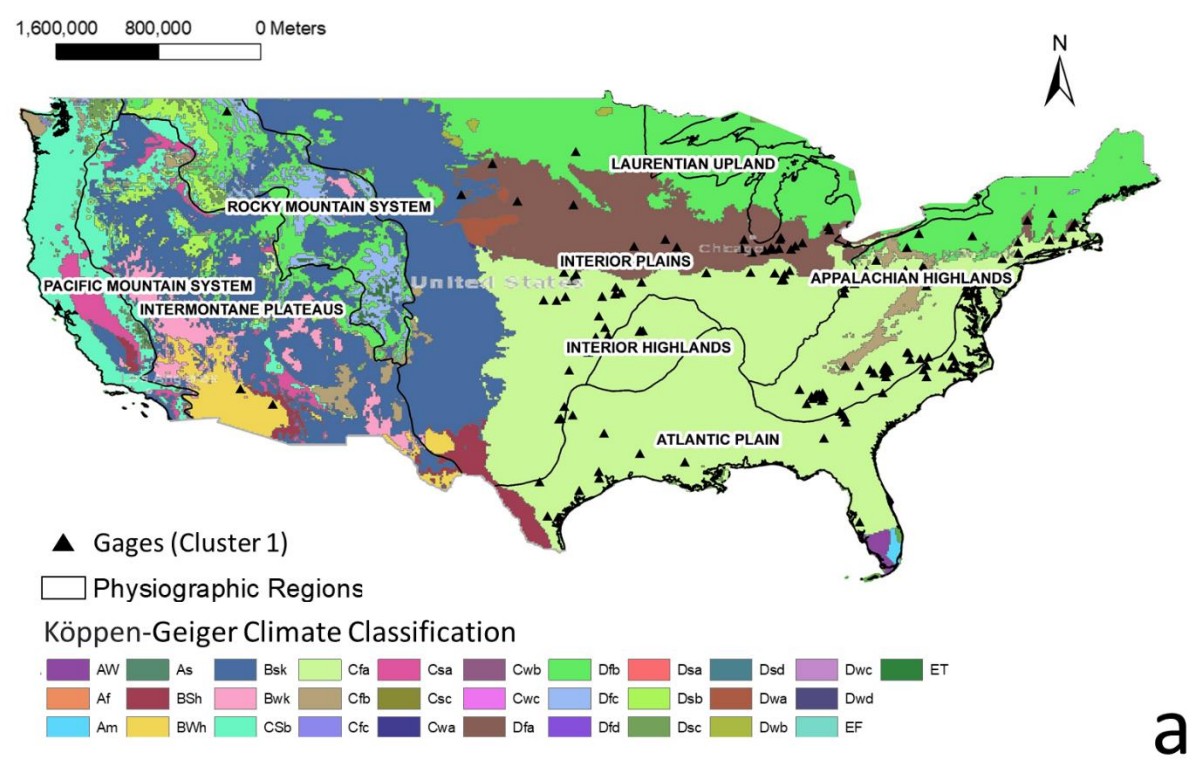

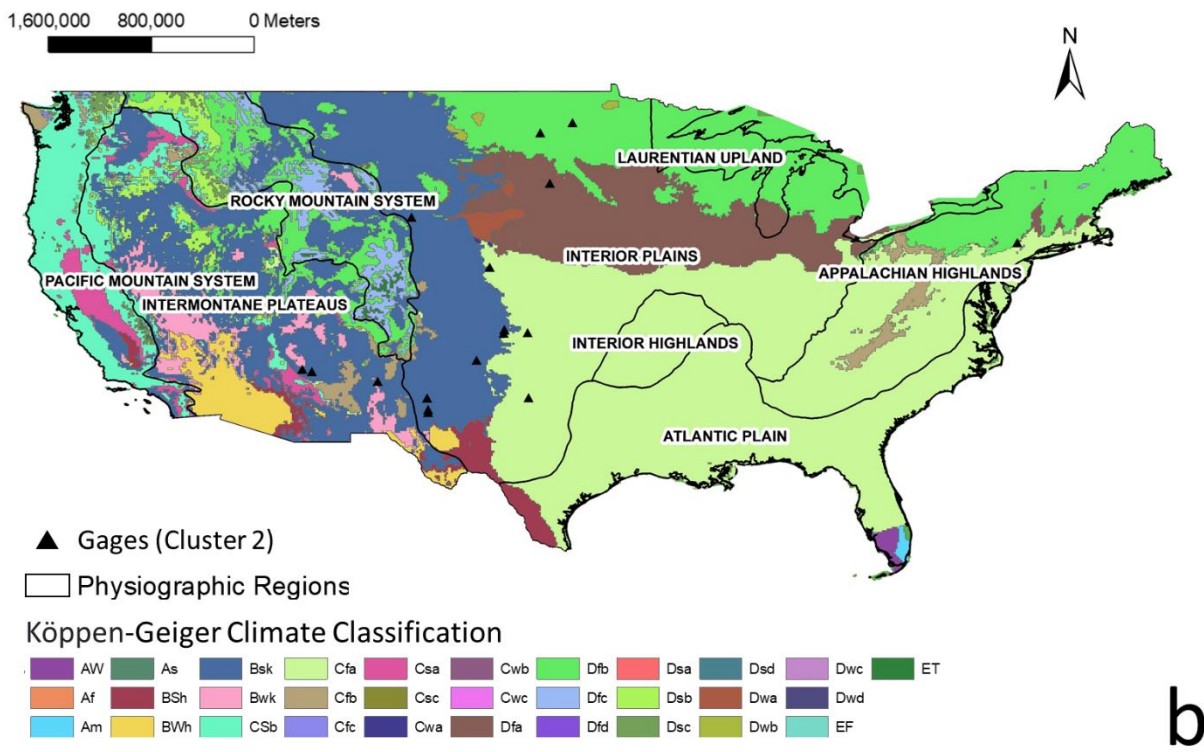

▲ Gages (Cluster 2)

☐ Physiographic Regions

## Köppen-Geiger Climate Classification

| | | | | | | | | | |
|---|---|---|---|---|---|---|---|---|---|
| ■ AW | ■ As | ■ Bsk | ■ Cfa | ■ Csa | ■ Cwb | ■ Dfb | ■ Dsa | ■ Dsd | ■ Dwc | ■ ET |
| ■ Af | ■ BSh | ■ Bwk | ■ Cfb | ■ Csc | ■ Cwc | ■ Dfc | ■ Dsb | ■ Dwa | ■ Dwd | |
| ■ Am | ■ BWh | ■ CSb | ■ Cfc | ■ Cwa | ■ Dfa | ■ Dfd | ■ Dsc | ■ Dwb | ■ EF | |

b

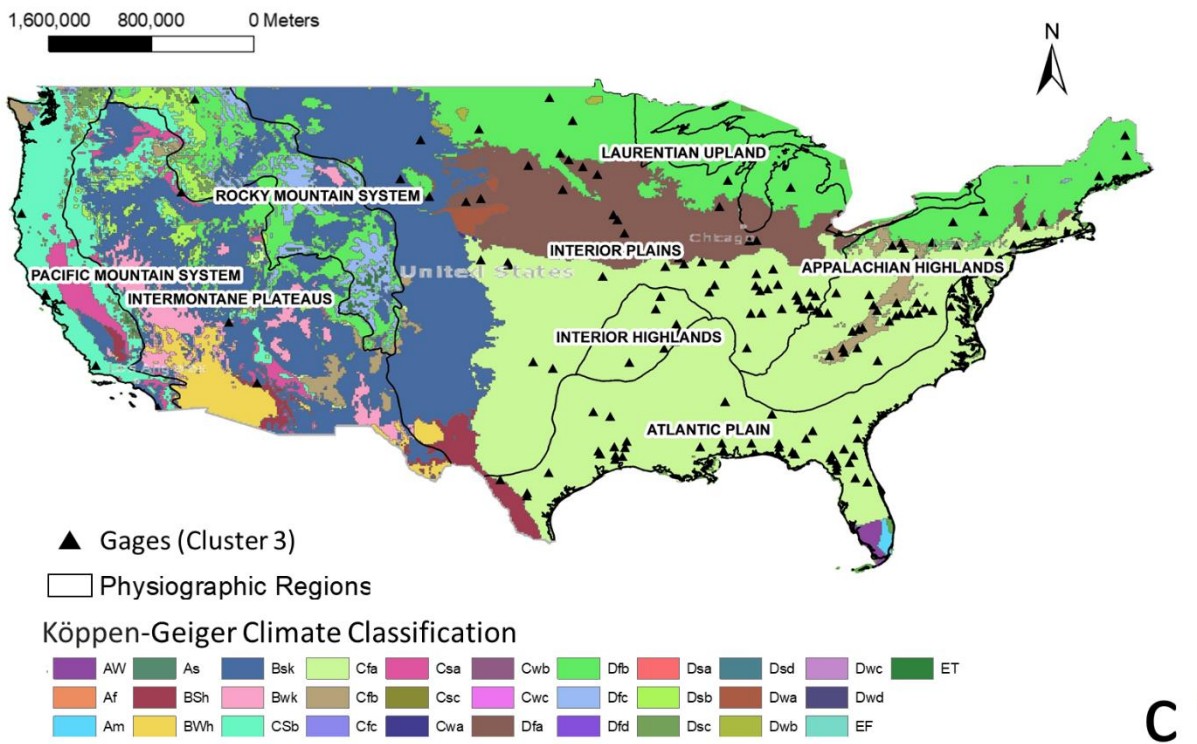

▲ Gages (Cluster 3)

☐ Physiographic Regions

## Köppen-Geiger Climate Classification

| | | | | | | | |
|---|---|---|---|---|---|---|---|
| AW | As | Bsk | Cfa | Csa | Cwb | Dfb | Dsa | Dsd | Dwc | ET |
| Af | BSh | Bwk | Cfb | Csc | Cwc | Dfc | Dsb | Dwa | Dwd | |
| Am | BWh | CSb | Cfc | Cwa | Dfa | Dfd | Dsc | Dwb | EF | |

C

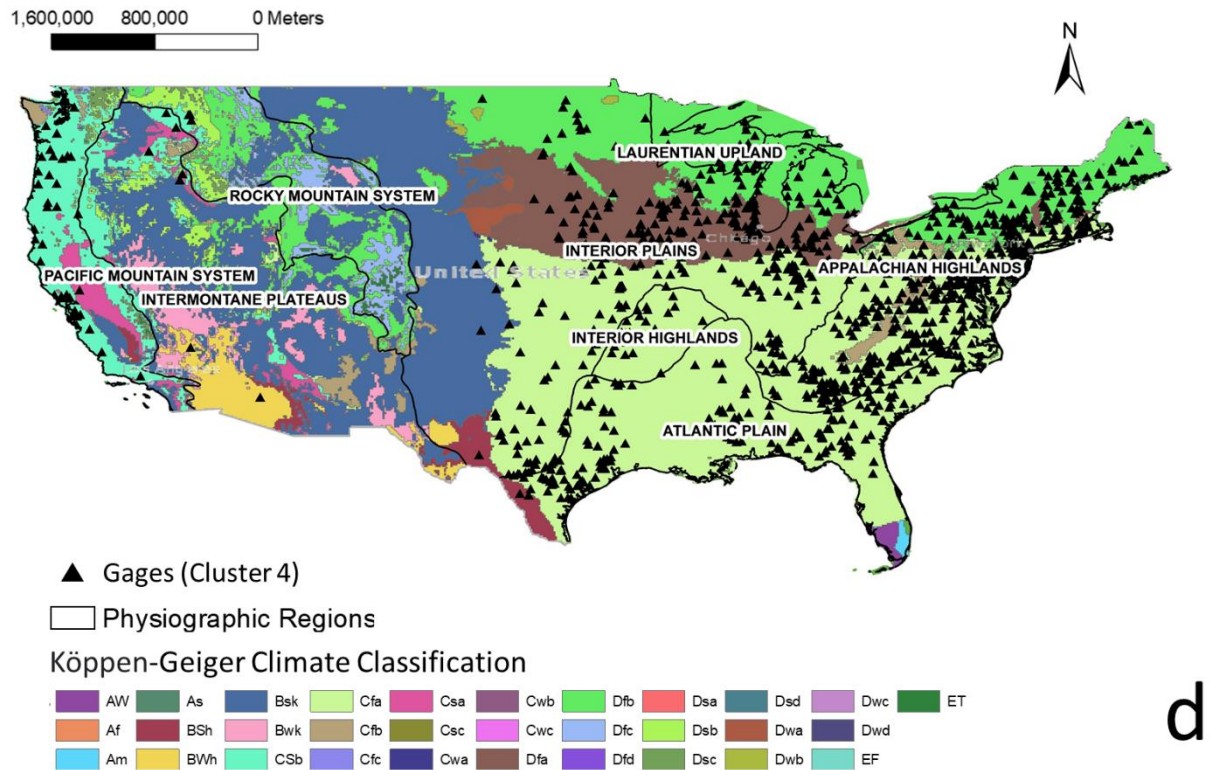

d


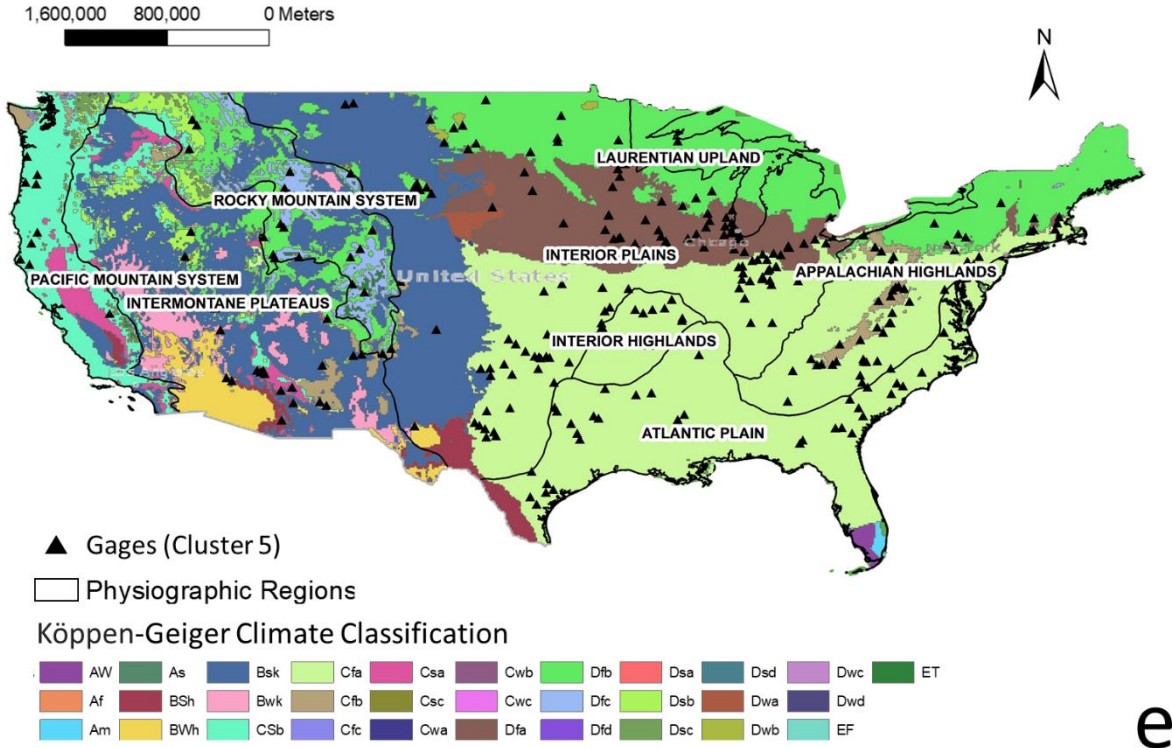

▲ Gages (Cluster 5)

☐ Physiographic Regions

Köppen-Geiger Climate Classification

| | | | | | | | | | | | | |
|---|---|---|---|---|---|---|---|---|---|---|---|---|
| AW | As | Bsk | Cfa | Csa | Cwb | Dfb | Dsa | Dsd | Dwc | ET | | |
| Af | BSh | Bwk | Cfb | Csc | Cwc | Dfc | Dsb | Dwa | Dwd | | | |
| Am | BWh | CSb | Cfc | Cwa | Dfa | Dfd | Dsc | Dwb | EF | | | |

e

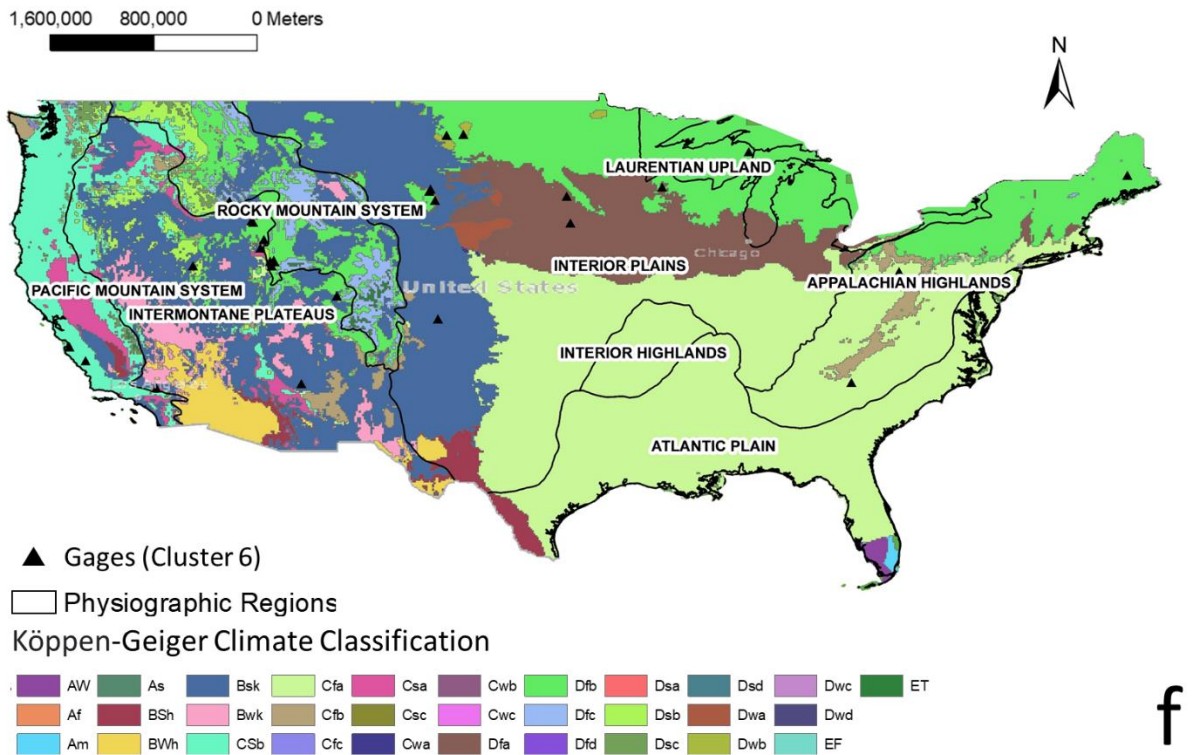

▲ Gages (Cluster 6)

☐ Physiographic Regions

Köppen-Geiger Climate Classification

| | | | | | | |
|---|---|---|---|---|---|---|
| AW | As | Bsk | Cfa | Csa | Cwb | Dfb | Dsa | Dsd | Dwc | ET |
| Af | BSh | Bwk | Cfb | Csc | Cwc | Dfc | Dsb | Dwa | Dwd | |
| Am | BWh | CSb | Cfc | Cwa | Dfa | Dfd | Dsc | Dwb | EF | |

f

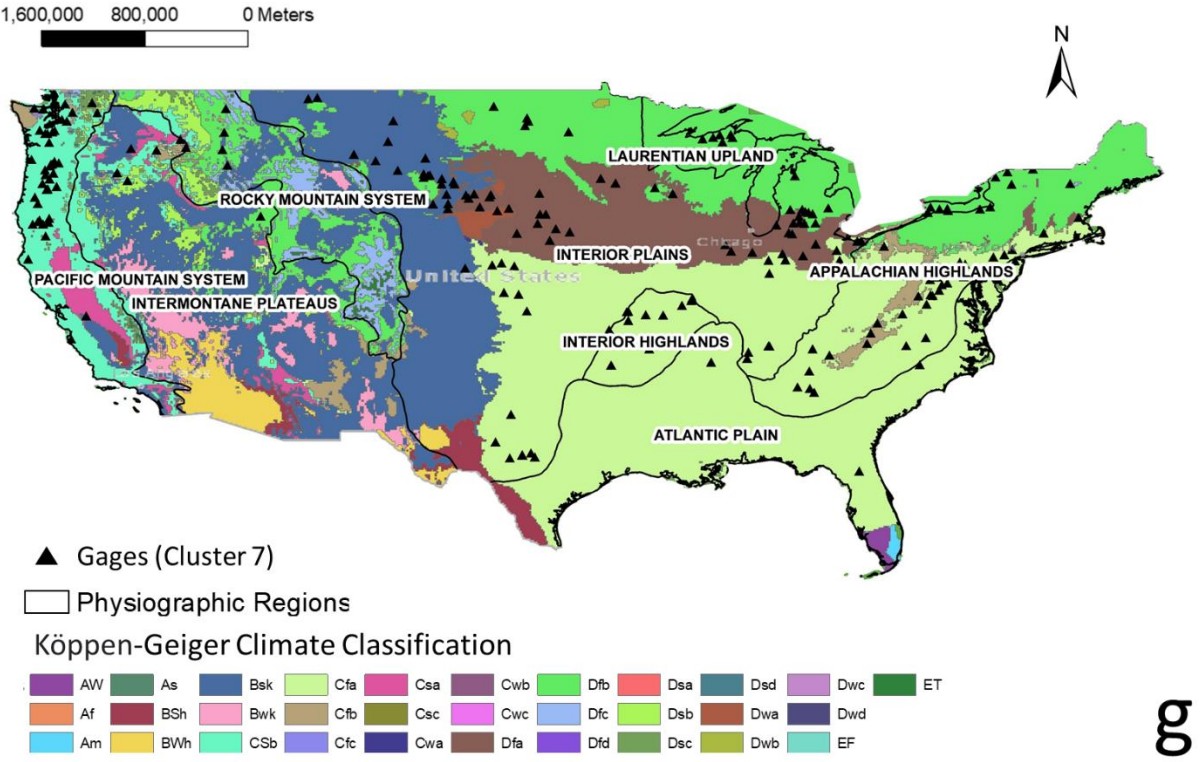

▲ Gages (Cluster 7)

☐ Physiographic Regions

Köppen-Geiger Climate Classification

| | | | | | | | | | | | | |
|---|---|---|---|---|---|---|---|---|---|---|---|---|
| ■ AW | ■ As | ■ Bsk | ■ Cfa | ■ Csa | ■ Cwb | ■ Dfb | ■ Dsa | ■ Dsd | ■ Dwc | ■ ET |
| ■ Af | ■ BSh | ■ Bwk | ■ Cfb | ■ Csc | ■ Cwc | ■ Dfc | ■ Dsb | ■ Dwa | ■ Dwd |
| ■ Am | ■ BWh | ■ CSb | ■ Cfc | ■ Cwa | ■ Dfa | ■ Dfd | ■ Dsc | ■ Dwb | ■ EF |

g

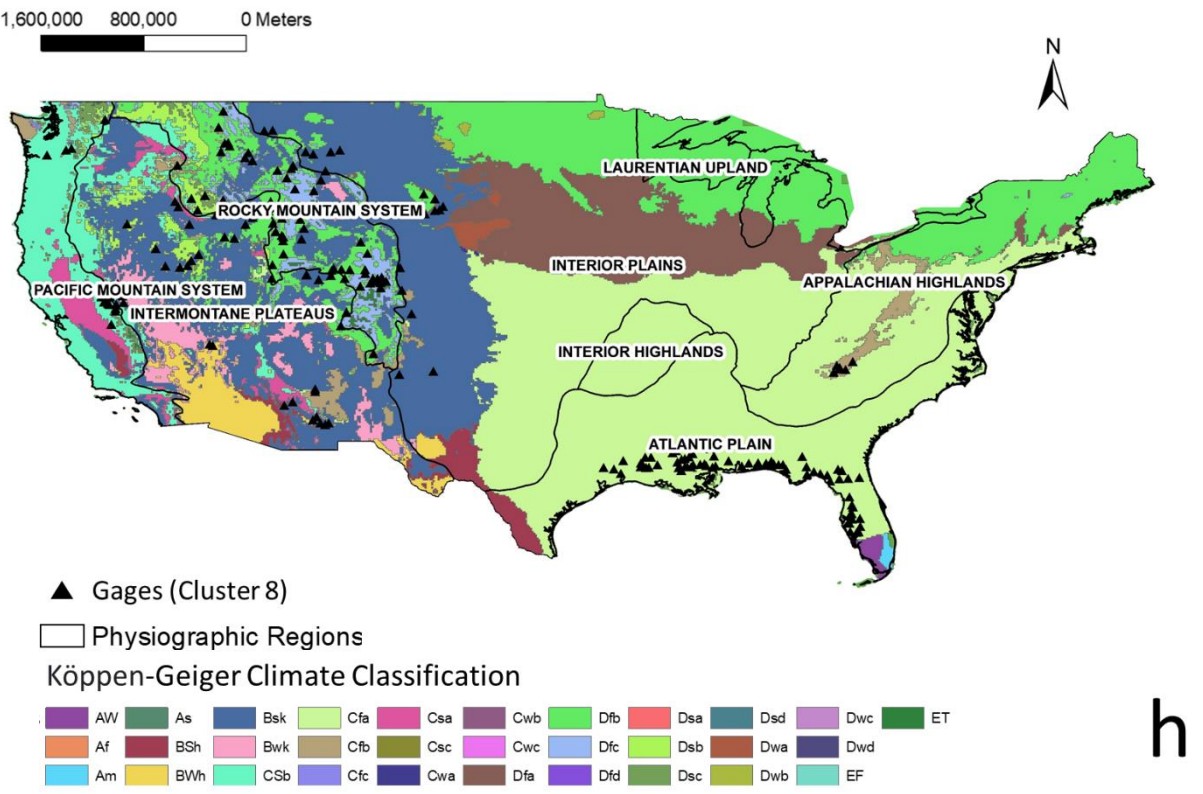

▲ Gages (Cluster 8)

☐ Physiographic Regions

Köppen-Geiger Climate Classification

| | | | | | |
|---|---|---|---|---|---|
| AW | As | Bsk | Cfa | Csa | Cwb | Dfb | Dsa | Dsd | Dwc | ET |
| Af | BSh | Bwk | Cfb | Csc | Cwc | Dfc | Dsb | Dwa | Dwd | |
| Am | BWh | CSb | Cfc | Cwa | Dfa | Dfd | Dsc | Dwb | EF | |

h

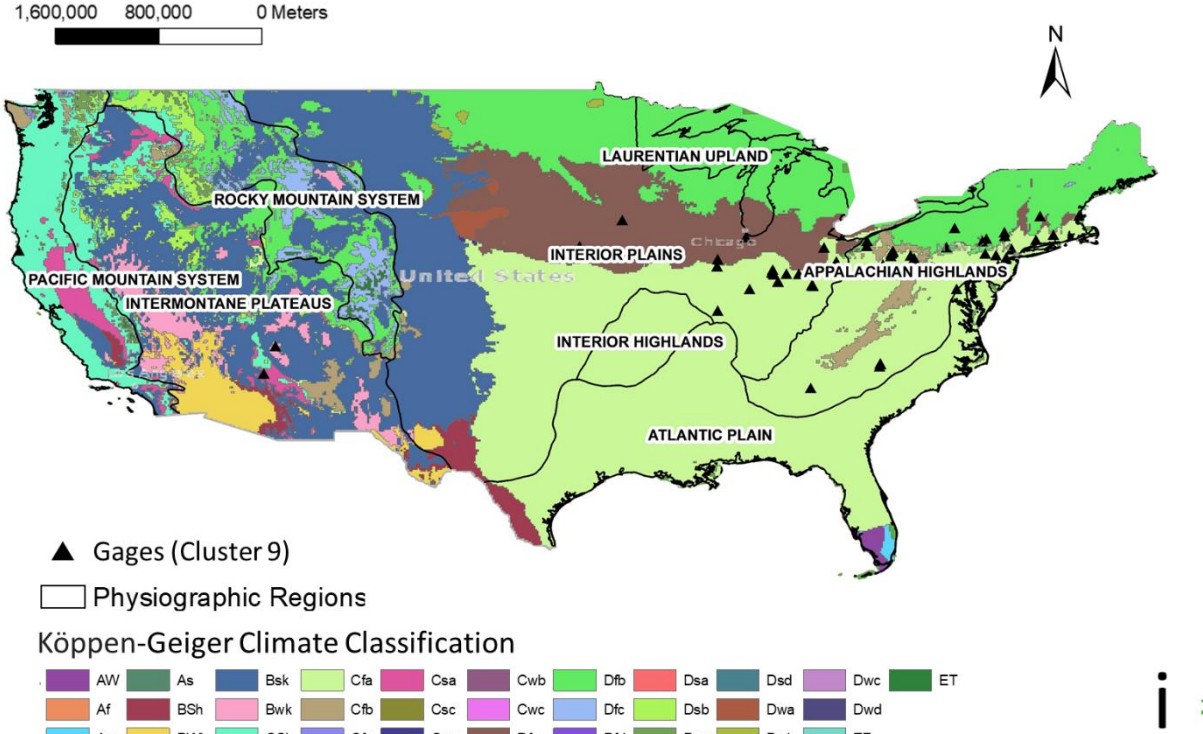

▲  Gages (Cluster 9)

☐  Physiographic Regions

Köppen-Geiger Climate Classification

| | | | | | | | | | | | | | | | | |
|---|---|---|---|---|---|---|---|---|---|---|---|---|---|---|---|---|
| ■ | AW | ■ | As | ■ | Bsk | ■ | Cfa | ■ | Csa | ■ | Cwb | ■ | Dfb | ■ | Dsa | ■ | Dsd | ■ | Dwc | ■ | ET |
| ■ | Af | ■ | BSh | ■ | Bwk | ■ | Cfb | ■ | Csc | ■ | Cwc | ■ | Dfc | ■ | Dsb | ■ | Dwa | ■ | Dwd |
| ■ | Am | ■ | BWh | ■ | CSb | ■ | Cfc | ■ | Cwa | ■ | Dfa | ■ | Dfd | ■ | Dsc | ■ | Dwb | ■ | EF |


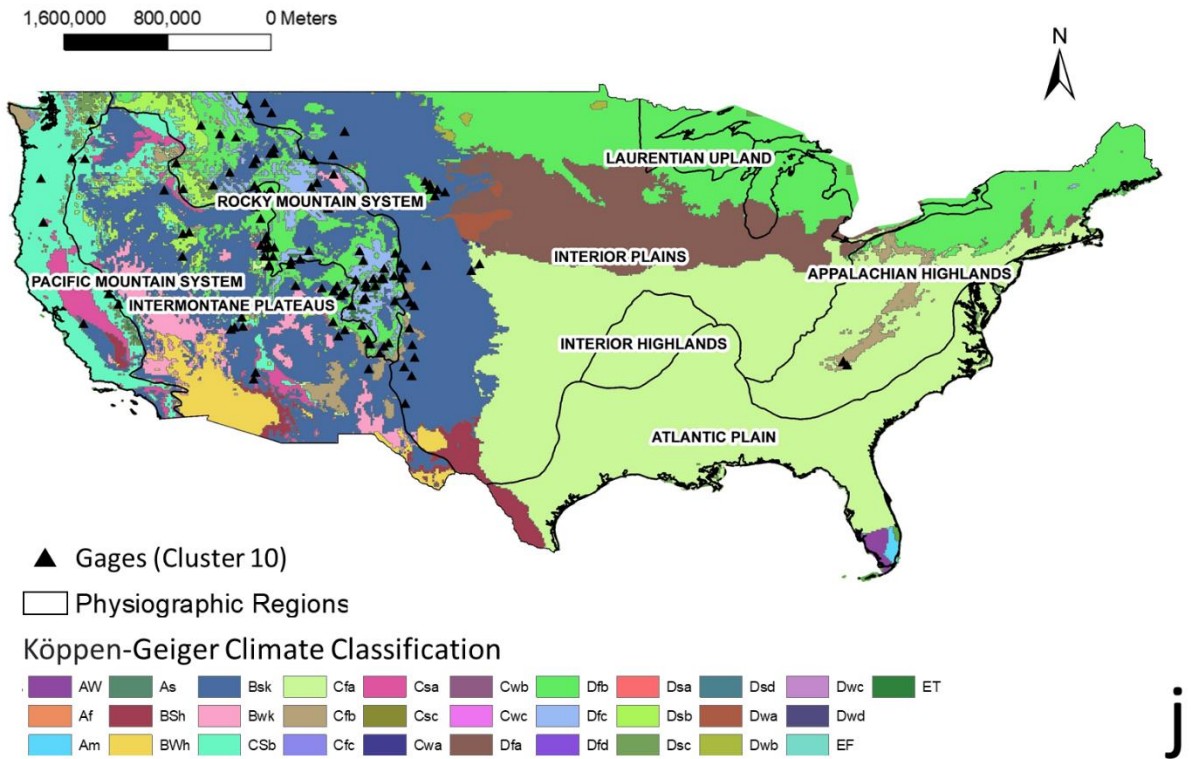

▲ Gages (Cluster 10)

☐ Physiographic Regions

Köppen-Geiger Climate Classification

| | | | | | | | | |
|---|---|---|---|---|---|---|---|
| AW | As | Bsk | Cfa | Csa | Cwb | Dfb | Dsa | Dsd | Dwc | ET |
| Af | BSh | Bwk | Cfb | Csc | Cwc | Dfc | Dsb | Dwa | Dwd | |
| Am | BWh | CSb | Cfc | Cwa | Dfa | Dfd | Dsc | Dwb | EF | |

j

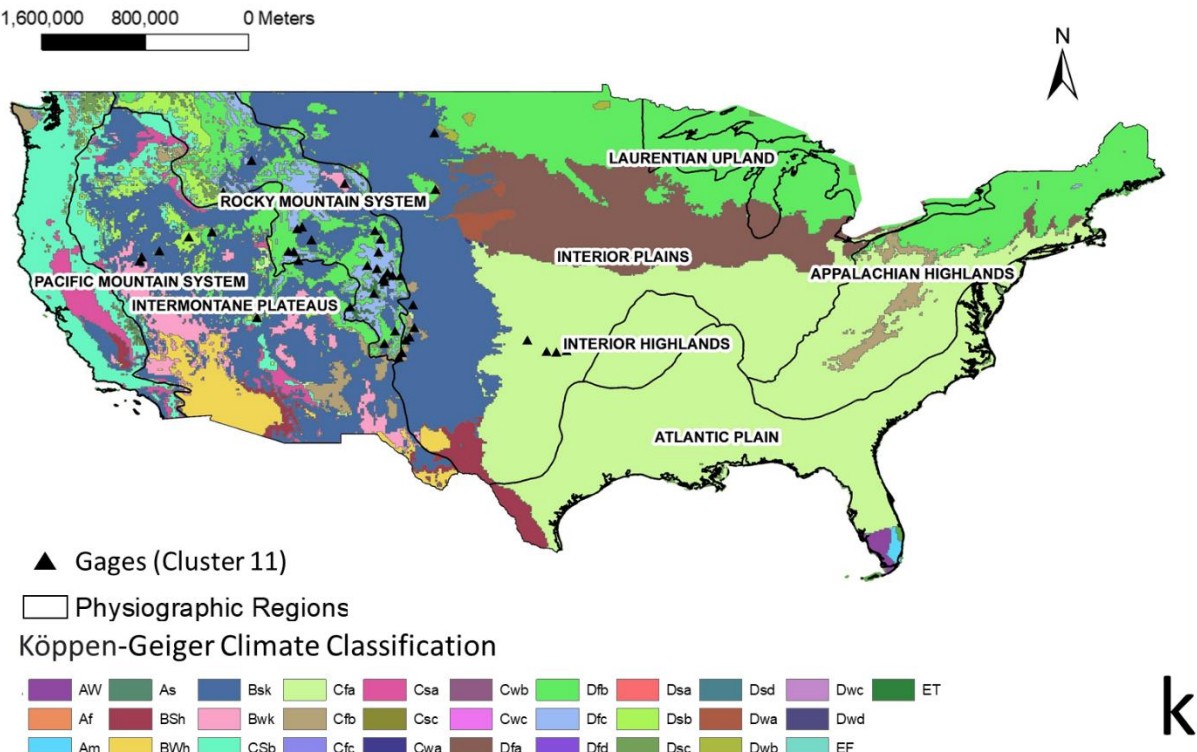

▲ Gages (Cluster 11)

☐ Physiographic Regions

Köppen-Geiger Climate Classification

| | | | | | | | | | | | | | | | | | |
|---|---|---|---|---|---|---|---|---|---|---|---|---|---|---|---|---|---|
| AW | As | Bsk | Cfa | Csa | Cwb | Dfb | Dsa | Dsd | Dwc | ET |
| Af | BSh | Bwk | Cfb | Csc | Cwc | Dfc | Dsb | Dwa | Dwd |
| Am | BWh | CSb | Cfc | Cwa | Dfa | Dfd | Dsc | Dwb | EF |

k

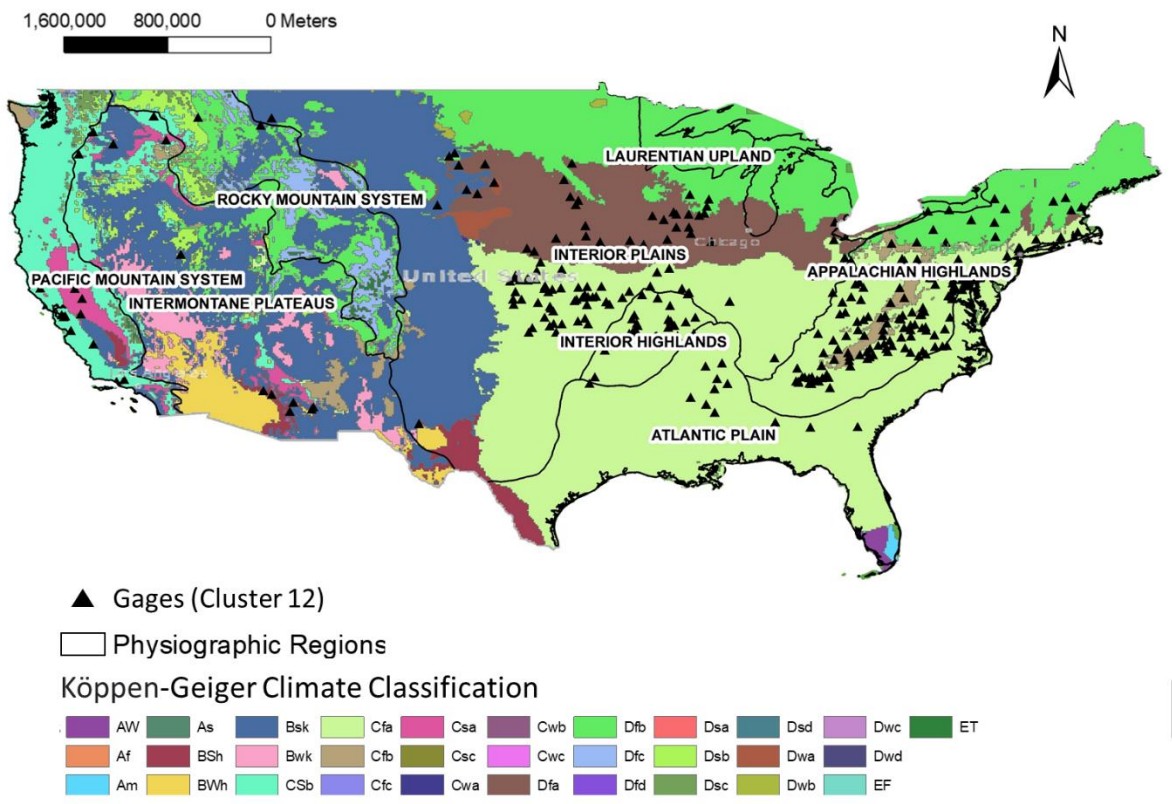

**Figure A1: Gages with clustering identification assigned by SOM unsupervised clustering (a-l). For the acronym description of Physiographic regions and climate types please refer to Table A1 and A2.**

**Table A1: Description of climate types from Köppen-Geiger climate classification (Beck et al., 2018) used in Figure 2 and A1**

| Climate types | Description |
| --- | --- |
| Af | Tropical rainforest |
| Am | Tropical monsoon |
| Aw | Tropical Savanna (Wet and Dry Climate) |
| BWh | Hot desert climate |
| BWk | Cold desert climate |
| BSh | Hot semi-arid climate |
| BSk | Cold semi-arid climate |
| Csa | Hot-summer mediterranean climate |
| Csb | Warm-summer mediterranean climate |
| Csc | Temperate dry summer  cold summer |
| Cwa | Warm oceanic climate / humid subtropical climate |

| | |
|---|---|
| Cwb | Subtropical highland climate or temperate oceanic climate with dry winters |
| Cwc | Cold subtropical highland/Subpolar Oceanic |
| Cfa | Humid subtropical climate |
| Cfb | Temperate oceanic climate |
| Cfc | Subpolar oceanic climate |
| Dsa | Humid continental climate - dry warm summer |
| Dsb | Humid continental climate - dry cool summer |
| Dsc | Continental subarctic - cold dry summer |
| Dsd | Continental subarctic – dry summer very cold winter |
| Dwa | Humid continental hot summers dry winters |
| Dwb | Humid continental mild summer dry winters |
| Dwc | Subarctic with cool summers dry winters |
| Dfa | Humid continental hot summers year around precipitation |
| Dfb | Humid continental mild summer wet all year |
| Dfc | Subarctic with cool summers year around rainfall |
| Dfd | Subarctic with cold winters year around rainfall |
| ET | Tundra climate |
| EF | Ice cap climate |

**Table A2: Description of Physiographic regions (Fenneman and Johnson, 1964) presented in Figure 2 and A1**

| Physiographic Regions | Description |
|---|---|
| ApHigh | Appalachian Highlands |
| AtlPlain | Atlantic Plain |
| IntHigh | Interiors Highlands |
| IntPlain | Interior Plains |
| IntermPlat | Intermontane Plateaus |
| LaurUpl | Laurentian Upland |
| PacMounSys | Pacific Mountain System |
| RockMounSys | Rocky Mountain System |