# Peer review of "To What Extent Do Extreme Storm Events Change Future Flood Hazards?"

_EGUsphere, 2023_

## Referee Comment (RC1)

**General Comments**

This study quantified residuals in average stage-discharge rating curve from manual field measurements at U.S. Geologic Survey stream gaging stations. The residuals about the average stage-discharge curve quantify changes in channel capacity by evaluating the change in discharge required to achieve a certain water stage. For each measurement, at each gage, the authors quantified a set of geomorphic, hydrologic, and atmospheric variables. Included in these variables were individual storm properties. Assign of storm properties for each measurement were quantified by considering a lag time and computing the median storm property for all storms within that lag time. Lag times of 15, 30, 90, 180, and 365 days prior to the stage-discharge measurement were considered. The authors then trained and validated a machine learning model to predict residuals based on the suite of geomorphic, hydrologic, and atmospheric variables. They evaluated abrupt loss of channel capacity by identifying shifts from a positive residual to a negative residual about the average stage-discharge curve. Only residuals outside the 95% confidence bounds of the stage discharge curve were considered. They quantified the likelihood of change after a storm as the percentage of residuals that underwent a shift from positive to negative and where the residual was outside the 95% confidence bounds of the average stage discharge curve. The authors also identify correlation and important variables for accurately predicting residuals from the machine learning model.

The overall method, data, and evaluation technique has the potential to provide a valuable contribution to predicting variability in channel capacity through residuals of the average stage-discharge curve. Inquiry into relevant scientific questions are presented. However, the current interpretation and analysis makes assumptions that may not be valid, lacks clarity, and requires more direct links between cause and effect than are stated within the article. Therefore, the article requires major revisions, including specificity of research aims, results interpretation, consideration of applied terminology, and acknowledgement of additional limitations. For instance, the first aim of the paper is to map the spatial variability of geomorphic response to extreme storm events, but the authors fail to acknowledge or address spatial correlation and bias in the stream gaging network. The definition of extreme in this article is unclear and it is unknown to what extent the included storms are extreme or quite frequent. The second aim is to understand the impacts of these storms on the stage-discharge relationships at gaged sites as a proxy for changes in flood hazard. However, this makes the assumptions that the storms alone are responsible for any observed changes in the residuals. While possible, other geomorphically significant events could have occurred that are unaccounted for. Further, the authors include other metrics in addition to the storms for predicting residuals, which makes it difficult to separate the impact of other drivers from the storms. The following subsections include general comments on various sections of the paper. Subsequently, specific comments are provided on a line-by-line basis.

**Introduction**

- the short paragraphs appear and read choppy. Consider combining paragraphs where subject matter allows.
- In the introduction, the authors imply that "extreme" storms or events are predominantly responsible for abrupt shifts in channel capacity and thus flood hazards. It is important to recognize that extreme storms/events likely contribute significantly to the population of abrupt shifts in channel capacity. However, there might be more frequent events that contribute to

these changes as well, particularly depending on channel response potential (e.g., a sand bed river with high sediment supply and non-cohesive banks vs. a gravel bed river with heavily vegetated banks.) Thus, it is recommended to re-consider the use of "extreme" and apply more focus on "abrupt" channel changes to more accurately state the study objectives. For instance, it is not clear to what degree the population of storms included in the analysis is composed of "extreme" storms and what classifies those storms as extreme.

- Line 102: How might this tool be used at ungagged sites without the detailed and rich dataset available? If applicable, it would be beneficial to highlight the use and importance of the tool in the conclusions.

**Materials and Methods**

- The authors should acknowledge the bias of stream size representation and spatial density in the gaging network and how this might impact spatial interpretation of results. Some sizes and areas are vastly under- and over-represented.
- The method for computing likelihood of change ignores monotonic trends in decreasing capacity – increasingly negative residual. If the residuals become more and more negative, it indicates channel capacity is decreasing, but this is not accounted for by only counting shifts from positive to negative. This limitation should be acknowledged. To some degree, the reported method only accounts for oscillating shifts – positive residual to negative residual then positive residual to negative residual.

**Results Analysis**

- Why did the authors choose to provide a results analysis section instead of organizing as results and discussion. The overall coherence and understanding of the results would be improved by breaking the results analysis section up into a results and discussion section.

**Specific Comments**

- Line 30: It is not entirely clear what is meant by traditional "cause-effect" studies. I presume the authors are referring to changes in peak flows due to changes in causal mechanisms such as climate, land use, etc.
- Line 32: How are might they over- or under-estimate actual damage, and what damage? Perhaps a follow-up example or additional explanation would clarify this sentence.
- Line 34: This is, in effect, what fluvial geomorphology is, and this sentence is somewhat redundant with the rest of the paragraph.
- Line 39: also critically modify the landscape and climate(???)
- Line 40: I am not sure flood risk is something that we measure more so than we estimate. Flood risk in fact can be highly uncertain Further, it is not only based on flood frequency, but the relationship between magnitude and frequency as is typically described by a distribution of peak flow, which are discretized as either annual maxima or peaks over threshold. Not just based on flood frequency.
- Line 41 - 43: Recent works have employed methods that incorporate changing channel capacity:
  - Stephens, T. A., & Bledsoe, B. P. (2023). Flood Protection Reliability: The Impact of Uncertainty and Nonstationarity. *Water Resources Research*, *59*(2), e2021WR031921.

- o Stephens, T. A., & Bledsoe, B. P. (2020). Probabilistic mapping of flood hazards: Depicting uncertainty in streamflow, land use, and geomorphic adjustment. *Anthropocene*, *29*, 100231.

- Line 44: This is poor wording, the amount of water that flows through the river systems during floods could in fact change in some situations. Revise to a more correct sentence or consider removing the first portion.
- Line 47: I presume by the use of frequency, the authors are describing the discharge magnitude of the flood. Instead of frequency, consider revising to magnitude, flow, or discharge since they are referring to the size and not how often it floods during a single event.
- Line 49: please give an example of some flood properties.
- Line 54: magnitude, frequency, and risk.
- Line 55: Do the changes have to be rapid? What about long term trends that are not accounted for. Consider shifts in the mean vs. monotonic trends. Sometime flood hazard maps are not updated for a decade or more, beckoning a definition of rapid in this context.
- Line 58: are the trends in stage or erosion/deposition or both comparable to trends in peak streamflow?
- Figure 1 would benefit from a scale bar.
- Line 70: How do we know these are "sharp", and how do we know the revisions are "upward"? Couldn't they be downward if erosion occurred?
- Line 95: "Despite some limitations" is used to start the previous sentence. Consider removing from one of the sentences. This sentence would read more formally by re-writing to remove the words "we" and "us".
- Line 148: please define gaps in the measurements. The manual field measurements may follow irregular frequency. Therefore, what constituted a gap? Minor gaps or missing data in the regular stage-flow measurements by the gage may not have a substantial impact on the analysis.
- Line 155: stage, water level, or water surface elevation is more clear than "levels"
- Figure 3a would be improved by indicating the flood stage. Near a stage of 2m, there is not much difference in the pre and post 2007 measurements. Is this due to overbank flow?
- Figure 3: "In (b), some outlier residuals are evident, likely due to shifts in measurement locations. These points were filtered out before performing the ML training." Belongs in the text rather than the figure caption.
- Figure 3 c and d caption: Is it in fact channel area and width or wetted area and width? The use of channel over wetted mean two different things. The wetted area and width can change for a single channel geometry. Please clarify at line 160 as well.
- Figure 3d: Should the y-axis label and caption be area or volumetric rate? Contradicts what is reported at lines 160 – 162. For area use area. For capacity use flow rate. Please clarify.
- Figure 3: Please note that Figure 3c and possibly 3d (depending on capacity or area) could fluctuate due to differences in measurement location, which can vary substantially from measurement to measurement. Even if measurement locations are close in distance, they may be upstream or downstream of a bridge. These factors must be considered when comparing widths to evaluate changes in the channel.
- Line 178: does a frequency of 520 events at a gage disqualify them as "extreme"? This seems like a high frequency.

- Line 181: The authors might improve clarity by explicitly stating each gage measurement contained 5 different median storm characteristics – 1 median storm characteristics for the five different lag times considered. If I am interpreting this correctly.
- I understand it would be difficult to graphically convey this in the paper, but I am wondering if the authors investigated the sensitivity of median storm characteristics to lag time. I wonder how much difference there is here. It is not essential, but if available, a note on this would be interesting.
- Table 1 would be more easily viewed in landscape layout and perhaps broken into 3 different tables. One table for each variable classification (geomorphic, hydrologic, and atmospheric).
- Table 1: Should the RFACT (Rainfall runoff factor) be classified as hydrologic instead of geomorphic?
- Line 196: As per previous comments, how do we know they are "severe"? Do the median characteristics reflect this?
- Line 326: comparing the predicted residual with the average residual - Why was this done? Was it for validation?
- Line 332: Some change is neglected in this computation: negative to positive, positive to more positive, and negative to more negative. Therefore, this sentence is somewhat inaccurate.
- Line 332 to 335: How was the confidence interval for the stage-discharge relationship computed?
- Line 344: Does this show the importance of geomorphology  of the watersheds or bias in the number of variables selected to represent each variable class? In this interpretation, the authors have neglected the fact that there are different numbers of variable classes. Simply the inclusion of more in once class than the other does not directly translate to its importance in this case. The following sentence does fit the authors interpretation.
- Line 351: drainage density is correlated with other variables as well, such as precipitation.
- Line 360: There is no evidence that flow regulation structures are the cause for these findings. It might suggest it if hydro_disturb_index only reflects flow regulation structures, but it could also include urbanization.
- Table 2: The caption should state what the Corr and RMSE compare.
- Line 374: It would be helpful to know something about the distribution of residuals to provide context to the RMSE magnitudes.
- Line 391: Is it the spatial "spread" or spatial "patterns"?
- Line 418: Low flows are more of a hydrologic property rather than a morphodynamic property.
- Line 450: Vulnerability was not defined, quantified, or reported anywhere prior to this.
- Line 473: In this sentence, it is not clear how the FHF increased logarithmically. How do we know this from the data presented?
- Line 481: Directly comparing regions does not account for spatial correlation or representation bias in the gaging network. Some areas/regions and streams are more represented than others making a comparison between regions misleading.
- Line 494: Only a portion of the streams in the Atlantic Plain are tidally influenced by the ocean. Further, an even smaller portion of the gages are. This sentence is not supported and speculative at best.

- Line 508 – 509: How does it confirm this? Please explicitly link the supporting evidence with this statement. As it is written, the previous sentences do not provide any clear evidence of this.
- Line 511: A citation would strengthen this statement.
- Line 515: many different sediment types may exist within a physiographic region. Sand bed channels, gravel bed, cobble, etc. The Appalachian highlands for example. Broadly inferring sediment types by physiographic regions is tenuous. I would suggest trying to relate sediment types to other variables such as drainage area and slope. See:
  - Montgomery, D. R., & Buffington, J. M. (1998). Channel processes, classification, and response. *River ecology and management: lessons from the Pacific Coastal Ecoregion*, 13-42.
  - Flores, A. N., Bledsoe, B. P., Cuhaciyan, C. O., & Wohl, E. E. (2006). Channel-reach morphology dependence on energy, scale, and hydroclimatic processes with implications for prediction using geospatial data. *Water Resources Research*, *42*(6).
- Line 545 – 555: the connections between the centroid of perception, flash floods, and residuals is not made clear here. The centroid of precipitation is an important variable in the analysis for predicting residuals. How does this tie to flash floods? Please explain more clearly.
- Line 575 – 580: this not necessarily true. Just because the channel conveyance capacity is exceeded does not mean the channel is expected to change. The flood must result in a geomorphically significant conditions of hydraulic and sediment supply conditions. The authors mentioned previously the importance of hysteresis in sediment deposition. This sentence over-simplifies and incorrectly categorizes a complex, situationally unique, and nuanced process.
- Line 599: The use of "future" here is misleading since the authors explicitly evaluate short-term or abrupt shifts. The temporal persistence of that shift is not addressed.
- Line 600: As it stands, specific impacts from individual drivers is insufficiently addressed. A more accurate representation of the analysis would be to say that the method identified important drivers for predicting residuals from the average stage discharge curve. From my understanding, the analysis does not necessarily reveal the actual impact of specific variables on the predicted residual.
- Line 608: Did the authors mean to say channel capacity here instead of "river discharge"?
- Line 615: More specifically, the risk of immediate reduction in channel capacity. The authors did not evaluate increases in channel capacity.
- Line 616: Knowing the temporal persistence of these changes would provide insight to the feasibility of these updates or alternative methods for quantifying flood risk if the process is highly variable in time.

**Technical Corrections**
- Line 284 and 285: The numbers indicating the numbered list should contain parentheses after the number e.g.: 1) text…., 2) text…,
- Line 341: typo ". analysis."
- Line 387-388: Revise: "We have got". Improved writing would be something like "There are 12 clusters of gages"
- Line 517: typo. Remove period.
- Line 542: add a "the" between "in" and "literature".

- Line 543: Missing a space after the parentheses.
- Line 551: New paragraph? Indent if so.

---

## Community Comment (CC1)

[revised manuscript text omitted]

**Appendix A**

[Figure]

[Figure]

[Figure]

945

[Figure]

[Figure]

[Figure]

[Figure]

[Figure]

[Figure]

[Figure]

[Figure]

[Figure]

[Figure]

[Figure]

g

950

[Figure]

[Figure]

[Figure]

[Figure]

[Figure]

[Figure]

[Figure]

[Figure]

[Figure]

[Figure]

955

**Figure A1: Gages with clustering identification assigned by SOM unsupervised clustering (a-l)**

---

## Author Comment (AC2)

*First, we want to thank the reviewer for the insightful detailed comments and recommendations. We appreciate your outlook on the potential of our study.*

*Following the suggestions, we will revise the manuscript accordingly. Below we provide responses detailing the revision actions we plan to take to address the reviewers' comments.*
**Note: Below is our response (italics) to each reviewer's comment (regular font)**

General Comments

This study quantified residuals in average stage-discharge rating curve from manual field measurements at U.S. Geologic Survey stream gaging stations. The residuals about the average stage-discharge curve quantify changes in channel capacity by evaluating the change in discharge required to achieve a certain water stage. For each measurement, at each gage, the authors quantified a set of geomorphic, hydrologic, and atmospheric variables. Included in these variables were individual storm properties. Assign of storm properties for each measurement were quantified by considering a lag time and computing the median storm property for all storms within that lag time. Lag times of 15, 30, 90, 180, and 365 days prior to the stage-discharge measurement were considered. The authors then trained and validated a machine learning model to predict residuals based on the suite of geomorphic, hydrologic, and atmospheric variables. They evaluated abrupt loss of channel capacity by identifying shifts from a positive residual to a negative residual about the average stage-discharge curve. Only residuals outside the 95% confidence bounds of the stage discharge curve were considered. They quantified the likelihood of change after a storm as the percentage of residuals that underwent a shift from positive to negative and where the residual was outside the 95% confidence bounds of the average stage discharge curve. The authors also identify correlation and important variables for accurately predicting residuals from the machine learning model.

The overall method, data, and evaluation technique has the potential to provide a valuable contribution to predicting variability in channel capacity through residuals of the average stage-discharge curve. Inquiry into relevant scientific questions are presented. However, the current interpretation and analysis makes assumptions that may not be valid, lacks clarity, and requires more direct links between cause and effect than are stated within the article. Therefore, the article requires major revisions, including specificity of research aims, results interpretation, consideration of applied terminology, and acknowledgement of additional limitations. For instance, the first aim of the paper is to map the spatial variability of geomorphic response to extreme storm events, but the authors fail to acknowledge or address spatial correlation and bias in the stream gaging network.

*Response 1: We thank the reviewer for all the insightful comments: in our revised manuscript we will incorporate the recommended changes to represent the novelty and originality of our work more clearly. We will particularly focus on explaining the methods and rationale behind every step in the procedure. We will do our best to bridge the stated aim, interpretation, and limitations of the study following the reviewer's suggestions.*
*Regarding the coverage of stream gages, we agree on the intrinsic limits of the dataset, based on the fact that there is variability across CONUS regarding the spatial and temporal coverage of stream gages. These limits in general have been addressed in literature and are very well summarized in the publication by* Kiang et al., (2013)*.*

*Overall, gage coverage is higher in the Eastern United States compared to the Western United States. The arid Southwestern United States, Alaska, and Hawaii show the lowest spatial coverage, and these regions,*

*except for Hawaii, often have short streamflow records. Gage statistics quality, according to Kiang et al. 2013 also varies across the country, mostly due to variations in hydrology. Notably, the arid and semiarid areas in the Central and Southwestern United States exhibit higher interannual variability in flow, leading to greater uncertainty in flow statistics. These findings will be discussed further in our results section. Despite these observations, it's important to note that any research relying on gaging sites faces challenges of potential over or underrepresentation, which we will also emphasize in the discussion of the limitations and advantages of our proposed model.*

The definition of extreme in this article is unclear and it is unknown to what extent the included storms are extreme or quite frequent.

***Response 2:** We will revise the manuscript and we will revise the wordings*

The second aim is to understand the impacts of these storms on the stage-discharge relationships at gaged sites as a proxy for changes in flood hazard. However, this makes the assumptions that the storms alone are responsible for any observed changes in the residuals. While possible, other geomorphically significant events could have occurred that are unaccounted for.

***Response 3:** We thank the reviewer for this comment. Indeed, channel changes can be due to other geographically significant events (e.g. landslides, debris flow etc), however, such occurrences could also be triggered by storm events. At this stage, we have a complete database of storm properties, but we did not include a complete analysis of additional events such as mass movements. This goes beyond the scope of this work. In the revision, however, we will also highlight this point.*

Further, the authors include other metrics in addition to the storms for predicting residuals, which makes it difficult to separate the impact of other drivers from the storms.

***Response 4:** We thank the reviewer for this comment. We believe that adding the other variables gives a better context to the impact of the storms. Overall, we have done a feature importance analysis and selected only those drivers that are the most important and influential. The feature importance itself, highlighted which variables mattered the most in predicting the residuals. We decided to use hydrologic and geomorphological variables because landscape properties are also linked to the potential effects of storms.*

The following subsections include general comments on various sections of the paper. Subsequently, specific comments are provided on a line-by-line basis.

Introduction

the short paragraphs appear and read choppy. Consider combining paragraphs where subject matter allows.

***Response 5:** We will read through the manuscript carefully and introduce edits as needed.*

In the introduction, the authors imply that "extreme" storms or events are predominantly responsible for abrupt shifts in channel capacity and thus flood hazards. It is important to recognize that extreme storms/events likely contribute significantly to the population of abrupt shifts in channel capacity. However, there might be more frequent events that contribute to these changes as well, particularly depending on channel response potential (e.g., a sand bed river with high sediment supply and non-cohesive banks vs. a gravel bed river with heavily vegetated banks.) Thus, it is recommended to re-consider the use of "extreme" and apply more focus on "abrupt" channel changes to more accurately state the

study objectives. For instance, it is not clear to what degree the population of storms included in the analysis is composed of "extreme" storms and what classifies those storms as extreme.

*Response 6: Thank you for this recommendation. We have thought about this comment and decided to find an alternative to the word "extreme events" in the revised manuscript.*

Line 102: How might this tool be used at ungagged sites without the detailed and rich dataset available? If applicable, it would be beneficial to highlight the use and importance of the tool in the conclusions.

*Response 7: We thank the reviewer for this comment. We believe that as USGS stream gage information could potentially be transferred from nearby stream gages if there is sufficient similarity between the gaged watersheds and the ungaged watersheds of interest, our model could also be applied to ungaged sites.*
*However, one must always keep in mind that the successful 'translation' to ungaged environments depends on the correlation of the stream gages in the surrounding areas. For example, there are areas of CONUS (mostly mountainous) that show highly correlated stream gages (Kiang et al., 2013), whereas the Central United States and coastal areas of the Southeastern United States show much uncorrelated gages. Therefore, the goodness of the information transfer might not work as well. Also, transferability would be most likely to be successful when basin attributes show high similarity and storm properties are within the range of variability of the training set used for this work. We will add some consideration about this in the manuscript*

Materials and Methods

The authors should acknowledge the bias of stream size representation and spatial density in the gaging network and how this might impact spatial interpretation of results. Some sizes and areas are vastly under- and over-represented.

*Response 8: Regarding the coverage of stream gages, we agree on the intrinsic limits of the dataset, based on the fact that there is variability across CONUS regarding the spatial and temporal coverage of stream gages. These limits in general have been addressed in literature and are very well summarized in the publication by Kiang et al., (2013).*

*Broadly speaking, the Eastern United States has better coverage compared to its Western counterpart. Particularly, the arid Southwestern United States, Alaska, and Hawaii show notably lacking spatial coverage. Except for Hawaii, these regions also tend to be covered by shorter streamflow records. Discrepancies in hydrology contribute to variations in the statistical uncertainty calculated across different parts of the country (Kiang et al., 2013). The Central and Southwestern United States, characterized by arid and semiarid conditions, generally display higher interannual variability in flow, resulting in increased uncertainty in flow statistics. In the revised manuscript, we will incorporate these comments. Despite these distinctions, it's essential to recognize that any research relying on gaging sites faces similar limits and is overall affected by potential over or underrepresentation of flows. This aspect will be emphasized further in the revised manuscript, in the section about limitations and advantages of the proposed model.*

The method for computing likelihood of change ignores monotonic trends in decreasing capacity – increasingly negative residual. If the residuals become more and more negative, it indicates channel capacity is decreasing, but this is not accounted for by only counting shifts from positive to negative. This

limitation should be acknowledged. To some degree, the reported method only accounts for oscillating shifts – positive residual to negative residual then positive residual to negative residual.

*Response 9: We thank the reviewer for this comment. Indeed, we focus mainly on sudden shifts, rather than on permanent shifts. The main reasons for this were - 1. Short-term conveyance capacity changes are not considered in typical flood hazard assessments and could substantially overstate or understate flood threats at any particular time for subsequent floods; 2. there is a plethora of complex and sometimes not linear-processes and coupled feedback that we would need to 'model' in the training set, to provide a comprehensive benchmark to identify permanent shifts vs sudden ones, and this could be a potentially interesting research that could be tackled by further studies building on our model, but at this stage it goes beyond the scope of this work. We will highlight this point better in the manuscript.*

Results Analysis

Why did the authors choose to provide a results analysis section instead of organizing as results and discussion. The overall coherence and understanding of the results would be improved by breaking the results analysis section up into a results and discussion section.

*Response 10: We will add a discussion section as per reviewer's suggestion*

Specific Comments

Line 30: It is not entirely clear what is meant by traditional "cause-effect" studies. I presume the authors are referring to changes in peak flows due to changes in causal mechanisms such as climate, land use, etc.

*Response 11: yes, this is correct. Giving this comment, we will rephrase this sentence*

Line 32: How are might they over- or under-estimate actual damage, and what damage? Perhaps a follow-up example or additional explanation would clarify this sentence.
*Response 12: We will revise this part of the manuscript*

Line 34: This is, in effect, what fluvial geomorphology is, and this sentence is somewhat redundant with the rest of the paragraph.
*Response 13: We will revise this part of the manuscript*

Line 39: also critically modify the landscape and climate(???)
*Response 14: We will rephrase the sentence for better clarity.*

Line 40: I am not sure flood risk is something that we measure more so than we estimate. Flood risk in fact can be highly uncertain Further, it is not only based on flood frequency, but the relationship between magnitude and frequency as is typically described by a distribution of peak flow, which are discretized as either annual maxima or peaks over threshold. Not just based on flood frequency.
*Response 15: Yes, this is correct.  We will rephrase it to 'flood risk estimation'. We will also improve the wording of this sentence.*

*Flood risk measurement has traditionally been based on flood frequency, derived from variability in streamflow, assuming constant channel capacity (Merz et al., 2012; Slater et al., 2015). The relationship between magnitude and frequency is also generally built upon the peak flow distribution, whereas peaks*

*are discretized as either annual maxima or peaks over threshold, but mostly assuming that river capacity remains constant over the investigation records.*

Line 41 - 43: Recent works have employed methods that incorporate changing channel capacity:

- o Stephens, T. A., & Bledsoe, B. P. (2023). Flood Protection Reliability: The Impact of Uncertainty and Nonstationarity. *Water Resources Research*, *59*(2), e2021WR031921
- o Stephens, T. A., & Bledsoe, B. P. (2020). Probabilistic mapping of flood hazards: Depicting uncertainty in streamflow, land use, and geomorphic adjustment. *Anthropocene*, *29*, 100231.

**Response 16:** *Thank you for the references. We will add these to the manuscript and rephrase the text.*

Line 44: This is poor wording, the amount of water that flows through the river systems during floods could in fact change in some situations. Revise to a more correct sentence or consider removing the first portion.
**Response 17:** *We will revise this part of the manuscript*

Line 47: I presume by the use of frequency, the authors are describing the discharge magnitude of the flood. Instead of frequency, consider revising to magnitude, flow, or discharge since they are referring to the size and not how often it floods during a single event.
**Response 18:** *We will consider the suggestion and revise this part of the manuscript*

Line 49: please give an example of some flood properties.
**Response 19:** *We will clarify that we refer to inundation extent and depth*

Line 54: magnitude, frequency, and risk.
**Response 20:** *We will rephrase*

Line 55: Do the changes have to be rapid? What about long-term trends that are not accounted for. Consider shifts in the mean vs. monotonic trends. Sometime flood hazard maps are not updated for a decade or more, beckoning a definition of rapid in this context.
**Response 21:** *For this work, we investigated sudden changes of positive to negative residuals. We acknowledge that these might not be permanent changes. Given the complexity of processes involved in the 'restoration' of river forms, or the permanence of a channel shift, we decided to focus this work on the sudden changes, as proposed by Slater et al. 2015 in her work. With this idea, with the proposed method we highlight rivers more prone to changes in the aftermath of a storm, highlighting potential increased flood hazard in the case of subsequent storms.*

*In literature, using Slater's concept, the work by Ahrendt et al 2022 offers an overview of historic long-term and short-term conveyance changes for WA, whereas the work of Li et al 2020 highlighted how relatively modest long-term changes in river channel capacity are composed of numerous short-term transients which are of much larger magnitude. We referred to this work in our manuscript and will add some considerations on the fact that this work only considers sudden changes but not their persistence in time.*

Line 58: are the trends in stage or erosion/deposition or both comparable to trends in peak streamflow?
**Response 22:** *Other works in literature highlighted that some channel changes could provoke a higher change in flood hazard than shifts in discharge alone (Slater et al. 2015, 2016, Li et al. 2020, Ahrendt*

*et al 2022). For this work, we did not assess changes in streamflow, as we are training the model based on storm properties, and not on long term discharge properties.*

Figure 1 would benefit from a scale bar.
**Response 23:** *We will add this in the revised figure*

Line 70: How do we know these are "sharp", and how do we know the revisions are "upward"? Couldn't they be downward if erosion occurred?
**Response 24:** *Indeed, the changes could be downward if erosion occurred. We imply that an upward revision is a proxy for an increase in flood hazard, whereas a downward revision potentially could mean a reduced hazard. Our analysis is consistent with other works in the literature relating shifts in the stage-discharge relationship as a proxy for flood hazard.*

Line 95: "Despite some limitations" is used to start the previous sentence. Consider removing from one of the sentences. This sentence would read more formally by re-writing to remove the words "we" and "us".
**Response 25:** *We will consider the suggestion and revise this part of the manuscript*

Line 148: please define gaps in the measurements. The manual field measurements may follow irregular frequency. Therefore, what constituted a gap? Minor gaps or missing data in the regular stage-flow measurements by the gage may not have a substantial impact on the analysis.

**Response 26:** *We have excluded the gages that do not have continuous data for the tie frame from 2002-2013. We will clarify this better in the manuscript. For the work, aside from considering consistent gages present in the Shen et al. Database, and covered by stream measurements, we applied the same criteria as Slater et al. (2015), who only considered field measurements in which the discharge is within one percent of the product of channel velocity and cross-sectional channel area, as reported by the USGS, and those made in close proximity to the gage station.*

Line 155: stage, water level, or water surface elevation is more clear than "levels"
**Response 27:** *We will rephrase this based on the reviewer's suggestion*

Figure 3a would be improved by indicating the flood stage. Near a stage of 2m, there is not much difference in the pre and post 2007 measurements. Is this due to overbank flow?

**Response 28:** *The figure was shown as an example of shifts present in the measurement data. For this gage, the flood stage is at 10ft, and the peak discharge of the 2007 event was 11.51 ft, and the Quinnipiac River itself (at the gage right upstream of the one in the picture) measured the maximum discharges for the period of record of the station during the 2007 flood. Aside from the information provided by USGS on that event, we do not have direct knowledge of the event itself so we cannot make a precise statement on the reason behind the similarities highlighted by the reviewer.*

Figure 3: "In (b), some outlier residuals are evident, likely due to shifts in measurement locations. These points were filtered out before performing the ML training." Belongs in the text rather than the figure caption.
**Response 29:** *We will move this part to the text*

Figure 3 c and d caption: Is it in fact channel area and width or wetted area and width? The use of channel over wetted mean two different things. The wetted area and width can change for a single channel geometry. Please clarify at line 160 as well.

***Response 30:*** *The figure reports the channel width as reported in the gage measurements. We will clarify this in the manuscript.*

Figure 3d: Should the y-axis label and caption be area or volumetric rate? Contradicts what is reported at lines 160 – 162. For area use area. For capacity use flow rate. Please clarify.

***Response 31:*** *There was a mistake in the text, the line should have read 'and channel conveyance (Figure. 3d).' We will rephrase this in the text.*

Figure 3: Please note that Figure 3c and possibly 3d (depending on capacity or area) could fluctuate due to differences in measurement location, which can vary substantially from measurement to measurement. Even if measurement locations are close in distance, they may be upstream or downstream of a bridge. These factors must be considered when comparing widths to evaluate changes in the channel.

***Response 32:*** *According to the information of the gage, the measurements did not shift in location. For the work itself, consistently with Slater and the open codes provided in her work, we removed all field measurements made in a different (or potentially different) location, and all field measurements made in icy conditions, as these might affect measurements of channel geometry. We will highlight this more clearly in the manuscript.*

Line 178: does a frequency of 520 events at a gage disqualify them as "extreme"? This seems like a high frequency.
***Response 33:*** *Indeed, the reviewer is correct. The magnitude of these events varied in time, and in the revised manuscript we will refrain from defining 'extreme' the events.*

Line 181: The authors might improve clarity by explicitly stating each gage measurement contained 5 different median storm characteristics – 1 median storm characteristics for the five different lag times considered. If I am interpreting this correctly.
***Response 34****: Yes, this is correct. We will clarify this in the manuscript as suggested.*

I understand it would be difficult to graphically convey this in the paper, but I am wondering if the authors investigated the sensitivity of median storm characteristics to lag time. I wonder how much difference there is here. It is not essential, but if available, a note on this would be interesting.
***Response 35****: Thank you for this comment. We have not considered this at this stage. For the revised work, we will add a note if we see any meaningful results.*

Table 1 would be more easily viewed in landscape layout and perhaps broken into 3 different tables. One table for each variable classification (geomorphic, hydrologic, and atmospheric).
***Response 36****: Thank you for the suggestion. We will try to revise this as suggested*

Table 1: Should the RFACT (Rainfall runoff factor) be classified as hydrologic instead of geomorphic?
***Response 37****: There is a mistake. We will correct this.*

Line 196: As per previous comments, how do we know they are "severe"? Do the median characteristics reflect this?
***Response 38****: We will rephrase this in the manuscript*

Line 326: comparing the predicted residual with the average residual - Why was this done? Was it for validation?

**Response 39**: *The reviewer is correct. We did this to validate the model.*

Line 332: Some change is neglected in this computation: negative to positive, positive to more positive, and negative to more negative. Therefore, this sentence is somewhat inaccurate.

**Response 40**: *We will rephrase this*

Line 344: Does this show the importance of geomorphology of the watersheds or bias in the number of variables selected to represent each variable class? In this interpretation, the authors have neglected the fact that there are different numbers of variable classes. Simply the inclusion of more in once class than the other does not directly translate to its importance in this case. The following sentence does fit the authors interpretation.

**Response 41**: *We will rephrase this and clarify.*

Line 360: There is no evidence that flow regulation structures are the cause for these findings. It might suggest it if hydro_disturb_index only reflects flow regulation structures, but it could also include urbanization.

**Response 42**: *We will rephrase this.*

Line 481: Directly comparing regions does not account for spatial correlation or representation bias in the gaging network. Some areas/regions and streams are more represented than others making a comparison between regions misleading.

**Response 43**: *We will rephrase this adding the comments regarding the distribution of gages across CONUS.*

Line 494: Only a portion of the streams in the Atlantic Plain are tidally influenced by the ocean. Further, an even smaller portion of the gages are. This sentence is not supported and speculative at best.

**Response 4**: *We will rephrase this*

Technical Corrections

**Response:** *We will carefully fix all the technical corrections mentioned by the reviewer and revise the manuscript.*

References:

Kiang, J. E., Stewart, D. W., Archfield, S. A., Osborne, E. B., Eng, K., & Survey, U. S. G. (2013). A national streamflow network gap analysis. In *Scientific Investigations Report*. https://doi.org/10.3133/sir20135013

---

## Author Comment (AC3)

*First of all, we want to thank the reviewer for the insightful detailed comments and recommendations. Specific responses to the reviewers' comments are added below.*
***Note: Below is our response (italics) to each reviewer's comment (regular font)***

1. The articles needs desperately a discussion section separated from the results. Right now, everything is cramped into a single section that makes difficult to take the main points out.

***Response:*** *We will add a separate discussion section as the reviewer suggested and state the main points clearly for the readers.*

2. I would also add as a minimum a paragraph in the conclusion section (if not its own standalone section) about the framework limitations. This is crucial in any prediction framework so readers are aware of it when taking decision.

***Response:*** *We will revise the conclusion section and add more information on the limitations of the framework.*

3. The introduction (and other sections) have too many small paragraphs (1-3 sentences) that disrupts the reading of the manuscript. I will suggest to combine paragraphs that convey the same message.

***Response:*** *We will try to revise the introduction section as per the reviewer's suggestion. We will try to make sure of the flow of reading by grouping together consistent messages.*

4. Did the authors had a minimum years of data treshold when selecting the gauges? It seems imperative to have one, since a gauge with 5 years of data will yield very different results than one with 50 years of data. Also, what are the general statistics of the gauge data? For example, what is the mean length of record, amount of cross sections measurements, etc.

***Response:*** *For this work, for the gages data, we followed Slater's 2015 workflow and used the system they provided accompanying the published work. Overall, one must consider that for our model we focus on sudden changes of the stage discharge, and not their persistence in time. We evaluate the stage discharge relationship based on measured river properties, as by Slater's 2015 work. Following the referenced work, we detected and excluded sites featuring artificial controls at the gauging station that could impede the natural adjustment of the channel's shape. Additionally, we eliminated all field measurements conducted at a different location or potentially different location, along with those taken in icy conditions, as these factors could impact the accuracy of channel geometry measurements. Our selection process retained only sites with comprehensive time series data, and as per Slater's et al. 2015 work, only kept gages with 99.7% completeness in streamflow records and 40 channel cross-section measurements. We will clarify this in the revised manuscript.*

Specific Comments:

Line 44: is not clear the statement. Since during a flood event, flow within the channel can change due to external factors, stormwater discharge o compounding flood at coastal estuaries.

***Response:*** *The reviewer is correct, many other parameters could be the direct cause of change, also considering for example debris flows or mass movements. Nonetheless, these could also be triggered by the storm properties themselves. We will add some comments on this in the revised manuscript.*

Line 64: the "secondary channel" that is referring in Figure 1 should be highlighted in the figure itself to help the reader understand the point.

***Response:*** *We will revise the figure for better understanding.*

Figure 1: The figure needs a north arrow and scale bar. I also strongly suggest the authors to use a GIS platform to enhance the quality of the figure. The figure also needs a location map.

***Response:*** *We will add the mentioned specifics to the figure.*

Line 93 and 95: both sentences start with "Despite some limitations ..." Please rephrase.

***Response:*** *We will rephrase the sentences.*

Line 106: I will summarize all the gauges selected with their corresponding ID in a text file (or any other format file) and upload it to a repository for easy sharing. Then, the reader could see exactly which stations were selected. This helps the open data statement in the research community.

***Response:*** *Thank you for the suggestion. We will do this.*

Line 110: did the authors downloaded also discharge values from the NWS or it was just flood stages as it is mentioned in this line? If the cross section data (width and depth of the river) were obtained from the USGS, why not also use their created flow-stage curves. My biggest question is from where the authors obtained the discharge values for the creation of their rating curves, since NWS only provides stage level whereas USGS provides both stage and discharge in most gauges.

*For this work we followed Slater's 2015 workflow, and used the system they provided accompanying the published work. We evaluate the stage discharge relationship based on measured river values, as by Slater's 2015 work. We will clarify this in the manuscript.*

Figure 2: I do not support have several lines of text if the figure caption just to describe the different climate regions in the map. Also, the authors also explain the abbreviation in the results section when talking about it. Thus, I strongly recommend having a nomenclature section that summarizes all of these, including the variables from table 1. Then, the reader can easily find it.

***Response:*** *We will revise this.*

Line 161: the statement of the reason for change in capacity (deposition) has been already mentioned in Line 159. Please rephrase or remove.

***Response:*** *We will revise this.*

Line 177: why that the authors only focused on a very narrow range of years for their storm event? This seems like a big limitation, especially since the latest year of the record was a decade ago. The authors needs to justify their selection as a minimum.

***Response:*** *We want to thank the reviewer for this comment. We would like to draw the reviewer's attention to the fact that in our study we have used a published dataset (Shen et al. 2017) of storm events and the properties of the events. Most of them were calculated properties. This dataset was from 2002-2013. Our framework showcases the intercorrelation of the different event properties that can affect the channel changes and that can be of any timelines. Here we have established a framework that can be used for extended timelines. Researchers can use the trained model with additional years of data, if they have available the same storm properties proposed by Shen for more recent events.*

Table 1: there are some variables that have their "unit" column empty. For example, Peak , Q2, etc. This might be a typo since if the variable does not has unit the authors specify with a dimensionless or N/A. Also, the table is too long for a peer-review article. I strongly suggest dividing the table into three separate ones, one for each variable type. There are also some variables like BFI_AVE that their description is a quarter of the page due to being squeezed in the small column width. I would suggest the authors to place the long variable description as a footer in the table or in the appendix as part of the nomenclature section.

**Response:** *We will revise the table as per the reviewer's suggestion.*

---

## Author Comment (AC4)

*First, we want to thank the reviewer for the insightful detailed comments and recommendations. We appreciate your outlook on the potential of our study.*

*Following the suggestions, we will revise the manuscript accordingly. Below we provide responses detailing the revision actions we plan to take to address the reviewers' comments.*

**Note: Below is our response (italics) to each reviewer's comment (regular font)**

Major Comments:

1. The authors state in their paper objectives and title that they are exploring the impact of major storm events on flood hazard through changes in channel capacity. However, the explanatory variables used in their machine learning models include not only storm-related properties but also hydrologic and geomorphologic variables. It's unclear how the authors discern from the ML models that the changes in channel capacity are primarily attributed to storm properties and not influenced by other factors. Additionally, the analysis is focused on the dataset containing "major storm events," which implies that changes in channel capacity are associated in the ML to major events. What about changes in channel capacity during non-storm events? In other words, can the ML capture changes in channel capacity without storm property variables (variables as described by Falcone, 2011)?

**Response:** *we thank the reviewer for this comment. The ML model was trained considering both storm properties and watershed properties. We do not make distinction on which element triggers the change, nonetheless in the paper we provided an assessment of feature importance, highlighting that the shifts, for how the model works, are mostly explained by a combination of storm and watershed properties. We would not suggest using the model, as it is trained currently, to predict changes without having information on the storm properties. We will highlight this in the revised manuscript.*

2. The overall organization of the introduction and methods sections lacks necessary details. To better understand the techniques used to estimate the residuals and the various simplifications (such as manual filtering of outliers) required for the methodology, I had to refer to Slater et al., (2015). For example, in Figure 3, panel b, the authors mention outliers but do not clearly identify them or provide specific details.

**Response:** *we thank the reviewer for this comment. We will provide a more detailed description for the revised paper.*

3. The results and discussion sections need to be reorganized. It is recommended to create a separate section for the discussion. Additionally, for improved readability, it is advisable to create a single section dedicated to limitations and future work.

**Response:** *we thank the reviewer for this comment. We will reorganize the work into separate sections for discussion and results.*

Minor Comments:

Figure 3: It would be helpful to include a time series with streamflow data to illustrate the magnitude of the April 2007 flood. Panels c and d are confusing since they may give the impression that there's only one change in flood capacity per gauge, which might not be the case.

**Response:** *we thank the reviewer for this comment. We will improve the clarity of the figure*

Line [15]: Please clarify what you mean by "geomorphologic impacts."

Consider adding a schematic figure that explains the core concept of conveyance capacity before and after a storm event. Real data examples would be beneficial in illustrating this concept. Slater et al., 2015, offers a useful example in this regard.

> **_Response:_** _we thank the reviewer for this comment. We will try and create a figure to express this._

Figure 9: Please provide information on how the 95% confidence bound of the current stage-discharge relationship was calculated.

> **_Response:_** _we thank the reviewer for this comment. We will clarify this. The fitting is performed using a loess fit, and the R packet for this provides the estimation of the 95% confidence interval._

---

## Author Comment (AC5)

*First, we want to thank Dr Gesch for the insightful suggestions and recommendations. Below please find some detailed response to the raised comments*
***Note: Below is our response (italics) to each comment (regular font)***

What was the magnitude (i.e., percent) of excluded measurements? Were any of the 3,101 gages excluded completely, and if so, how many?

**Response:** *we thank Dr Gesch for this comment. Following the referenced work, for the work we also detected and excluded sites featuring artificial controls at the gauging station that could impede the natural adjustment of the channel's shape. Additionally, we eliminated all field measurements conducted at a different location or potentially different location, along with those taken in icy conditions, as these factors could impact the accuracy of channel geometry measurements. Our selection process retained only sites with comprehensive time series data, and as per Slater's et al. 2015 work, only kept gages with 99.7% completeness in streamflow records and 40 channel cross-section measurements. We will clarify this in the revised manuscript.*

How many gages are excluded due to this criteria?

**Response:** *We will clarify this in the revised work*

Because STREAMS_KM_SQ_KM comes from 100k NHD, are there any concerns about "artificial" variability of stream density of cartographically-derived streams?

**Response:** *Indeed this is a critical point. Actual stream density could be different than that from cartography. Nonetheless, to avoid having biases and further fluctuations of values, we decided to consider this official source as it is available consistently for all gages. Researchers could also consider using different methods to define the drainage network, but we would suggest, in that case, to re-train the model and verify once again the importance of this parameter in the re-trained model. We will add a comment on this in the revised manuscript.*

What about channelized streams in urban areas (where the channel cannot change in response to extreme events)? Would the method used in this study recognize these cases?

**Response:** *This is a critical point. We excluded sites featuring artificial controls at the gauging station that could impede the natural adjustment of the channel's shape. We would not recommend this approach for engineered river reached where flood protection measures or artificial channelization is present. We will highlight this in the manuscript.*

Technical comments:
     *We will address all the missing references and highlighted needed technical clarifications.*

---

## Author Response (AR1)

Dear Editor,
We are pleased to resubmit for publication the revised version of
**"To What Extent Do Flood-inducing Storm Events Change Future Flood Hazards"**
We appreciate the constructive comments of the reviewers, which we have addressed as outlined below.
Along with the revised manuscript, we also enclose a tracked version of the document, with all the changes outlined.
Please also note that we have made a few further minor editing and changes throughout the manuscript to improve clarity and readability. These changes are indicated in the track changes document, but they might not be listed in our response.

**Response to the review HESS_RC1**

*First, we want to thank the reviewer for the insightful detailed comments and recommendations. We appreciate your outlook on the potential of our study.*
*Below we provide a detailed response (italics) to all raised issues (regular font).*

Please find my comments in the attached .pdf. The overall method, data, and evaluation technique has the potential to provide a valuable contribution to predicting variability in channel capacity through residuals of the average stage-discharge curve. Inquiry into relevant scientific questions are presented. However, the current interpretation and analysis makes assumptions that may not be valid, lacks clarity, and requires more direct links between cause and effect than are stated within the article. Therefore, the article requires major revisions, including specificity of research aims, results interpretation, consideration of applied terminology, and acknowledgement of additional limitations. For instance, the first aim of the paper is to map the spatial variability of geomorphic response to extreme storm events, but the authors fail to acknowledge or address spatial correlation and bias in the stream gaging network. The definition of extreme in this article is unclear and it is unknown to what extent the included storms are extreme or quite frequent. The second aim is to understand the impacts of these storms on the stage-discharge relationships at gaged sites as a proxy for changes in flood hazard. However, this makes the assumptions that the storms alone are responsible for any observed changes in the residuals. While possible, other geomorphically significant events could have occurred that are unaccounted for. Further, the authors include other metrics in addition to the storms for predicting residuals, which makes it difficult to separate the impact of other drivers from the storms. For these reasons among others, I suggest major revisions prior to reconsideration for publication. More detailed comments are provided in the attached document.

**Response:** *First, we want to thank the reviewer for the insightful detailed comments and recommendations. We appreciate your outlook on the potential of our study. Following the suggestions, we revised the manuscript accordingly.*

General Comments

This study quantified residuals in average stage-discharge rating curve from manual field measurements at U.S. Geologic Survey stream gaging stations. The residuals about the average stage- discharge curve quantify changes in channel capacity by evaluating the change in discharge required to achieve a certain water stage. For each measurement, at each gage, the authors quantified a set of geomorphic, hydrologic, and atmospheric variables. Included in these variables were individual storm properties. Assign of storm properties for each measurement were quantified by considering a lag time and computing the median storm

property for all storms within that lag time. Lag times of 15, 30, 90, 180, and 365 days prior to the stage-discharge measurement were considered. The authors then trained and validated a machine learning model to predict residuals based on the suite of geomorphic, hydrologic, and atmospheric variables. They evaluated abrupt loss of channel capacity by identifying shifts from a positive residual to a negative residual about the average stage-discharge curve. Only residuals outside the 95% confidence bounds of the stage discharge curve were considered. They quantified the likelihood of change after a storm as the percentage of residuals that underwent a shift from positive to negative and where the residual was outside the 95% confidence bounds of the average stage discharge curve. The authors also identify correlation and important variables for accurately predicting residuals from the machine learning model.

The overall method, data, and evaluation technique has the potential to provide a valuable contribution to predicting variability in channel capacity through residuals of the average stage-discharge curve. Inquiry into relevant scientific questions are presented. However, the current interpretation and analysis makes assumptions that may not be valid, lacks clarity, and requires more direct links between cause and effect than are stated within the article. Therefore, the article requires major revisions, including specificity of research aims, results interpretation, consideration of applied terminology, and acknowledgement of additional limitations. For instance, the first aim of the paper is to map the spatial variability of geomorphic response to extreme storm events, but the authors fail to acknowledge or address spatial correlation and bias in the stream gaging network.

*Response 1: We thank the reviewer for all the insightful comments. In our revised manuscript, we incorporated the recommended changes to represent the novelty and originality of our work more clearly. We mainly focused on explaining the methods and rationale behind every step in the procedure. We tried to bridge the stated aim, interpretation, and limitations of the study following the reviewer's suggestions.*
*Regarding the coverage of stream gages, we agree on the dataset's intrinsic limits, due to the variability across CONUS regarding the spatial and temporal coverage of stream gages. These limits in general have been addressed in literature and are very well summarized in the publication by* Kiang et al., (2013).
*Overall, gage coverage is higher in the Eastern United States compared to the Western United States. The arid Southwestern United States, Alaska, and Hawaii show the lowest spatial coverage, and these regions, except for Hawaii, often have short streamflow records. Gage statistics quality, according to Kiang et al. 2013 also varies across the country, mostly due to variations in hydrology. Notably, the arid and semiarid areas in the Central and Southwestern United States exhibit higher interannual variability in flow, leading to greater uncertainty in flow statistics. These findings have been discussed further in the revised manuscript. Despite these observations, it's important to note that any research relying on gaging sites faces challenges of potential over or underrepresentation. We added comments about this in the discussion of the limitations and advantages of our proposed model.*

The definition of extreme in this article is unclear and it is unknown to what extent the included storms are extreme or quite frequent.

*Response 2: We have revised the manuscript, and we have used "Flood-inducing Storm Events" instead of "extreme events", to be consistent with the work of Shen et al., 2017 where they refer to the dataset as a dataset of flood events.*

The second aim is to understand the impacts of these storms on the stage-discharge relationships at gaged sites as a proxy for changes in flood hazard. However, this makes the assumptions that the storms alone are responsible for any observed changes in the residuals. While possible, other geomorphically significant events could have occurred that are unaccounted for.

***Response 3:*** *We thank the reviewer for this comment. Indeed, channel changes can be due to other geographically significant events (e.g. landslides, debris flow etc), however, such occurrences could also be triggered by the storm events that caused the flood hazards. At this stage, we have a complete database of storm properties, but we did not include a complete analysis of additional parameters such as mass movements as this would require a large dataset regarding this point which is not available consistently for all the considered watersheds. Please note that the model itself, however, considers other parameters, other than just the storm properties. The results highlight how geomorphological, hydrological, and Atmospheric properties also are responsible for the variability in residuals. We provided a comment about this in the revised paper.*

Further, the authors include other metrics in addition to the storms for predicting residuals, which makes it difficult to separate the impact of other drivers from the storms.

***Response 4:*** *We thank the reviewer for this comment. We believe that adding the other variables gives a better context to the impact of the storms. Overall, we have done a feature importance analysis and selected only those drivers that are the most important and influential. The feature importance itself, highlighted which variables mattered the most in predicting the residuals. We decided to use hydrologic and geomorphological variables because landscape properties are also linked to the potential effects of storms, as also highlighted in the previous comment referring to mass movements.*

*Minor comments*

Introduction

the short paragraphs appear and read choppy. Consider combining paragraphs where subject matter allows.

***Response 5:*** *We revised the text improving its readability.*

In the introduction, the authors imply that "extreme" storms or events are predominantly responsible for abrupt shifts in channel capacity and thus flood hazards. It is important to recognize that extreme storms/events likely contribute significantly to the population of abrupt shifts in channel capacity. However, there might be more frequent events that contribute to these changes as well, particularly depending on channel response potential (e.g., a sand bed river with high sediment supply and non-cohesive banks vs. a gravel bed river with heavily vegetated banks.) Thus, it is recommended to re-consider the use of "extreme" and apply more focus on "abrupt" channel changes to more accurately state the study objectives. For instance, it is not clear to what degree the population of storms included in the analysis is composed of "extreme" storms and what classifies those storms as extreme.

***Response 6:*** *Thank you for this recommendation. In the revised text we refrained from using severe or extreme, and we have used the term "Flood-inducing Storm Events", to be consistent with the work of Shen et al., 2017 where they refer to the dataset as a dataset of flood events. In the database, the authors also report the percentile of the peak flows in the entire time series of the watershed, and all the reported events show a value>80 for all storms. While the $80^{th}$ percentile might not be an extreme value itself, it is still a representation of less frequent events. Following the reviewer's comments, we rephrased the text referring to 'abrupt' changes, to more accurately frame the work.*

Line 102: How might this tool be used at ungagged sites without the detailed and rich dataset available? If applicable, it would be beneficial to highlight the use and importance of the tool in the conclusions.

*Response 7: We thank the reviewer for this comment. We believe that as USGS stream gage information could potentially be transferred from nearby stream gages if there is sufficient similarity between the gaged watersheds and the ungaged watersheds of interest, our model could also be applied to ungaged sites. However, one must always keep in mind that the successful 'translation' to ungaged environments depends on the correlation of the stream gages in the surrounding areas. For example, there are areas of CONUS (mostly mountainous) that show highly correlated stream gages (Kiang et al., 2013), whereas the Central United States and coastal areas of the Southeastern United States show significant number of uncorrelated gages. Therefore, the goodness of the information transfer might not work as well. Also, transferability would most likely be successful when basin attributes show high similarity and storm properties are within the range of variability of the training set used for this work. We added some consideration about this in the manuscript.*

Materials and Methods
The authors should acknowledge the bias of stream size representation and spatial density in the gaging network and how this might impact spatial interpretation of results. Some sizes and areas are vastly under- and over-represented.

*Response 8: Regarding the coverage of stream gages, we agree on the intrinsic limits of the dataset, based on the fact that there is variability across CONUS regarding the spatial and temporal coverage of stream gages. These limits in general have been addressed in literature and are very well summarized in the publication by Kiang et al., (2013).*

*In general, the Eastern United States has better coverage than the Western United States. The arid Southwestern United States, Alaska, and Hawaii were observed to have the poorest spatial coverage. Except in Hawaii, these areas also tended to have short streamflow records. Differences in hydrology lead to differences in the uncertainty of statistics calculated in different regions of the country. Arid and semiarid areas of the Central and Southwestern United States generally exhibited the highest levels of interannual variability in flow, leading to larger uncertainty in flow statistics.*
*We added considerations about this in the discussion of our results. In general, however, any research based on gaging sites faces the same challenges of over or underrepresentation. We highlighted this better in the limits and advantages of the proposed model.*

The method for computing likelihood of change ignores monotonic trends in decreasing capacity – increasingly negative residual. If the residuals become more and more negative, it indicates channel capacity is decreasing, but this is not accounted for by only counting shifts from positive to negative. This limitation should be acknowledged. To some degree, the reported method only accounts for oscillating shifts – positive residual to negative residual then positive residual to negative residual.

*Response 9: We thank the reviewer for this comment. Indeed, we focus mainly on abrupt shifts, rather than on permanent shifts. The main reasons for this were - 1. Short-term conveyance capacity changes are not considered in typical flood hazard assessments and could substantially overstate or understate flood threats at any particular time for subsequent floods; 2. there is a plethora of complex and*

*sometimes not linear- processes and coupled feedback that we would need to 'model' in the training set, to provide a comprehensive benchmark to identify permanent shifts vs sudden ones, and this could be a potentially interesting research that could be tackled by further studies building on our model, but at this stage it goes beyond the scope of this work.*
*We highlighted this point better in the manuscript.*

Results Analysis

Why did the authors choose to provide a results analysis section instead of organizing as results and discussion. The overall coherence and understanding of the results would be improved by breaking the results analysis section up into a results and discussion section.

**Response 10:** *We added a discussion section as per reviewer's suggestion*

Specific Comments

Line 30: It is not entirely clear what is meant by traditional "cause-effect" studies. I presume the authors are referring to changes in peak flows due to changes in causal mechanisms such as climate, land use, etc.

**Response 11:** *yes, this is correct.*

Line 32: How are might they over- or under-estimate actual damage, and what damage? Perhaps a follow-up example or additional explanation would clarify this sentence.

**Response 12:** *We revised this part of the manuscript.*

Line 34: This is, in effect, what fluvial geomorphology is, and this sentence is somewhat redundant with the rest of the paragraph.

**Response 13:** *We revised this part of the manuscript.*

Line 39: also critically modify the landscape and climate(???)

**Response 14:** *We revised the sentence for better clarity.*

Line 40: I am not sure flood risk is something that we measure more so than we estimate. Flood risk in fact can be highly uncertain Further, it is not only based on flood frequency, but the relationship between magnitude and frequency as is typically described by a distribution of peak flow, which are discretized as either annual maxima or peaks over threshold. Not just based on flood frequency.

**Response 15:** *Yes, this is correct. We revised the sentence.*

*Nonetheless, flood risk measurement has traditionally been based on flood frequency, derived from variability in streamflow, assuming constant channel capacity (Merz et al., 2012; Slater et al., 2015). The relationship between magnitude and frequency is also generally built upon the peak flow distribution, whereas peaks are discretized as either annual maxima or peaks over threshold, but mostly assuming that river capacity remains constant over the investigation records.*

Line 41 - 43: Recent works have employed methods that incorporate changing channel capacity:
- Stephens, T. A., & Bledsoe, B. P. (2023). Flood Protection Reliability: The Impact of Uncertainty and Nonstationarity. *Water Resources Research*, *59*(2), e2021WR031921.
- Stephens, T. A., & Bledsoe, B. P. (2020). Probabilistic mapping of flood hazards: Depicting uncertainty in streamflow, land use, and geomorphic adjustment. *Anthropocene*, *29*, 100231.

*Response 16: Thank you for the references. We added these to the manuscript and rephrased the text.*

Line 44: This is poor wording, the amount of water that flows through the river systems during floods could in fact change in some situations. Revise to a more correct sentence or consider removing the first portion.
*Response 17: We revised this part of the manuscript.*

Line 47: I presume by the use of frequency, the authors are describing the discharge magnitude of the flood. Instead of frequency, consider revising to magnitude, flow, or discharge since they are referring to the size and not how often it floods during a single event.
*Response 18: We revised this part of the manuscript.*

Line 49: please give an example of some flood properties.
*Response 19: We revised it.*

Line 54: magnitude, frequency, and risk.
*Response 20: We revised it.*

Line 55: Do the changes have to be rapid? What about long term trends that are not accounted for. Consider shifts in the mean vs. monotonic trends. Sometime flood hazard maps are not updated for a decade or more, beckoning a definition of rapid in this context.
*Response 21: For this work, we investigated sudden changes of positive to negative residuals. We acknowledge that these might not be permanent changes. Given the complexity of processes involved in the 'restoration' of river forms, or the permanence of a channel shift, we decided to focus this work on the sudden changes. With this idea, with the proposed method we highlight rivers more prone to changes in the aftermath of a storm, highlighting potential increased flood hazard in the case of subsequent storms.*

*In literature, using Slater's concept, the work by Ahrendt et al., 2022 offers an overview of historic long-term and short-term conveyance changes for WA, whereas the work of Li et al 2020 highlighted how relatively modest long-term changes in river channel capacity are composed of numerous short-term transients which are of much larger magnitude.*

*We referred to this work in our manuscript and added some considerations on the fact that this work only considers sudden changes but not their persistence in time.*

Line 58: are the trends in stage or erosion/deposition or both comparable to trends in peak streamflow?
*Response 22: Other works in literature highlighted that some channel changes could provoke a higher change in flood hazard than shifts in discharge alone (Slater et al. 2015, 2016, Li et al. 2020, Ahrendt et al 2022). For this work, we did not assess changes in streamflow, as we are training the model based on storm properties, and not on long-term discharge properties.*

Figure 1 would benefit from a scale bar.
*Response 23: We added the scale bar.*

Line 70: How do we know these are "sharp", and how do we know the revisions are "upward"? Couldn't they be downward if erosion occurred?
*Response 24: Indeed, the changes could be downward if erosion occurred. We imply that an upward revision is a proxy for an increase in flood hazard, whereas a downward revision potentially could*

*mean a reduced hazard. Our analysis is consistent with other works in the literature relating shifts in the stage-discharge relationship as a proxy for flood hazard.*

Line 95: "Despite some limitations" is used to start the previous sentence. Consider removing from one of the sentences. This sentence would read more formally by re-writing to remove the words "we" and "us".
**Response 25:** *We revised this part of the manuscript.*

Line 148: please define gaps in the measurements. The manual field measurements may follow irregular frequency. Therefore, what constituted a gap? Minor gaps or missing data in the regular stage-flow measurements by the gage may not have a substantial impact on the analysis.
**Response 26:** *We have excluded the gages that do not have continuous data for the tie frame from 2002-2013.*
*We clarified this more in the manuscript. For the work, aside from considering consistent gages present in the Shen et al. 2017 Database, and covered by stream measurements, we applied the same criteria as Slater et al. (2015), who only considered field measurements in which the discharge is within one percent of the product of channel velocity and cross-sectional channel area, as reported by the USGS, and those made close to the gage station.*

Line 155: stage, water level, or water surface elevation is more clear than "levels"
**Response 27:** *We revised this.*

Figure 3a would be improved by indicating the flood stage. Near a stage of 2m, there is not much difference in the pre and post 2007 measurements. Is this due to overbank flow?
**Response 28:** *The figure was shown as an example of shifts present in the measurement data. For this gage, the flood stage is at 10ft, and the peak discharge of the 2007 event was 11.51 ft, and the Quinnipiac River itself (at the gage right upstream of the one in the picture) measured the maximum discharges for the period of record of the station during the 2007 flood. Aside from the information provided by USGS on that event, we do not have direct knowledge of the event itself so we cannot make a precise statement on the reason behind the similarities highlighted by the reviewer.*

Figure 3: "In (b), some outlier residuals are evident, likely due to shifts in measurement locations. These points were filtered out before performing the ML training." Belongs in the text rather than the figure caption.
**Response 29:** *We moved this part to the text.*

Figure 3 c and d caption: Is it in fact channel area and width or wetted area and width? The use of channel over wetted mean two different things. The wetted area and width can change for a single channel geometry. Please clarify at line 160 as well.
**Response 30:** *The figure reports the channel width as reported in the gage measurements. We clarified this in the manuscript.*

Figure 3d: Should the y-axis label and caption be area or volumetric rate? Contradicts what is reported at lines 160 – 162. For area use area. For capacity use flow rate. Please clarify.
**Response 31:** *There was a mistake in the text, the line should have read 'and channel conveyance (Figure. 3d).' We rephrased this in the text.*

Figure 3: Please note that Figure 3c and possibly 3d (depending on capacity or area) could fluctuate

due to differences in measurement location, which can vary substantially from measurement to measurement. Even if measurement locations are close in distance, they may be upstream or downstream of a bridge. These factors must be considered when comparing widths to evaluate changes in the channel.

*Response 32: According to the information of the gage, the measurements did not shift in location. For the work itself, consistently with Slater et al. (2015) and the open codes provided in her work, we removed all field measurements made in a location where there is known infrastructure like a bridge for example, and all field measurements made in icy conditions, as these might affect measurements of channel geometry. We highlighted this more clearly in the manuscript.*

Line 178: does a frequency of 520 events at a gage disqualify them as "extreme"? This seems like a high frequency.

*Response 33: Indeed, the reviewer is correct. The magnitude of these events varied in time, and in the revised manuscript we used "flood-inducing event".*

Line 181: The authors might improve clarity by explicitly stating each gage measurement contained 5 different median storm characteristics – 1 median storm characteristics for the five different lag times considered. If I am interpreting this correctly.

*Response 34: Yes, this is correct and we revised the text accordingly.*

I understand it would be difficult to graphically convey this in the paper, but I am wondering if the authors investigated the sensitivity of median storm characteristics to lag time. I wonder how much difference there is here. It is not essential, but if available, a note on this would be interesting.

*Response 35: Thank you for this comment. We investigated this point and checked the variability of the properties as compared to lag time. However, as highlighted by the referee, conveying this analysis in the manuscript would be very difficult, as we have many variables for each gages and they vary for each channel measurement, and each lag time. Given the comment, we highlighted how this could be possible independent research stemming from our work.*

Table 1 would be more easily viewed in landscape layout and perhaps broken into 3 different tables. One table for each variable classification (geomorphic, hydrologic, and atmospheric).

*Response 36: Thank you for the suggestion. Please note that we have separated the variables in the same table, this should improve the quality of the table itself. The table can be set to landscape in the production phase also depending on how the journal arranges the work, if it is accepted for publication*

Table 1: Should the RFACT (Rainfall runoff factor) be classified as hydrologic instead of geomorphic?

*Response 37: We corrected this in the revised paper.*

Line 196: As per previous comments, how do we know they are "severe"? Do the median characteristics reflect this?

*Response 38: We thank the reviewer for the comment. In the revised text we refrained from using severe. Please note that the storm properties were taken from a published paper (Shen et al. 2017) classifying the storms as 'flood-inducing' properties. In the database, the authors also report the percentile of the peak flows in the entire time series of the watershed, and all the reported events show a value>80 for all storms. The median characteristic per se is not a 'severe' value, nonetheless,*

*it would be representative of the typical storm characteristics for storms which in general encompass events having peak flows >80$^{th}$ percentile. We added this clarification to the manuscript.*

Line 326: comparing the predicted residual with the average residual - Why was this done? Was it for validation?

***Response 39****: No. The idea was to compare the predicted residual with the most recent characteristic for the watershed: this provided an idea of how critical the change would be. A watershed having overall positive residuals for the most recent measurements, for which we predict a sudden shift to negative outside the confidence bound of the stage-discharge curve, represents a critical condition that should be monitored, as the current flood stage might underestimate the flood risk. We clarified this in the manuscript.*

Line 332: Some change is neglected in this computation: negative to positive, positive to more positive, and negative to more negative. Therefore, this sentence is somewhat inaccurate.
***Response 40****: Yes, this is correct and we revised it.*

Line 332 to 335: How was the confidence interval for the stage-discharge relationship computed?
***Response 41****: We added this part in the manuscript. As LOESS smoothers fit a unique linear regression for every data point by including nearby data points to estimate the slope and intercept, the correlation in nearby data points helps ensure obtaining a smooth curve fit. Therefore, the µ+1.96σ of the nearby data points considered for each fitted value can be considered as a measure of the 95% confidence interval. This information is calculated directly from the R package fANCOVA (https://CRAN.R-project.org/package=fANCOVA) used for the fitting.*

Line 344: Does this show the importance of geomorphology of the watersheds or bias in the number of variables selected to represent each variable class? In this interpretation, the authors have neglected the fact that there are different numbers of variable classes. Simply the inclusion of more in once class than the other does not directly translate to its importance in this case. The following sentence does fit the authors interpretation.
***Response 42****: We thank the reviewer for the comment. Please note that permutation feature importance serves as a method for examining the impact of individual features on the statistical performance of the model itself: by disrupting the association between a feature and the target variable, we can ascertain the extent to which the model depends on that particular feature. This technique has been recognized in the literature (e.g., (Breiman, 2016; Wei et al., 2015; Fisher et al., 2018) and it is widely implemented in many statistic packages as well (e.g., Biecek et al., 2018, 2019; Molnar & Schratz, 2008). Considering the reviewer's comment. We revised the text. As our dataset encompasses all CONUS, and keeping in mind the limitation pertains to gage properties, as we highlighted in the revised manuscript, amongst all the variables, only some mattered in allowing a satisfactory performance of the model, whereas others did not impact the results too much.*

Line 351: drainage density is correlated with other variables as well, such as precipitation.
***Response 43:*** *Yes, this is correct, and we added a reference related to this.*

Line 360: There is no evidence that flow regulation structures are the cause for these findings. It might suggest it if hydro_disturb_index only reflects flow regulation structures, but it could also include urbanization.
***Response 44:*** *T**he reviewer is correct; we realized the text could be misleading and reworded the*

*paragraph.*

*Our model highlighted in Figure 6, that the most important hydrologic variable was the condition of the watershed, whether it is anthropogenically modified or natural. This confirms that human modifications are an important element to be considered when analyzing flood hazard changes (Bormann et al., 2011; Pinter et al., 2006a, b). Ahrendt et al. (2022) demonstrated that channel regulation is important to conveyance changes which resonates with the variable importance analysis results from Figure 6. Similarly, the construction of dikes, bridges, dams, meander cutoffs, channel constriction by wing dikes, groynes, and other engineering projects can alter channel conveyance within rivers and the characteristics of their floodplains (Bormann et al., 2011; Pinter et al., 2006b, a). The importance of this variable in the model highlighted the potential interaction of flood-inducing events that generate high sediment deposition with the effects of channel modification. As well numerous works in literature (Feng et al., 2021; Mazzoleni et al., 2022) also highlighted how urbanization processes and landscape changes induced by human activities have large impacts on flood hazards worldwide.*

Table 2: The caption should state what the Corr and RMSE compare.
**Response 45:** *We revised this.*

Line 374: It would be helpful to know something about the distribution of residuals to provide context to the RMSE magnitudes.
**Response 46:** *The residual variability is quite high, as the values range from –3 to +3 overall, for the measurements retrieved.*

Line 391: Is it the spatial "spread" or spatial "patterns"?
**Response 47:** *We revised this.*

Line 418: Low flows are more of a hydrologic property rather than a morphodynamic property.
**Response 48:** *Thank you for this comment. Low flows are mainly referring to bankfull discharge which is a morphodynamic property. We clarified this in the manuscript.*

Line 450: Vulnerability was not defined, quantified, or reported anywhere prior to this.
**Response 48:** *We revised this.*

Line 473: In this sentence, it is not clear how the FHF increased logarithmically. How do we know this from the data presented?
**Response 49:** *We were referring to Slater's work for this particular comment. We revised this to clarify.*

Line 481: Directly comparing regions does not account for spatial correlation or representation bias in the gaging network. Some areas/regions and streams are more represented than others making a comparison between regions misleading.
**Response 50:** *We added a comment in the limitations section.*

*Comments on line 494 to 515.*
**Response 51:** *We revised the text, adding literature and references as suggested.*

Line 545 – 555: the connections between the centroid of perception, flash floods, and residuals is not made clear here. The centroid of precipitation is an important variable in the analysis for predicting residuals. How does this tie to flash floods? Please explain more clearly.

**Response 52:** *We thank the reviewer for this comment. We have added the following discussion to*

*the manuscript-*

*Many papers in the literature (e.g., (Borga et al., 2008; Woods and Sivapalan, 1999; Woods, 1999; Smith et al., 2004, 2005, 2002; Zhang et al., 2001) highlighted the relationship between the centroid of precipitation and runoff production. Most works showed that, for example, the position of the storm centroid relative to the watershed outlet is an important driver of runoff: storms having a precipitation centroid positioned in the central portion of the watershed tend to produce a higher runoff than storms having a centroid near the outlet or the head of the watershed. This is in line with the fact that rainfall runoff spatial variability influences flash flood severity relative to basin physiography and climatology. Flash flood severity, or flashiness, as defined by Saharia et al., (2017), assesses a basin's capacity to produce severe floods by considering both the volume and timing of a flood. It is, therefore, not unexpected that the centroid of precipitation appears to be highly correlated with the shifts in residuals.*

Line 575 – 580: this not necessarily true. Just because the channel conveyance capacity is exceeded does not mean the channel is expected to change. The flood must result in a geomorphically significant conditions of hydraulic and sediment supply conditions. The authors mentioned previously the importance of hysteresis in sediment deposition. This sentence over- simplifies and incorrectly categorizes a complex, situationally unique, and nuanced process.

**Response 53**: *We have revised the sentence and rephrased this paragraph.*
*When the volume surpasses the channel's conveyance capacity, flooding is anticipated, and if substantial sediment movement happens, there is potential for channel adjustments. The significance of these properties is a reaffirmation of the established notion that regular flows, such as baseflow below bankfull levels, are sufficient to determine channel shape, as they prevent the substantial accumulation of fine sediments and organic matter (Phillips, 2002). On the other hand, rare extreme floods are also essential for transporting coarser bed material and eroding channel banks (Phillips, 2002), promoting changes.*

Line 599: The use of "future" here is misleading since the authors explicitly evaluate short-term or abrupt shifts. The temporal persistence of that shift is not addressed.

**Response 53**: *We thank the reviewer for this comment. We revised this in the manuscript and clarified the temporal persistence issue.*

Line 600: As it stands, specific impacts from individual drivers is insufficiently addressed. A more accurate representation of the analysis would be to say that the method identified important drivers for predicting residuals from the average stage discharge curve. From my understanding, the analysis does not necessarily reveal the actual impact of specific variables on the predicted residual.
**Response 54**: *We thank the reviewer for this comment, and we rephrased the text.*

Line 608: Did the authors mean to say channel capacity here instead of "river discharge"?
**Response 55:** *We rephrased the text. Our research reveals that the assumption of channel stationarity may result in either over or under-prediction of the river discharge for a certain flood stage, as the existing stage-discharge relationship might be temporarily (or permanently if the shift pertains) underperforming. This would in turn eventually over/under-estimation of flood hazard (recurrence interval, duration, depth, and inundation extent of flooding), especially in the case of subsequent floods.*

Line 615: More specifically, the risk of immediate reduction in channel capacity. The authors did not evaluate increases in channel capacity.

*Response 56*: *We revised it.*

Line 616: Knowing the temporal persistence of these changes would provide insight to the feasibility of these updates or alternative methods for quantifying flood risk if the process is highly variable in time.
*Response 57*: *We thank the reviewer for this comment, and we modified the text accordingly.*

Technical Corrections
*Response 58*: *We fixed all the technical corrections mentioned by the reviewer and revised the manuscript.*

**References:**
- *Kiang, J. E., Stewart, D. W., Archfield, S. A., Osborne, E. B., Eng, K., and Survey, U. S. G.: A national streamflow network gap analysis, Scientific Investigations Report, Reston, VA, https://doi.org/10.3133/sir20135013, 2013.*
- *Shen, X., Mei, Y., and Anagnostou, E. N.: A comprehensive database of flood events in the contiguous United States from 2002 to 2013, Bull Am Meteorol Soc, 98, 1493–1502, https://doi.org/10.1175/BAMS-D-16-0125.1, 2017.*
- *Slater, L. J., Singer, M. B., and Kirchner, J. W.: Hydrologic versus geomorphic drivers of trends in flood hazard, Geophys Res Lett, 42, 370–376, https://doi.org/10.1002/2014GL062482, 2015*
- *Merz, B., Vorogushyn, S., Uhlemann, S., Delgado, J., and Hundecha, Y.: HESS Opinions "More efforts and scientific rigour are needed to attribute trends in flood time series", Hydrol Earth Syst Sci, 16, 1379–1387, https://doi.org/10.5194/hess-16-1379-2012, 2012.*
- *Ahrendt, S., Horner-Devine, A. R., Collins, B. D., Morgan, J. A., and Istanbulluoglu, E.: Channel Conveyance Variability can Influence Flood Risk as Much as Streamflow Variability in Western Washington State, Water Resour Res, 58, e2021WR031890, https://doi.org/10.1029/2021WR031890, 2022.*
- *Li, Y., Wright, D. B., and Byrne, P. K.: The Influence of Tropical Cyclones on the Evolution of River Conveyance Capacity in Puerto Rico, Water Resour Res, 56, https://doi.org/10.1029/2020WR027971, 2020.*
- *Breiman, L.: RANDOM FORESTS, International Journal of Advanced Computer Science and Applications, 7, 1–33, https://doi.org/10.14569/ijacsa.2016.070603, 2016.*
- *Wei, P., Lu, Z., and Song, J.: Variable importance analysis: A comprehensive review, Reliab Eng Syst Saf, 142, 399–432, https://doi.org/10.1016/j.ress.2015.05.018, 2015.*
- *Fisher, A., Rudin, C., and Dominici, F.: All Models are Wrong, but Many are Useful: Learning a Variable's Importance by Studying an Entire Class of Prediction Models Simultaneously, Journal of Machine Learning Research, 20, 2018.*
- *Biecek, P., Baniecki, H., and Izdebski, A.: Effects and Importances of Model Ingredients, Journal of Machine Learning Research, 19, 2018.*
- *Biecek, P., Gosiewska, A., Baniecki, H., Izdebski, A., and Komosinski, D.: Model Agnostic Instance Level Variable Attributions, R Journal, 10, 395–409, https://doi.org/10.32614/RJ-2018-072, 2019.*
- *Molnar, C. and Schratz, P.: Interpretable Machine Learning, Annals of Applied Statistics, 2, 916–954, https://doi.org/10.1214/07-AOAS148, 2008.*
- *Bormann, H., Pinter, N., and Elfert, S.: Hydrological signatures of flood trends on German rivers: Flood frequencies, flood heights and specific stages, J Hydrol (Amst), 404, 50–66, https://doi.org/10.1016/J.JHYDROL.2011.04.019, 2011.*
- *Pinter, N., Van der Ploeg, R. R., Schweigert, P., and Hoefer, G.: Flood magnification on the River Rhine, Hydrol Process, 20, 147–164, https://doi.org/10.1002/hyp.5908, 2006a.*
- *Pinter, N., Ickes, B. S., Wlosinski, J. H., and van der Ploeg, R. R.: Trends in flood stages: Contrasting*

results from the Mississippi and Rhine River systems, J Hydrol (Amst), 331, 554–566, https://doi.org/10.1016/J.JHYDROL.2006.06.013, 2006b.

o Feng, B., Zhang, Y., and Bourke, R.: Urbanization impacts on flood risks based on urban growth data and coupled flood models, Natural Hazards, 106, 613–627, https://doi.org/10.1007/S11069-020-04480-0/TABLES/3, 2021.

o Mazzoleni, M., Dottori, F., Cloke, H. L., and Di Baldassarre, G.: Deciphering human influence on annual maximum flood extent at the global level, Commun Earth Environ, 3, https://doi.org/10.1038/s43247-022-00598-0, 2022.

o Borga, M., Gaume, E., Creutin, J. D., and Marchi, L.: Surveying flash floods: gauging the ungauged extremes, Hydrol Process, 22, 3883–3885, https://doi.org/10.1002/HYP.7111, 2008.

o Woods, R. and Sivapalan, M.: A synthesis of space-time variability in storm response: Rainfall, runoff generation, and routing, Water Resour Res, 35, 2469–2485, https://doi.org/10.1029/1999WR900014, 1999.

o Woods, R.: Rain • Distributed • Hillslope • Channel _• Q ( t ), 35, 2469–2485, 1999.

o Smith, J. A., Baeck, M. L., Meierdiercks, K. L., Nelson, P. A., Miller, A. J., and Holland, E. J.: Field studies of the storm event hydrologic response in an urbanizing watershed, Water Resour Res, 41, https://doi.org/10.1029/2004WR003712, 2005.

o Smith, J. A., Baeck, M. L., Morrison, J. E., Sturdevant-Rees, P., Turner-Gillespie, D. F., and Bates, P. D.: The regional hydrology of extreme floods in an urbanizing drainage basin, J Hydrometeorol, 3, 267–282, https://doi.org/10.1175/1525-7541(2002)003<0267:TRHOEF>2.0.CO;2, 2002.

o Smith, M. B., Koren, V. I., Zhang, Z., Reed, S. M., Pan, J. J., and Moreda, F.: Runoff response to spatial variability in precipitation: An analysis of observed data, J Hydrol (Amst), 298, 267–286, https://doi.org/10.1016/j.jhydrol.2004.03.039, 2004.

o Zhang, Y., Smith, J. A., and Baeck, M. L.: The hydrology and hydrometeorology of extreme floods in the Great Plains of Eastern Nebraska, Adv Water Resour, 24, 1037–1049, https://doi.org/10.1016/S0309-1708(01)00037-9, 2001.

o Saharia, M., Kirstetter, P. E., Vergara, H., Gourley, J. J., Hong, Y., and Giroud, M.: Mapping Flash Flood Severity in the United States, J Hydrometeorol, 18, 397–411, https://doi.org/10.1175/JHM-D-16-0082.1, 2017.

o Phillips, J. D.: Geomorphic impacts of flash flooding in a forested headwater basin, J Hydrol (Amst), 269, 236–250, https://doi.org/10.1016/S0022-1694(02)00280-9, 2002.

**Response to the review HESS_RC2**

*First, we want to thank the reviewer for the insightful detailed comments and recommendations. We are glad that you have acknowledged our work as relevant and novel. Following the suggestions, we revised the manuscript. Specific responses to the reviewers' comments are added below.*
***Note: Below is our response (italics) to each reviewer's comment (regular font)***

The proposed study is relevant and novel for the field. Thus, it should be worthy of publishing in this journal. However, there are some details that needs to be revised before accepting it for publishing. Below you will find a list of my main points.

The articles needs desperately a discussion section separated from the results. Right now, everything is cramped into a single section that makes difficult to take the main points out.

***Response:*** *We added and separated the discussion section.*

I would also add as a minimum a paragraph in the conclusion section (if not its own standalone section) about the framework limitations. This is crucial in any prediction framework so readers are aware of it when taking decision.

***Response:*** *We have a section in the discussions that discusses limitations related to the method. Also, as suggested by the reviewer we added limitations of the framework in the conclusions. In the revised manuscript we extended the discussion chapter, highlighting further strengths and limitation of the proposed model.*

The introduction (and other sections) have too many small paragraphs (1-3 sentences) that disrupts the reading of the manuscript. I will suggest to combine paragraphs that convey the same message.

***Response:*** *We revised the introduction as per the reviewer's suggestion. We tried to keep the flow of reading by grouping the same messages.*

Did the authors had a minimum years of data treshold when selecting the gauges? It seems imperative to have one, since a gauge with 5 years of data will yield very different results than one with 50 years of data. Also, what are the general statistics of the gauge data? For example, what is the mean length of record, amount of cross sections measurements, etc.

***Response:*** *For this work, for the gages data, we followed Slater's 2015 workflow and used the system they provided accompanying the published work. Overall, one must consider that for our model we focus on sudden changes in the stage-discharge, and not their persistence in time. We evaluate the stage-discharge relationship based on measured river properties, as by Slater's 2015 work. Following the referenced work, we detected and excluded sites featuring artificial controls at the gauging station that could impede the natural adjustment of the channel's shape. Additionally, we eliminated all field measurements conducted at a different location or potentially different location, along with those taken in icy conditions, as these factors could impact the accuracy of channel geometry measurements. Our selection process retained only sites with comprehensive time series data, and as per Slater's et al. 2015 work, only kept gages with 99.7% completeness in streamflow records and 40 channel cross-section measurements. We clarified this in the revised manuscript.*

Specific Comments:

Line 44:  is not clear the statement. Since during a flood event, flow within the channel can change due to external factors, stormwater discharge o compounding flood at coastal estuaries.

***Response:*** *The reviewer is correct, many other parameters could be the direct cause of change, also considering for example debris flows or mass movements. Nonetheless, these could also be triggered by the storm properties themselves. We added some comments on this in the revised manuscript.*

Line 64: the "secondary channel" that is referring in Figure 1 should be highlighted in the figure itself to help the reader understand the point.

***Response:*** *We revised the figure.*

Figure 1: The figure needs a north arrow and scale bar. I also strongly suggest the authors to use a GIS platform to enhance the quality of the figure. The figure also needs a location map.

***Response:*** *We added a north arrow and scale bar to the figure. We decided to mention the location in the caption rather than adding the location map.*

Line 93 and 95: both sentences start with "Despite some limitations …" Please rephrase.

***Response:*** *We rephrased the sentence.*

Line 106: I will summarize all the gauges selected with their corresponding ID in a text file (or any other format file)  and upload it to a repository for easy sharing. Then, the reader could see exactly which stations were selected. This helps the open data statement in the research community.

*Response: We uploaded the text file as a supplementary file.*

Line 110: did the authors downloaded also discharge values from the NWS or it was just flood stages as it is mentioned in this line? If the cross section data (width and depth of the river) were obtained from the USGS, why not also use their created flow-stage curves. My biggest question is from where the authors obtained the discharge values for the creation of their rating curves, since NWS only provides stage level whereas USGS provides both stage and discharge in most gauges.

*Response: For this work, we followed Slater's 2015 workflow, and used the system they provided to accompany the published work. We evaluated the stage-discharge relationship based on measured river values, as in Slater's 2015 work. The discharge data are actually from the measurements, not from the continuous dataset from NWIS. We clarified this in the manuscript.*

Figure 2: I do not support have several lines of text if the figure caption just to describe the different climate regions in the map. Also, the authors also explain the abbreviation in the results section when talking about it. Thus, I strongly recommend having a nomenclature section that summarizes all of these, including the variables from table 1. Then, the reader can easily find it.

*Response: We thank the reviewer for the comment. Please note that adding the climate regions as the legend itself would make the figure unreadable. We, however, believe that a reader who might not be familiar with the climate in our study domain can benefit from having the explanation in the caption. We have added the climate types and physio region description in the appendix in separate tables.*

*Table 1 provides a full description of all the used acronyms for the variables.*

Line 161: the statement of the reason for change in capacity (deposition) has been already mentioned in Line 159. Please rephrase or remove.

*Response: We revised this.*

Line 177: why that the authors only focused on a very narrow range of years for their storm event? This seems like a big limitation, especially since the latest year of the record was a decade ago. The authors needs to justify their selection as a minimum.

*Response: We want to thank the reviewer for this comment. We would like to draw the reviewer's attention to the fact that in our study we have used a published dataset (Shen et al. 2017) of storm events and the properties of the events. Most of them were calculated properties. This dataset was from 2002-2013. Our framework showcases the intercorrelation of the different event properties that can affect the channel changes and that can be of any timelines. Here we have established a framework that can be used for extended timelines. Researchers can use the trained model with additional years of data, if they have available the same storm properties proposed by Shen for more recent events.*

Table 1: there are some variables that have their "unit" column empty. For example, Peak , Q2, etc. This might be a typo since if the variable does not has unit the authors specify with a dimensionless or N/A. Also, the table is too long for a peer-review article. I strongly suggest dividing the table into three separate ones, one for each variable type. There are also some variables like BFI_AVE that their description is a quarter of the page due to being squeezed in the small column width. I would suggest

the authors to place the long variable description as a footer in the table or in the appendix as part of the nomenclature section.

*Response: We have revised the table as requested while keeping it consistent with the original datasets considered for the study*

***Reference:***

- *Shen, X., Mei, Y., and Anagnostou, E. N.: A comprehensive database of flood events in the contiguous United States from 2002 to 2013, Bull Am Meteorol Soc, 98, 1493–1502, https://doi.org/10.1175/BAMS-D-16-0125.1, 2017.*
- *Slater, L. J., Singer, M. B., and Kirchner, J. W.: Hydrologic versus geomorphic drivers of trends in flood hazard, Geophys Res Lett, 42, 370–376, https://doi.org/10.1002/2014GL062482, 2015*

**Response to the review HESS_RC3**

*First, we want to thank the reviewer for the insightful detailed comments and recommendations. We appreciate your outlook on the potential of our study.*
*Following the suggestions, we revised the manuscript accordingly. Below we provide responses detailing the revision actions we took to address the reviewers' comments.*
***Note: Below is our response (italics) to each reviewer's comment (regular font)***

This paper focuses on predicting changes in flood hazard, primarily driven by 'major storm events.' The study analyzed flood hazard changes for 3,101 gauges across the Continental United States (CONUS) using a machine learning model (self-organized maps) that incorporated 38 explanatory variables, including atmospheric, hydrologic, and geomorphologic factors. The findings are highly relevant and could serve as a valuable reference for understanding channel capacity in relation to storm events. However, I believe that the authors should enhance the manuscript's overall structure and clarify certain technical aspects. Therefore, I recommend major revisions. I am optimistic that by addressing these comments, the authors can enhance this already interesting work.

*We want to thank the reviewer for this comment. We appreciate your outlook on the potential of our study. We did our best to include in the revision all suggested changes.*

Major Comments:

The authors state in their paper objectives and title that they are exploring the impact of major storm events on flood hazard through changes in channel capacity. However, the explanatory variables used in their machine learning models include not only storm-related properties but also hydrologic and geomorphologic variables. It's unclear how the authors discern from the ML models that the changes in channel capacity are primarily attributed to storm properties and not influenced by other factors. Additionally, the analysis is focused on the dataset containing "major storm events," which implies that changes in channel capacity are associated in the ML to major events. What about changes in channel capacity during non-storm events? In other words, can the ML capture changes in channel capacity without storm property variables (variables as described by Falcone, 2011)?

*Response: we thank the reviewer for this comment. The ML model was trained considering both storm properties and watershed properties. We do not make distinction on which element triggers the change, nonetheless in the paper we provided an assessment of feature importance, highlighting that the shifts, for how the model works, are mostly explained by a combination of storm and watershed properties. We*

*would not suggest using the model, as it is trained currently, to predict changes without having information on the storm properties. We highlighted this in the revised manuscript.*

The overall organization of the introduction and methods sections lacks necessary details. To better understand the techniques used to estimate the residuals and the various simplifications (such as manual filtering of outliers) required for the methodology, I had to refer to Slater et al., (2015). For example, in Figure 3, panel b, the authors mention outliers but do not clearly identify them or provide specific details.The results and discussion sections need to be reorganized. It is recommended to create a separate section for the discussion. Additionally, for improved readability, it is advisable to create a single section dedicated to limitations and future work.

*Response: We thank the reviewer for this comment. We separated the discussion and results section. We have also added a section in the conclusion about the limitations of the framework and future scopes. Regarding Figure 3, we added the text as per the reviewer's suggestion. We also added more clarifications on the methodology we used in the work.*

Minor Comments:

Figure 3: It would be helpful to include a time series with streamflow data to illustrate the magnitude of the April 2007 flood. Panels c and d are confusing since they may give the impression that there's only one change in flood capacity per gauge, which might not be the case.

*Response: We added some clarification about this in the text.*

Consider adding a schematic figure that explains the core concept of conveyance capacity before and after a storm event. Real data examples would be beneficial in illustrating this concept. Slater et al., 2015, offers a useful example in this regard.

*Response: We thank the reviewer for this comment. We believe however that adding such an image would be redundant as the referenced works such as Slater et al. 2015 are open access. Please note however, that we did add information on the method, and made more clear references to the method in Slater et al.*

Figure 9: Please provide information on how the 95% confidence bound of the current stage-discharge relationship was calculated.

*Response: We thank the reviewer for this comment. We clarified this in the manuscript. As LOESS smoothers fit a unique linear regression for every data point by including nearby data points to estimate the slope and intercept, the correlation in nearby data points helps ensure obtaining a smooth curve fit. Therefore, the $\mu+1.96\sigma$ of the nearby data points considered for each fitted value can be considered as a measure of the 95% confidence interval. This information is calculated directly from the R package fANCOVA ([https://CRAN.R-project.org/package=fANCOVA](https://CRAN.R-project.org/package=fANCOVA)) used for the fitting.*

*Reference:*

o *Shen, X., Mei, Y., and Anagnostou, E. N.: A comprehensive database of flood events in the contiguous United States from 2002 to 2013, Bull Am Meteorol Soc, 98, 1493–1502, https://doi.org/10.1175/BAMS-D-16-0125.1, 2017.*

o *Slater, L. J., Singer, M. B., and Kirchner, J. W.: Hydrologic versus geomorphic drivers of trends in flood hazard, Geophys Res Lett, 42, 370–376, https://doi.org/10.1002/2014GL062482, 2015*

**Response to online community comment by Dr Gesch**

*First, we want to thank Dr Gesch for the insightful suggestions and recommendations. We provided a full review of the manuscript, and incorporated all raised suggestions. We addressed all the minor points raised in terms of errors and typos and added all the suggested references. Please consider that the manuscript was toroughly rephrased, so some of the raised suggestions might have not been directly addressed as the text was entirely removed or rephrased.*

*Below please find some detailed response (italics) to the raised comments (regular font)*

What was the magnitude (i.e., percent) of excluded measurements? Were any of the 3,101 gages excluded completely, and if so, how many?

**Response**: *we thank Dr Gesch for this comment. Following the referenced work, for the analysis we also detected and excluded sites featuring artificial controls at the gauging station that could impede the natural adjustment of the channel's shape. Additionally, we eliminated all field measurements conducted at a different location or potentially different location, along with those taken in icy conditions, as these factors could impact the accuracy of channel geometry measurements. Our selection process retained only sites with comprehensive time series data, and as per Slater's et al. 2015 work, only kept gages with 99.7% completeness in streamflow records and 40 channel cross-section measurements. The revised manuscript provides a more detailed discussion on the procedure used to collect the gage data. We also added a list of the considered gages as supplementary material.*

Because STREAMS_KM_SQ_KM comes from 100k NHD, are there any concerns about "artificial" variability of stream density of cartographically-derived streams?

Response: *Indeed this is a critical point. Actual stream density could be different than that from cartography for many reasons. Nonetheless, to avoid having biases and further fluctuations of values, we decided to consider this official source as it is available for all gages from a reasonably consistent source. Researchers could also consider using different methods to define the drainage network, for example exploting the advantage of lidar and authomatic extraction techniques, but we would suggest, in that case, to re-train the model and verify once again the importance of this parameter in the re-trained model. In the revised manuscript, we added this critical point in the discussion of the ML method, and provided suggestion to include other datasets in our model.*

What about channelized streams in urban areas (where the channel cannot change in response to extreme events)? Would the method used in this study recognize these cases?

Response: *This is a critical point. We excluded sites featuring artificial controls at the gauging station that could impede the natural adjustment of the channel's shape. We would not recommend this approach for engineered river reached where flood protection measures or artificial channelization is present. We added this consideration in the ML limitation chapter.*

---

## Referee Report (RR1)

**Major Comments**

A separate discussion section is now included in the manuscript. However, the authors have unnecessarily complicated the section by adding subsections and being lengthy. I will try to summarize this section since most of the text has been repeated throughout the manuscript. For example, the text on Line 719-720 should be in the results sections since it describes what is being seeing in Figure 8. I assume this happen when the authors just move the exact same text from the results sections, instead of creating a new discussion section from they findings interpretation. Furthermore, the discussion section should not present any new figures like Figure 10, as this is done in the results section. Similarly, I will make the limitation subsection its own section (instead of a subsection within the discussion) since it has enough length and has been drafted properly. Thus, I urge the authors to review and rewrite the discussion section to be efficient.

While the author improves the introduction section, there are still some portions that read choppy, mainly because the authors have small paragraphs (2-3 sentences) with their own topic but failed to connect them properly (L92-112). Thus, I ask the authors to revise the later part of the introduction.

Overall, most of my comments/questions from the revised version were answer by referencing Slater's 2015 workflow. Thus, meaning that my question could be answer in that publication. This represents that the authors need to include more information about Slater's work in this manuscript so the reader does not have to be back and forward between this manuscript and Slater.

**Minor Comments**

- L43: combine that small paragraph (two sentences) with the previous one at L36.
- L776: The section index should be 4.2 and not 4.3.
- L855-860: If you are focusing in the amount of stream gauges within the east and west region of CONUS do not starting comparing with Alaska and Hawaii since they are not part of CONUS. If you do, then you need to talk about Puerto Rico and American Samoa (both US territories).
- Figure 1: Please add a description of the arrow and red circle on the figure caption.

---

## Referee Report (RR2)

Issues remain between the study's aims, methods, and conclusions. This type of dataset is not appropriate to "map the spatial variability" (line 100 -101). As stated in previous comments, the gaging network does not provide a representative sample to map the spatial variability. The authors have added some discussion acknowledging some of these limitations. However, it does not remedy the disconnect between the aim and methodology. For instance, smaller streams are vastly under-represented; and some individual larger rivers contain larger portions of gages than those surrounding them. These rivers and streams comprise a population of geomorphic characteristics, that if biased in representation, impacts the results of spatial variability in geomorphic response. While mapping the predicted residuals at streams is valuable and inferences can be made by comparing results across different regions, the authors should reconsider the information provided by the methods relative to the study's aims. The authors fail to mention spatial patterns in the conclusions, except perhaps, a slight alluding to at line 695.

The introduction should be more specific about the type of response considered in this analysis: abrupt loss of channel capacity following flood inducing storm events. This will provide context and clarity.

Minor Comments:

- Line 57 remove parenthesis around citation.
- Line 131. Revise sentence by removing "and" and capitalizing first word in sentence.
- Line 197: Greater than "the" $80^{th}$ percentile. And $80^{th}$ percentile of what? The annual flow duration curve?
- Line 372: To me, the relative magnitude of the RMSE is unclear. The authors state that it was close to 0, which is good. But the values range from 0.09 – 0.14 m. What would be considered unacceptable? While zero is the target; I do not have a frame of reference to know the relative magnitude of this error. This is important, if I understand correctly, because it influences confidence in the predicted residuals. What confidence can we have in the predicted residuals?
- Line 605: Remove the text "In the revised manuscript, we will incorporate these comments."
- Line 609: I am unsure about translating the model to gaged sites. In my previous comment, I was referring to a limitation in data availability. Wouldn't the model require a stage-discharge curve to relate the residual to? I a stage-discharge curve would not be determined from gage extrapolation.
- Line 619: Was the ML model trained using geomorphic properties as well?

---

## Author Response (AR2)

**Review of egusphere-2023-1969**

First of all we want to thank the reviewers for their further recommendations on our work. We did our best to incorporate all the proposed suggestions. Below our detailed response to each comment.

**egusphere-2023-1969-referee-report-3**

Issues remain between the study's aims, methods, and conclusions. Specifically, the second aim of the paper is not supported by the methods. Representative bias in the gage network prevents adequately "mapping the spatial variability". However, Figure 10 and the associated discussion provide valuable insight. Rather, consider re-evaluating the information provided by the methods relative to the study's aims. For this reason and other minor edits, revisions are required prior to publication. Please see the attached document for more detailed and additional comments.

Issues remain between the study's aims, methods, and conclusions. This type of dataset is not appropriate to "map the spatial variability" (line 100 -101). As stated in previous comments, the gaging network does not provide a representative sample to map the spatial variability. The authors have added some discussion acknowledging some of these limitations. However, it does not remedy the disconnect between the aim and methodology. For instance, smaller streams are vastly under-represented; and some individual larger rivers contain larger portions of gages than those surrounding them. These rivers and streams comprise a population of geomorphic characteristics, that if biased in representation, impacts the results of spatial variability in geomorphic response.

*Response: We want to thank the reviewer for this comment. We agree with the limitations of the gaging sites' distribution as we highlighted in the manuscript. Please consider that the 3101 gages in this study provide a vast dataset covering varying landscapes and climate zones. For our model, we added further information for the clustering, pertaining to storm properties and landscape properties, as well as watershed characteristics (e.g. Stream density, Strahler stream order). At this stage, the Gage dataset is the best available dataset we can get our hands on target shifts in stage discharge, by accepting the limitations on its wide but 'constrained' variability.*

*Considering the concerns regarding the disconnect between the aim and the methodology, we have revised the manuscript, also following the other reviewer's suggestions. As a consequence, we rephrased and added the following paragraphs.*

*Line 87-111:* "*For this purpose, the availability of a large dataset representing a wide range of flood-inducing storm characteristics and channel morphology under different boundary conditions, such as underlying climatic, hydrologic, and geomorphologic settings, is crucial. This set of information forms a complex interacting system. The processes underlining these boundary conditions vary in spatial and temporal scale, and this calls for the use of improved analysis methods, able to draw predictions interlocking data of varying nature. In this context, machine learning (ML) is gaining popularity in the field of hydrology, geomorphology, and climate studies (Bergen et al., 2019; Schlef et al., 2019; Valentine and Kalnins, 2016), thanks to its ability to tackle coupled processes across space and time. Despite some limitations (Karpatne et al., 2019), and provided that the benchmark data used for the training are of high quality (Bergen et al. 2019), ML offers a valuable tool to gain new data-driven insights with high accuracy, transferability, and scalability (Houser et al., 2022; Sarker, 2021; Schlef et al., 2019; Sofia, 2020a).*

*In the context of river morphology, specifically, in the last few years recent studies highlighted their capability to predict channel types (Guillon et al., 2020), providing a geomorphological characterization of*

*channels (Rabanaque et al., 2022), quantifying below-water (Woodget et al., 2019), or spatiotemporal changes (Boothroyd et al., 2021) in rivers, and guiding discharge estimation building from river morphology (Brinkerhoff et al., 2020), These works highlight how ML, when properly guided by field-based interpretation, can offer a valuable potential to push geomorphology into an increasingly predictive science (Fryirs and Brierley, 2022; Brierley et al., 2021). Tackling on the opportunities offered by ML potential, in this study, we sought to understand and predict the effects of flood-inducing storms on channel conveyance and, consequently, flood hazards. To achieve this, we have utilized stage-discharge "Residual" as a proxy of the channel capacity change, and we introduced an ML framework (section 2.3) that characterizes the interdependence of flood drivers, including atmospheric drivers (precipitation), hydrologic drivers (flow, stage), and geomorphologic drivers (channel width, depth, drainage area, geophysical characteristics). Overall, the analysis aims to: (1) highlight the variability of geomorphic response to flood-inducing storms across various climatic and geomorphologic regions in the Contiguous US (CONUS), and (2) understand the impact of these storms on the stage-discharge relationships at gaged sites as a proxy for changes in flood hazard. The study provided an independent test of discharge-based results and produced a tool for generating timely short-term updates of flood hazard estimates for dynamic rivers.*

While mapping the predicted residuals at streams is valuable and inferences can be made by comparing results across different regions, the authors should reconsider the information provided by the methods relative to the study's aims. The authors fail to mention spatial patterns in the conclusions, except perhaps, a slight alluding to at line 695.

**Response:** *We thank the reviewer for this comment. We have added the following to the conclusion-*

**Line 696-699:** *The gages used in the study although distributed across CONUS have intrinsic limitations in terms of stream size representations and spatial coverage of the river network. Therefore, careful considerations should be applied while considering the model for predicting the impact of flood-inducing storms on abrupt loss of channel capacity outside the basins used in our study. As for ML in general, successful translation is expected given that new sites under investigation possess similarities to the training sites.*

The introduction should be more specific about the type of response considered in this analysis: abrupt loss of channel capacity following flood inducing storm events. This will provide context and clarity.

**Response:** *Thank you for this comment. We have mentioned "abrupt loss of channel capacity" not only in the introduction but in the conclusions too to be consistent and specific about this.*

Minor Comments:

- Line 57 remove parenthesis around citation.

**Response:** *We have corrected this.*

- Line 131. Revise sentence by removing "and" and capitalizing first word in sentence.

**Response:** *We have corrected this.*

- Line 197: Greater than "the" 80th percentile. And 80th percentile of what? The annual flow duration curve?

**Response:** *We have corrected this. Please see below:*

*Line 201-2-3: The reader should consider that while the median characteristic per se is not a 'severe' value, given the sample of data in Shen et al., 2017, it is a value representative of the typical event, for*

*storms which in general encompass events having peak flows greater than 80th percentile of the entire flow series.*

- Line 372: To me, the relative magnitude of the RMSE is unclear. The authors state that it was close to 0, which is good. But the values range from 0.09 – 0.14 m. What would be considered unacceptable? While zero is the target; I do not have a frame of reference to know the relative magnitude of this error. This is important, if I understand correctly, because it influences confidence in the predicted residuals. What confidence can we have in the predicted residuals?

*Response: We thank the reviewer for this comment, and we agree with this. We have revised the text. Please see below-*

*Line 376-380: Table 2 also represents the correlation distance and RMSE between the measured and predicted residuals for each cluster of the validation datasets. The average correlation was close to 1 for all N values, suggesting the performance of the SOM model was satisfactory. The average RMSE was in the range of 0.09 – 0.14 m, which indicates a low random error relative to the dynamic range (-3 to 3) of the predicted variable. Both the unsupervised correlation distances and the average correlation showed the best results for N- 365 days. The RMSE diminished with the increase in the interval.*

- Line 605: Remove the text "In the revised manuscript, we will incorporate these comments."

*Response: We have corrected this.*

- Line 609: I am unsure about translating the model to gaged sites. In my previous comment, I was referring to a limitation in data availability. Wouldn't the model require a stage-discharge curve to relate the residual to? I a stage-discharge curve would not be determined from gage extrapolation.

*Response: We thank the reviewer for this comment. We added the following lines to the discussion*

*"The model predicts a shift in the discharge at the flood stage (residuals), as a proxy for flood hazard changes, implying that a certain discharge, expected to produce floodings, will be reached for lower stages than expected (residual shifting from positive to negative, at a specific gage). The approach starts from the concept that, typically, discharge time series are derived from water level measurements through an existing stage-discharge relationship. This is the general case for most gaging sites in the US, as well as other realities in other countries. As rating changes often happen during episodic storms, the proposed model can be adapted for other gage datasets, in different parts of the world, by assuming the operational existence of a similar approach". (Line 692-699)*

- Line 619: Was the ML model trained using geomorphic properties as well?

We considered the watershed properties included in the GAGE II dataset, such as slope and aspect, drainage properties, geological information, and information on the physiographic region. After the variable importance, only some of the parameters were retained as reported in Figure 6.

**egusphere-2023-1969-referee-report-2**

The authors have addressed all of my previous comments. For future review responses, I strongly recommend that the authors clearly specify in the response document the locations (e.g., line numbers) in the manuscript where changes have been made, to streamline the review process.

*Response: We want to thank the reviewer for the time and effort. We apologize for any inconvenience caused by the way we produced our response. We appreciate the suggestions and will keep this in mind for future review processes.*

**egusphere-2023-1969-referee-report-1**

The revised version of the manuscript still needs additional work in the format of the introduction and discussion sections.

**Major Comments**

A separate discussion section is now included in the manuscript. However, the authors have unnecessarily complicated the section by adding subsections and being lengthy. I will try to summarize this section since most of the text has been repeated throughout the manuscript. For example, the text on Line 719-720 should be in the results sections since it describes what is being seeing in Figure 8. I assume this happen when the authors just move the exact same text from the results sections, instead of creating a new discussion section from they findings interpretation. Furthermore, the discussion section should not present any new figures like Figure 10, as this is done in the results section.

Similarly, I will make the limitation subsection its own section (instead of a subsection within the discussion) since it has enough length and has been drafted properly. Thus, I urge the authors to review and rewrite the discussion section to be efficient.

*Response: We want to thank the reviewer for this comment. As per previous comments stating the results section is too long, we have created a discussion section. Yes, we have taken some of the inferences from the previous results section so that the readers can relate the discussion to the figures. Considering the reviewer's comment, we rephrased this part, to be more consistent in the discussion. Please note that we prefer to keep Figures 10 and 11 in the discussion because we believe that these figures strengthen the discussions rather than be a result. As per the reviewer's suggestion, we have created a section for the "advantages and limitations" rather than keeping that as a subsection of the discussion.*

While the author improves the introduction section, there are still some portions that read choppy, mainly because the authors have small paragraphs (2-3 sentences) with their own topic but failed to connect them properly (L92-112). Thus, I ask the authors to revise the later part of the introduction.

*We thank the reviewer for this comment. We revised the later part of the introduction as per the reviewer' suggestion.*

Overall, most of my comments/questions from the revised version were answer by referencing Slater's 2015 workflow. Thus, meaning that my question could be answer in that publication. This represents that the authors need to include more information about Slater's work in this manuscript so the reader does not have to be back and forward between this manuscript and Slater.

*Response: We want to thank the reviewer for this comment. As the work by Slater et al is public and provides also access to the codes used to retrieve and calculate the residuals, we believe that the revised manuscript*

*clarified sufficiently the methods used. We prefer not to add further information regarding Slater's work, to avoid lengthening even further the manuscript.*

**Minor Comments**

- L43: combine that small paragraph (two sentences) with the previous one at L36.

**Response:** *We combined the first two paragraphs in the introduction.*

- L776: The section index should be 4.2 and not 4.3.

**Response:** *We have corrected this.*

- L855-860: If you are focusing in the amount of stream gauges within the east and west region of CONUS do not starting comparing with Alaska and Hawaii since they are not part of CONUS. If you do, then you need to talk about Puerto Rico and American Samoa (both US territories).

**Response:** *Thank you for this comment. We have revised this in the manuscript. Please see below-*
*Line 606- 609: Broadly speaking, the Eastern United States has better coverage compared to its Western counterpart. Particularly, the arid Southwestern United States shows notably lacking spatial coverage. Discrepancies in hydrology contribute to variations in the statistical uncertainty calculated across various parts of the country (Kiang et al., 2013).*

- Figure 1: Please add a description of the arrow and red circle on the figure caption.

**Response:** *We have corrected this.  Please see below:*
*Line 69: Figure 1: Change in channel width in Boulder Creek, Colorado, before (2012) and after (2013-2015-2019) a flash flood in 2013 (Google Earth imagery). The Discharge reported here is the daily discharge measured at USGS 06730200 Boulder Creek at north 75th St. near Boulder, co. The red circles denote the section of the channel that has changed over the years and the blue arrow shows the missing channel from the year 2012 to 2019.*

---

## Author Response (AR3)

**Authors' response to technical edits, egusphere-2023-1969**

We want to thank the editor for pointing out the remaining minor technical edits and for handling the entire revision process.

We adjusted everything as suggested.